# The role of solid solutions in iron phosphate-based electrodes for selective electrochemical lithium extraction

Gangbin Yan[1], George Kim[2], Renliang Yuan[3], Eli Hoenig[1], Fengyuan Shi[4], Wenxiang Chen[3], Yu Han[1], Qian Chen[3], Jian-Min Zuo[3], Wei Chen[2] & Chong Liu[1] ✉

Electrochemical intercalation can enable lithium extraction from dilute water sources. However, during extraction, co-intercalation of lithium and sodium ions occurs, and the response of host materials to this process is not fully understood. This aspect limits the rational materials designs for improving lithium extraction. Here, to address this knowledge gap, we report one-dimensional (1D) olivine iron phosphate ($FePO_4$) as a model host to investigate the co-intercalation behavior and demonstrate the control of lithium selectivity through intercalation kinetic manipulations. Via computational and experimental investigations, we show that lithium and sodium tend to phase separate in the host. Exploiting this mechanism, we increase the sodium-ion intercalation energy barrier by using partially filled 1D lithium channels via non-equilibrium solid-solution lithium seeding or remnant lithium in the solid-solution phases. The lithium selectivity enhancement after seeding shows a strong correlation with the fractions of solid-solution phases with high lithium content (i.e., $Li_xFePO_4$ with $0.5 \leq x < 1$). Finally, we also demonstrate that the solid-solution formation pathway depends on the host material's particle morphology, size and defect content.

Electrochemical lithium extraction with intercalation hosts from dilute water sources shows great potential as an alternative method to secure Li supply and has received tremendous attention lately[1–11]. One-dimensional (1D) olivine iron phosphate ($FePO_4$) is a promising host material owing to its appropriate working potentials, framework stability, thermodynamic Li intercalation preference, and lower Li migration barrier[1,4,7,12,13]. Specifically, the calculated lithiation voltage of olivine $FePO_4$ host is around 3.45 V vs. Li/Li$^+$ (=0.213 V vs. Ag/AgCl), which is higher than the sodiation voltage (3.08 V vs. Na/Na$^+$ = 0.173 V vs. Ag/AgCl)[13]. The migration barrier of Li ion is only 0.17 eV, smaller than that of Na ion (0.29 eV)[13]. Even with the intrinsic material favorability to Li, during electrochemical Li extraction at low Li to Na ratio,

co-intercalation occurs with Na as the main competitor[7,9,13,14]. Despite intriguing proof of concept, the $FePO_4$ host structure response upon Li and Na competitive co-intercalation remains unknown[1,3,4,9]. The intercalation pathways and storage sites are critical in determining the energy barriers for both Li and Na intercalation (including formation enthalpy, migration barrier, nucleation barrier, and interfacial energy), affecting selectivity.

The intercalation behavior of single-component Li or Na in $FePO_4$ hosts has been well studied[13,15–29]. During pure Li intercalation, the pathway depends on the kinetics[21,23,25,28,30–33]. Both theoretical and experimental evidence has shown that at slow (de)lithiation rates, Li-ion intercalation follows the domino-cascade intercalation

[1]Pritzker School of Molecular Engineering, University of Chicago, Chicago, IL 60637, USA. [2]Department of Mechanical, Materials and Aerospace Engineering, Illinois Institute of Technology, Chicago, IL 60616, USA. [3]Department of Materials Science and Engineering, University of Illinois at Urbana-Champaign, Urbana, IL 61801, USA. [4]Electron Microscopy Core, Research Resources Center, University of Illinois Chicago, Chicago, IL 60612, USA. ✉e-mail: chongliu@uchicago.edu

model[20,32,34,35]. At high (de)lithiation rates, phase transformations in nanoparticles can proceed via a continuous change in structure without a distinct moving phase boundary, known as non-equilibrium solid solution (SS) model[25,28,30,36]. Besides the (de)lithiation rates, the impact of particle characteristics (including size[32,37–39], morphology[40], and defect level[21,41]) on the phase transformation of LiFePO$_4$ has also been studied intensively. For Na, high (de)sodiation rates were seldom studied due to the sluggish kinetics[13,26,27]. At slow (de)sodiation rates, according to the phase diagram at room temperature, olivine Na$_y$FePO$_4$ phase separates into FePO$_4$ and Na$_{2/3}$FePO$_4$ for $y < 2/3$ and remains a solid-solution single phase for $y > 2/3$[26]. With both Li and Na, there is competition for the storage sites, making the phase behavior more complex. As a result, accounting for the interaction between Na and Li during co-intercalation is crucial for manipulating the intercalation energy landscape for each ion and controlling the Li competitiveness.

In this study, we demonstrate, via density functional theory (DFT) calculation and direct structural characterization, that Li and Na tend to phase separate in 1D FePO$_4$ hosts. X-ray diffraction (XRD), scanning electron nanodiffraction (SEND), and energy dispersive X-ray spectroscopy (EDS) characterization all showed distinctive Li and Na phases at both the single-particle level and for particle ensembles. Guided by the Li and Na phase separation behavior, we produce Li SS phases with partially filled 1D Li channels via seeding to change the relative intercalation barriers between Na and Li to repel Na. Compared to empty hosts, the Li-seeded hosts showed selectivity increases of

~1.6-fold and ~3.8-fold with 20% and 40% Li-seeding, respectively. With 40% Li-seeding, the total Li fraction (Li/(Li+Na)$_{total}$) in the host is ~94%. The selectivity enhancement strongly correlates with the phase fraction of high-Li SS phases (Li$_x$FePO$_4$, $0.5 \leq x < 1$), but weakly correlates with the phase fraction of low-Li SS phases (Li$_x$FePO$_4$, $0 < x < 0.5$). The high-Li SS phases are effective in preventing the intercalation of Na ions at different specific currents and persist even upon pure Na-ion intercalation. Moreover, we investigate the effects of FePO$_4$ particle forms on the solid solution formation during the Li-seeding step and the consequential Li extraction performance. The defect levels, sizes, and morphologies play critical and complex roles in determining the formations of solid solutions under the same global specific current, which further affects the co-intercalation behavior and Li competitiveness. This work demonstrates the importance of manipulating intercalation kinetic pathways to control ion selectivity and points out that guiding the host phase evolution to undergo Li solid solution formation is an effective strategy to enhance the Li to Na selectivity.

## Results

### FePO$_4$ host phase behavior upon Li and Na co-intercalation

We first used DFT to calculate the formation enthalpies of different structures with Li and Na co-existence. Figure 1a shows the calculated ternary phase diagram of FePO$_4$-LiFePO$_4$-NaFePO$_4$. FePO$_4$, LiFePO$_4$, Na$_{2/3}$FePO$_4$, and NaFePO$_4$ are the ground state structures. Among the 506 structures we calculated with a maximum super cell size of 86, there is no ground state configuration with a mixture of Li and Na,

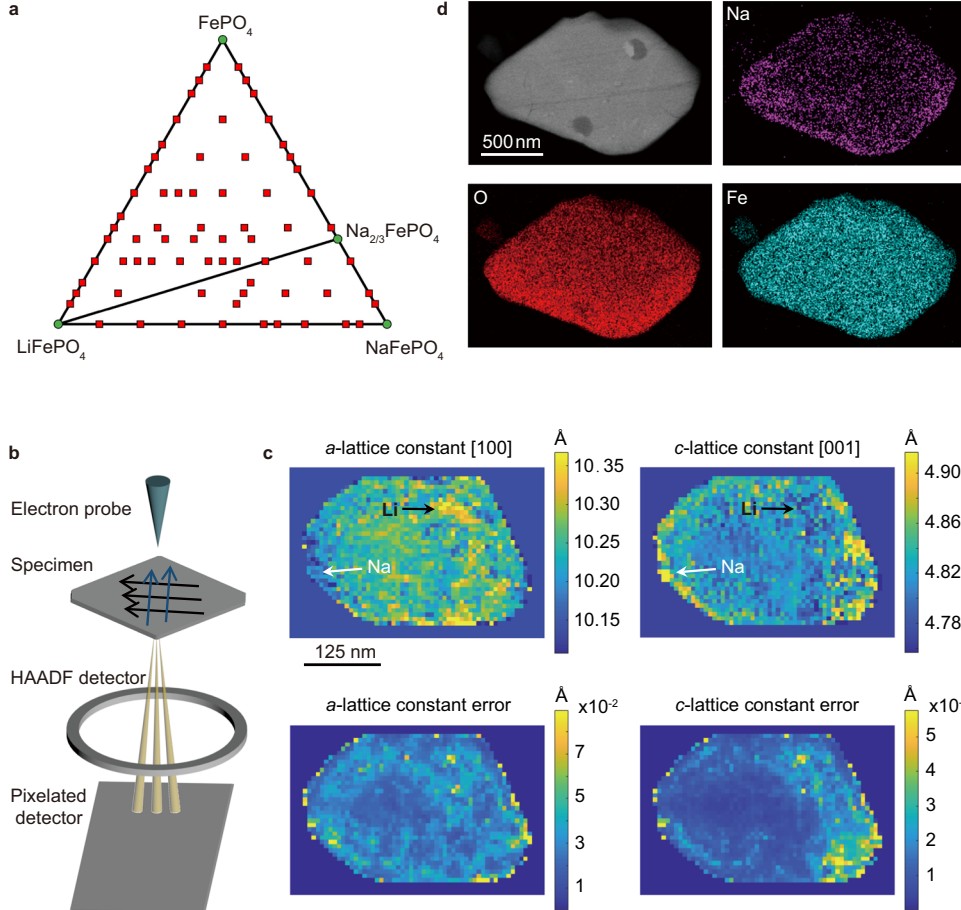

**Fig. 1 | Li and Na phase separation in FePO$_4$ host. a** Ternary phase diagram of FePO$_4$-LiFePO$_4$-NaFePO$_4$. The stable phases (green dots) are FePO$_4$, LiFePO$_4$, Na$_{2/3}$FePO$_4$, and NaFePO$_4$. The red squares denote the calculated intermediate compositions in the Li$_x$Na$_y$FePO$_4$ ($0 \leq x+y \leq 1$) system. **b** Schematic illustrating the setup of scanning electron nanodiffraction. The blue and black arrows denote the scanning motion of the electron beam. **c** Absolute $a$, $c$-lattice constant maps of the LN(0.7)$_{0.1C}$ particle with their estimated errors (arrows are a guide-to-the-eye for the phase identification). **d** STEM image and its corresponding EDS elemental mapping (Na, O, Fe) of the LN(0.7)$_{0.1C}$ particle.

indicating that Li and Na do not prefer to co-exist in the same [010] channel during co-intercalation (See Supplementary Note 1 for computation details). Intermediate compositions are expected to decompose into these four ground state phases depending on the composition.

To verify the calculation results experimentally, we first used scanning electron nanodiffraction (SEND) (Fig. 1b, c and Supplementary Fig. 1) to identify existing phases at a single-particle level. The particles were prepared by directly intercalating $FePO_4$ hosts in a 1 mM LiCl and 1 M NaCl (molar ratio Li: Na = 1: 1000) mixed solution under 0.1 C (14.7 mA/g) using 70% of the total capacity (The total capacity is 147 mAh/g, measured by cycling the electrodes in 60 mL 1 M LiCl aqueous solutions between −0.6 V and 0.6 V (vs. Ag/AgCl/KCl (4.0 M)) at room temperature (20-25 °C) under 0.1 C, labeled as $LN(0.7)_{0.1C}$. The ratio of 1: 1000 Li: Na is selected based on the compositions of brines and geothermal fluids[7,9,42–47]. The platelet particles are observed along [010] channel direction. We constructed the lattice constant maps with the estimated errors as shown in Fig. 1c. Lattice constants for potentially related Li or Na phases are summarized in Supplementary Table 1. The lattice constant range points to the co-existence of olivine $LiFePO_4$ (*a*, *b*, *c*-lattice constants = 10.329, 6.009, 4.695 Å) and olivine $Na_{2/3}FePO_4$ (*a*, *b*, *c*-lattice constants = 10.289, 6.082, 4.937 Å) phases and ruled out the existence of olivine $FePO_4$ (*a*-lattice constant = 9.821 Å), olivine $NaFePO_4$ (*a* = 10.406 Å; *c* = 4.947 Å) and maricite $NaFePO_4$ (*a* = 9.001 Å; *c* = 5.052 Å) phases, since their lattice constants are out of the range. As shown in the SEND mapping, $LiFePO_4$ phase mainly exists in the center of the particle (yellow areas in Fig. 1c, *a*-lattice constant map), and $Na_{2/3}FePO_4$ phase is mainly at the edges (yellow areas in Fig. 1c, *c*-lattice constant map) as guided by the arrows. The matching of lattice constants to pure Li and Na phases supports the Li and Na phase separation behavior. Meanwhile, lattice strains were calculated and shown in Supplementary Fig. 1. The contraction and expansion of *c*-lattice parameter in the [001] direction clearly show the results of Li and Na ions intercalating separately. It is also worth noting that from *c*-lattice constant map (Fig. 1c), we observed Na SS phases with intermediate *c*-lattice constants (0.482 nm-0.486 nm). The formation of metastable Na SS phases at 0.1 C was confirmed with pure Na-ion intercalation. XRD shows that the Na SS phase separated into $Na_{2/3}FePO_4$ and $FePO_4$ after one month of storage (Supplementary Fig. 2). Therefore, 0.1 C (14.7 mA/g) is already a high current rate to drive a solid solution pathway between $FePO_4$ and $Na_{2/3}FePO_4$ phases. Secondly, scanning transmission electron microscopy (STEM) with energy-dispersive X-ray spectroscopy (EDS) was used to map the elemental distribution (Fig. 1d). Na mapping showed nonuniform signal across the particle with higher intensity near the edges. The EDS mapping matches well with the SEND phase analysis and reveals the existence of $Na_{2/3}FePO_4$ near the edge of the particle (yellow areas in Fig. 1c, *c*-lattice constant mapping). At the electrode level for particle ensembles, the XRD pattern of the $LN(0.7)_{0.1C}$ electrode also showed a co-existence of $LiFePO_4$ and $Na_{0.71}FePO_4$ phases (Supplementary Fig. 3).

## Pre-seeding Li and quantification of SS phase fractions

Guided by the results that Li and Na tend to phase separate during co-intercalation, we propose to enhance the Li competitiveness in $FePO_4$ hosts by creating partially filled Li 1D channels to raise the Na phase formation energy barrier. The typical electrochemical (de)intercalation pathway of Li in $Li_xFePO_4$ particles with a sub-micrometer diameter at low specific currents (≤0.1 C) undergo phase separation during the vast majority of the process ($0.05 \leq x \leq 0.95$)[28,35,48]. To create partially filled Li 1D channels at room temperature (20-25 °C), we seed Li via the SS phase change pathway at high C rates (>1 C). The proposed seeding process is illustrated in Fig. 2a. We first extract Li from the host by chemical deintercalation of $LiFePO_4$ (See Methods for chemical extraction details). It is worth noting that depending on the defect

level of the synthesized $FePO_4$ particles, some Li can be trapped in the 1D channels as remnant Li. The host was then seeded a target amount of Li in 1 M $LiCl_{(aq)}$ solution under high C rates (>1 C) and labeled as "$L(X)_{nC}$", where X is the percentage of Li filled in $Li_xFePO_4$ and n is the C rate used in the seeding process. Specifically, the total capacity is 147 mAh/g and 1 C is equivalent to 147 mA/g (See Methods for more details). An ideal case would be for most of the channels to have some Li to repel Na.

The Li SS phases and their fractions were quantified by XRD characterization after seeding. Figure 2b shows the normalized XRD patterns of $FePO_4$ electrodes after seeding different amounts of Li (0, 10%, 20%, 30%, and 40% of the 147 mAh/g total capacity) under 4 C (588 mA/g), labeled as $L(0/0.1/0.2/0.3/0.4)_{4C}$ respectively (See Methods for sample preparation details). The strong intensity band between the characteristic (020) peaks for $FePO_4$ and $LiFePO_4$ indicates the formation of SS phases with a continuous structural change[18,21,31,49]. With more lithium seeded, the peak intensity ratios of SS phases to $FePO_4$ and $LiFePO_4$ to $FePO_4$ increase. Following previous work[48,49], we fit a sum of Gaussians to the diffraction patterns, deconvolving seven intermediate phases $Li_xFePO_4$, x = 0.125/0.250/0.375/0.500/0.625/0.750/0.875, from $FePO_4$ and $LiFePO_4$ end phases (See Supplementary Tables 2, 3, 4, and Supplementary Note 2 for deconvolution details). We fixed (within a pre-defined window) the center location of each Gaussian based on linear combinations of the refined phases for $LiFePO_4$ and $FePO_4$ (Supplementary Fig. 4). Figure 2c shows an example of a deconvoluted XRD pattern. The high $R^2$ value confirms the validity of our deconvolution method. The fitted accumulative phase fractions for samples representing the starting and four seeding ranges are summarized in Fig. 2d. The calculated weighted sum of Li from XRD fittings shows a good linear relationship with the electrochemical seeded Li amount (Supplementary Figs. 5, 6, and Supplementary Table 5). However, Li amounts showed deviations at the low seeding percentage. For example, before Li seeding (L(0)), the weighted sum of Li is -0.17 (Supplementary Table 5). This deviation could be from the remnant Li trapped in the hosts after chemical extraction or the contributions from the system substrate effect when the background intensity is unignorable. The deviations caused by porous structures of carbon cloth can be decreased by using flat glassy carbon as the substrate. It is worth mentioning that flexible carbon cloth is a better choice considering manufacturing and practical use. As shown in Supplementary Figs. 7, 8 and Supplementary Table 6, a better quantitative agreement between the calculated weighted sum of Li and the seeding amount is achieved. By eliminating the background intensity contributions, we can see from Supplementary Table 6 that -0.07 Li per formula was left in the host after the chemical extraction step before the seeding process which could be from defect-induced Li trapping (See Supplementary Note 3 for more discussions of the deviations).

## Correlation of high-Li SS phases to Li selectivity

The effect of seeded Li SS phases on Li selectivity was investigated using 1: 1000 Li to Na molar ratio solution (1 mM LiCl and 1 M NaCl mixed solution). It should be noted that two different Li/(Li+Na) ratios are reported. One is Li/(Li+Na)$_{total}$, which denotes the ratio of the total amount of Li detected in the recovery solution after emptying the host, and another one is Li/(Li+Na)$_{net}$, which subtracts the initially seeded Li (See Methods for more details about the calculation of recovered Li ratios). First, we seeded different amounts of Li (10%, 20%, 30% and 40% of the 147 mAh/g total capacity) under 4 C (588 mA/g). The total SS fraction increased monotonically with the seeding amount (Supplementary Fig. 9a). After seeding, we conducted Li extraction in 1: 1000 Li to Na molar ratio solution under 0.1 C (14.7 mA/g) until 70% of capacity (102.9 mAh/g) was used, which we label as $L(0.1/0.2/0.3/0.4)_{4C}$-$LN(0.7)_{0.1C}$. Both Li/(Li+Na)$_{total}$ and Li/(Li+Na)$_{net}$ showed a monotonic increase (Supplementary Fig. 9a), indicating the

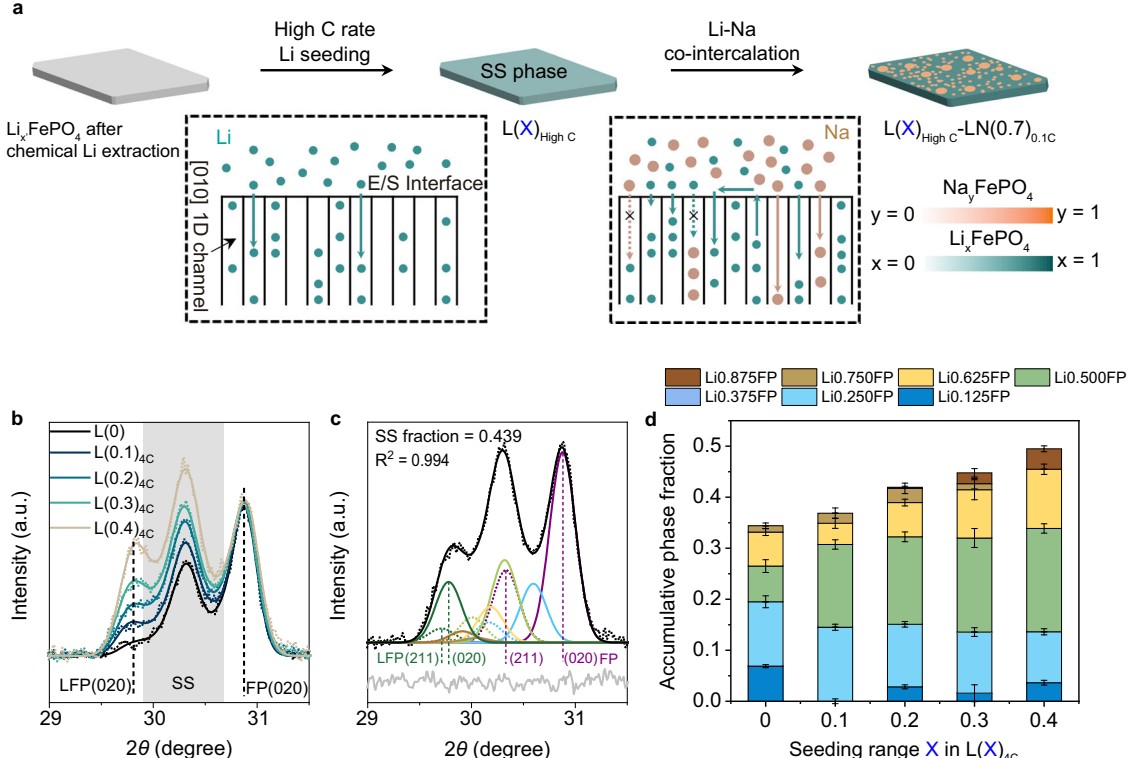

**Fig. 2 | Seeding and quantification of Li SS phases. a** Schematic of the Li seeding and Li/Na ions co-intercalation processes happening at high seeding C rates (>1 C). The inset illustrates the possible intercalation pathways at the electrode-electrolyte solution (E/S) interface. The initial host was prepared by chemical extraction. x' denotes the remnant quantity of Li in the structure. See "Pre-seeding Li and quantification of SS phase fractions" in the main text for more information. **b** Normalized XRD (dotted line: raw data; solid line: fit) patterns of FePO$_4$ electrodes before (L(0)) and after seeding with different amounts of Li (10%, 20%, 30% and 40% of the 147 mAh/g total capacity) under 4 C (588 mA/g), labeled as L(0.1/0.2/0.3/0.4)$_{4C}$. The normalization is based on the intensity of (020) peak for FePO$_4$ at 30.9°.

The (020) peak of LiFePO$_4$ is centered at 29.8°. The intensity bands between the two end-up phases are the intermediate SS phases. **c** An example of deconvoluted XRD pattern for the quantification of SS phases and corresponding R-squared value (R$^2$). The obtained pattern (black dots) of L(0.3)$_{4C}$ is fitted with nine different phases of Li$_x$FePO$_4$ with $_x$ = 0/0.125/0.250/0.375/0.500/0.625/0.750/0.875/1, as calculated based on Vegard's law for the (211) and (020) of the LiFePO$_4$ (Green) and FePO$_4$ (Purple) end phases. See Supplementary Note 2 for more fitting details. **d** Averaged accumulative SS phase fractions of L(0/0.1/0.2/0.3/0.4)$_{4C}$. (Error bars representing the standard deviation of three replicate measurements).

effectiveness of pre-seeding strategy in promoting Li competitiveness. Specifically, the Li/(Li+Na)$_{net}$ ratio increased from 0.61 ± 0.01 to 0.86 ± 0.01 from 10% to 40% seeding. With 40% seeding, we achieved ~3.8 fold increase of Li selectivity to 6.0 × 10$^3$, comparing to the empty host without seeding. However, we inevitably need more cycle repetition to obtain the same amount of lithium transferred with a Li pre-seeded host. Further improvement of the structrual response of the FePO$_4$ electrode could improve the capacity usage.

We then go on to examine whether all the seven intermediate SS phases are equivalently effective in enhancing the Li selectivity. First, we analyzed the Li selectivity trend to each SS phase (Supplementary Fig. 9b). None of the phases alone can explain the selectivity trend. We then divided the SS phases into two groups, the low-Li SS phases (Li$_x$FePO$_4$, x = 0.125/0.250/0.375) and the high-Li SS phases (Li$_x$FePO$_4$, x = 0.500/0.625/0.750/0.875) (Fig. 3a). The Li content in the SS phases should affect the energy barrier for Na-ion intercalation, since Li and Na intercalation into the FePO$_4$ cause contraction and expansion in the [001] direction, respectively. After grouping, we observe a clear trend that the selectivity increase follows the high-Li SS phase fraction increase for fitting data on both carbon cloth and glassy carbon substrates (Fig. 3a, b and Supplementary Fig. 8). However, the low-Li SS phases did not show monotonic trend. The correlation shows that high-Li SS phases contribute the most to the improvement of Li selectivity. Interestingly, when the seeding range was small at 10%, L(0.1)$_{4C}$-LN(0.7)$_{0.1C}$, the Li/(Li+Na)$_{net}$ was comparable with the empty host (L(0)-LN(0.7)$_{0.1C}$) (Fig. 3b). To explain this, we tested the capacity

range dependent selectivity of the empty FePO$_4$ host (Fig. 3c). The Li/(Li+Na) ratio is 0.92 ± 0.02 for the first 10% capacity and the Li/(Li+Na) ratio decreases with the increase of capacity usage (all 0.1 C, equivalent to 14.7 mA/g). The intrinsic solubility of Li in FePO$_4$ hosts (as SS phase) is around 5%$^{50}$. If the first 5% of the FePO$_4$ undergoes SS phase change, with >90% selectivity, this is equivalent to the 4.5% Li seeding process, which can promote the Li selectivity. The effect of seeding C rate on SS phase fractions and Li selectivity was also examined at 20% and 40% seeding conditions. In Supplementary Fig. 10a, at a slow seeding rate of 0.1 C (14.7 mA/g), the Li/(Li+Na)$_{net}$ of L(0.2)$_{0.1C}$-LN(0.7)$_{0.1C}$ was 0.66 ± 0.03, similar to that of empty FePO$_4$ hosts (0.63 ± 0.01), showing that, for the slow-rate Li pre-seeded hosts, the domino-cascade model starts to play a role, which opens more free channels for later Na-ion intercalation in the mixed solution (Supplementary Fig. 11). With increasing seeding C rates, from 2 C (294 mA/g) to 8 C (1176 mA/g), the Li/(Li+Na)$_{net}$ witnessed a monotonous increase (Supplementary Fig. 10a) with a 20% of seeding range, which is consistent with the corresponding high-Li SS fraction in each case (Supplementary Fig. 10b). We also tested the electrodes with 40% of seeding under different seeding C rates. As shown in Supplementary Fig. 10c, comparing to 20% seeding, we see a clearer trend that increasing the seeding C rate would lead to better Li selectivity. For the L(0.4)$_{8C}$-LN(0.7)$_{0.1C}$ case, we achieved the highest Li selectivity (Li$_{selectivity}$ = 1.48 × 10$^4$) in this work (Supplementary Table 7). The applied high seeding specific current induces concurrent, non-mosaic intercalation in the porous electrode$^{48}$.

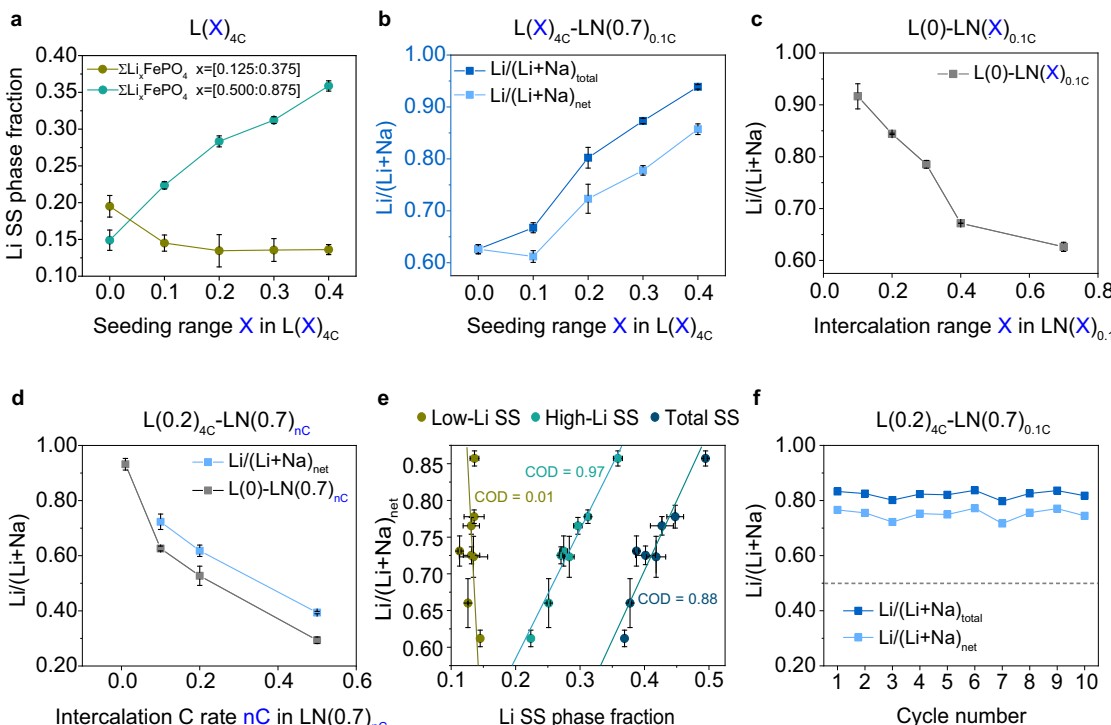

**Fig. 3 | Correlation between SS phase fractions and lithium extraction performance. a** High-Li SS fractions ($Li_xFePO_4$, x = 0.500/0.625/0.750/0.875) and low-Li SS fractions ($Li_xFePO_4$, x = 0.125/0.250/0.375) under the same seeding rate 4 C (588 mA/g) with different seeding ranges $L(0/0.1/0.2/0.3/0.4)_{4C}$. **b** $Li/(Li+Na)_{total}$ (including the contributions from seeded-Li) and $Li/(Li+Na)_{net}$ (subtracting the contributions from seeded-Li) of the same hosts from **a** after 0.1 C (14.7 mA/g) co-intercalation $L(X)_{4C}-LN(0.7)_{0.1C}$. **c** $Li/(Li+Na)$ ratio of the empty hosts with different intercalation range $L(0)-LN(X)_{0.1C}$. **d** $Li/(Li+Na)_{net}$ of the electrodes with the same seeding process $L(0.2)_{4C}$ but different intercalation rates $L(0.2)_{4C}-LN(0.7)_{0.1/0.2/0.5C}$, and $Li/(Li+Na)$ ratio of the empty hosts $L(0)-LN(0.7)_{0.01/0.1/0.2/0.5C}$. **e** $Li/(Li+Na)_{net}$ vs. low-Li/high-Li/total-Li SS fractions and corresponding COD. **f** $Li/(Li+Na)_{total}$ and $Li/(Li+Na)_{net}$ for the multi-intercalation stability test using $L(0.2)_{4C}-LN(0.7)_{0.1C}$. (Error bars representing the standard deviation of three replicate measurements).

We further investigated the effectiveness of high-Li SS phases in restricting Na-ion intercalation at different overpotentials. Higher overpotentials at larger currents can provide additional energy for Na to overcome its intercalation barrier and decrease the Li selectivity. At an slow current of 0.01 C (1.47 mA/g), the $Li/(Li+Na)_{net}$ molar ratio of $L(0)-LN(0.7)_{0.01C}$ was $0.93 \pm 0.02$ (Fig. 3d). This selectivity benefits from the higher Li-ion intercalation potential and lower Li migration barrier. The thermodynamic preference of Li-ion intercalation could compensate for more than three order of magnitudes molar concentration difference between Li and Na[7]. Without Li seeding, with increased intercalation C rates from 0.1 C (14.7 mA/g) to 0.5 C (73.5 mA/g), $Li/(Li+Na)$ of empty host decreased drastically from $0.63 \pm 0.01$ to $0.32 \pm 0.01$, as a result of higher intercalation overpotentials -0.22 V (final potential difference, Supplementary Fig. 12a). The increased co-intercalation C rates may also cause local depletion of Li on the cathode surface[6,9]. With Li seeding, as shown in Fig. 3d, at 0.2 C (29.4 mA/g) intercalation rate, the hosts could still maintain decent selectivities of $Li/(Li+Na)_{net} = 0.62 \pm 0.02$, compared to the case without seeding ($Li/(Li+Na)_{net} = 0.53 \pm 0.03$). From 0.1 C (14.7 mA/g) to 0.5 C (73.5 mA/g) intercalation rate, the seeding process was consistently promoting the Li competitiveness. This indicates that the high-Li SS phases are effective in preventing Na-ion intercalation even at fast kinetics and can tolerate a higher overpotential of at least -0.21 V (Supplementary Fig. 12b).

To illustrate the relationships between Li competitiveness and Li SS fractions, without the differentiation of seeding conditions, we plot $Li/(Li+Na)_{net}$ (Fig. 3e) and $Li/(Li+Na)_{total}$ (Supplementary Fig. 13) versus high-Li, low-Li, and total SS phase fractions for all the Li-seeded samples, with co-intercalation step run at 0.1 C (14.7 mA/g). Coefficients of determination (COD) were calculated to quantify the correlations. The fraction of high-Li SS phases is strongly correlated to Li ratios (e.g., COD ($Li/(Li+Na)_{net}$ vs. high-Li SS) = 0.97). In contrast, the fraction of low-Li SS phases is uncorrelated with the Li ratios (e.g., COD ($Li/(Li+Na)_{net}$ vs. low-Li SS) = 0.01). Meanwhile, the $L(0.2)_{4C}$ seeding condition was tested for multiple cycles on one electrode. Each cycle has Li seeding, Li extraction, and Li recovery steps. We measured the Li selectivity for each cycle. As shown in Fig. 3f, the $Li/(Li+Na)_{net}$ is maintained at ~0.73, proving the effect of seeding to improve Li selectivity as well as cycle stability. Additionally, we conducted two more cycling tests. Shown in Supplementary Fig. 14, we skip the Li seeding step from the 2nd cycle and use either 70% or 50% of capacity for further Li extraction. A decrease of the recovered $Li/(Li+Na)$ was observed without seeding from the 2nd cycle, indicating that the seeding effect can only work for one cycle.

### High-Li fraction SS phases promoting Li competitiveness

To further investigate the role of Li SS phases to Na-ion intercalation, we directly intercalated pure Na ions (1 M NaCl) in Li-seeded $L(0.2)_{4C}$ host to $L(0.2)_{4C}-N(0.5)_{0.1C}$ (XRD shown in Fig. 4a and Supplementary Fig. 15). Besides the expected increase of the $Na_{2/3}FePO_4$ phase and the decrease of empty $FePO_4$ phase, we saw a pronounced increase of the relative peak intensity at the $LiFePO_4$ position and a shift of the SS band to high-Li positions (Fig. 4a). To confirm the phases are $LiFePO_4$ and high-Li SS phases, we prepared $N(0.5)_{0.1C}$, $N(0.7)_{0.1C}$ and $N(0.5)_{0.1C}$-$L(0.2)_{0.1C}$ (first intercalating 50% Na and then intercalating 20% Li; 0.1 C is equivalent to 14.7 mA/g) for comparison (Fig. 4b). Both $N(0.5)_{0.1C}$ and $N(0.7)_{0.1C}$ showed a broad band between $Na_{2/3}FePO_4$ (020) and $FePO_4$ (020), indicating a SS phase transition pathway for Na-ion intercalation at 0.1 C. However, the SS bands of $N(0.5)_{0.1C}$ and $N(0.7)_{0.1C}$ do not overlap at the position of $LiFePO_4$. The onset of peak

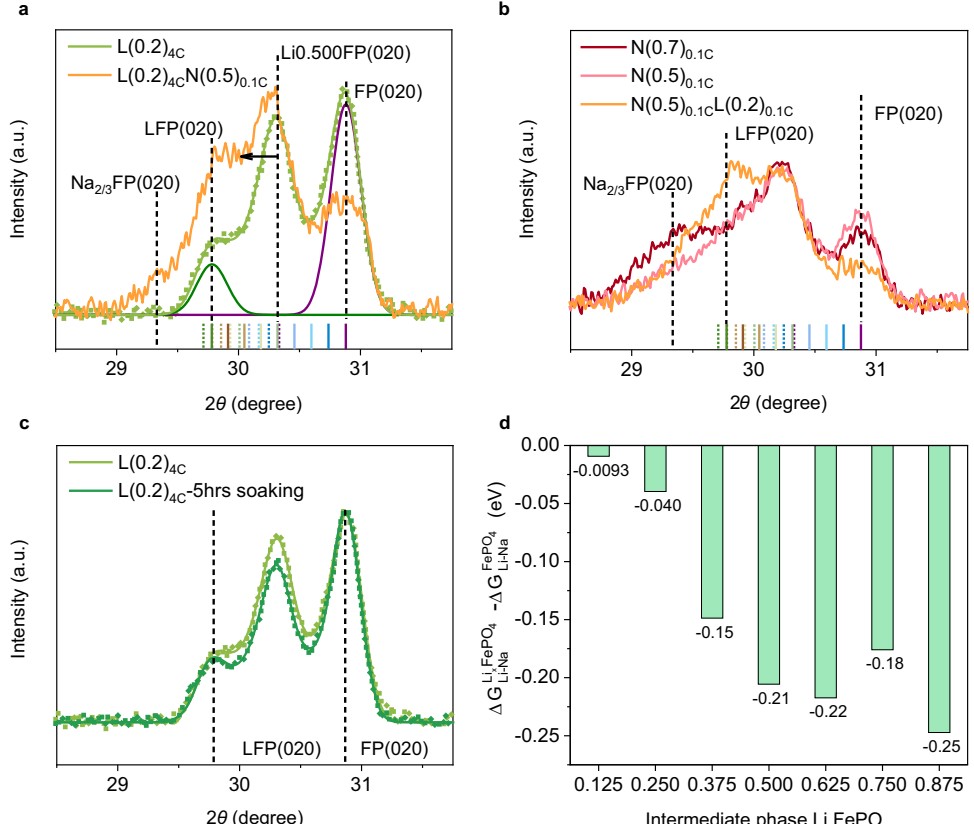

**Fig. 4 | Effect of high-Li SS phases in promoting Li competitiveness. a** XRD of L(0.2)$_{4C}$ electrode before (light green dots: raw data; solid light green line: fit) and after (orange: obtained pattern) 50% Na insertion. Dark green and purple peaks are fitted LFP and FP (020) peaks for L(0.2)$_{4C}$ to guide the peak positions. Bottom bars are the positions of (020) and (211) peaks of nine Li$_x$FePO$_4$, x = 0/0.125/0.250/ 0.375/0.500/0.625/0.750/0.875/1. **b** Raw XRD patterns of N(0.7)$_{0.1C}$ electrode (dark red), as well as N(0.5)$_{0.1C}$ electrode before (light red) and after (orange) 20% Li insertion. **c** XRD of L(0.2)$_{4C}$ electrode before (light green) and after (dark green) 5 h soaking in 1 M NaCl$_{(aq)}$ (Dotted lines: raw data; Solid lines: fit). **d** DFT calculation of Li and Na ions intercalation energy difference for each intermediate phase Li$_x$FePO$_4$ with respect to empty FePO$_4$ host.

around the LiFePO$_4$ and high-Li SS positions only occurs with Li-ion intercalation (also see N(0.5)$_{0.1C}$-L(0.2)$_{0.1C}$). This proves the formation of LiFePO$_4$ phase after pure Na-ion intercalation into pre-seeded L(0.2)$_{4C}$. We further confirmed using ICP-MS that the total amount of Li before and after Na-ion intercalation did not change. Therefore, the formation of LiFePO$_4$ and the increase of high-Li SS phases are caused by the rearrangement of original Li ions responding to Na-ion intercalation. Since the total Li amount does not change, the Li has to come from the low-Li SS phases. One possible pathway for the rearrangement is through Li ions moving out of the low-Li SS phases and adding to the high-Li SS phases[48]. These results support that high-Li SS phases are more stable than low-Li SS phases upon Na-ion intercalation. Meanwhile, without Na-ion intercalation, simply soaking the electrode in the NaCl$_{(aq)}$ solution for the same amount of time as Na-ion intercalation, the low-Li and high-Li SS phase fractions only showed a slight decrease (Fig. 4c and Supplementary Fig. 16), which could be attributed to the solvent assisted surface-ion diffusion at FePO$_4$[48]. The extent of Li rearrangement is much less than that in the case of Na-ion intercalation. Therefore, the significant Li rearrangement is mainly a response of Li phases upon Na competition. With the persistence of high-Li SS phases and diminish of low-Li phases, it reveals that the pre-seed Li SS phases are not equally effective in expelling Na ions during co-intercalation. The high-Li SS phases are more effective than the low-Li SS phases.

Moreover, we use DFT to calculate the energy barrier differences between Li and Na ions intercalation in each intermediate SS phase (See Supplementary Note 4 for calculation details) to prove the effect of high-Li SS phases in promoting Li competitiveness. First, we established low energy supercell configurations for the seven intermediate phases, as shown in Supplementary Fig. 17. The channel filling information was summarized in Supplementary Table 8. From the calculation results, all seven Li SS phases, as well as the empty FePO$_4$ host, showed negative $\Delta G_{Li-Na}$ (Supplementary Table 9), which means that thermodynamically Li-ion intercalation is always preferred in all the structures of the 1D olivine host. The Li and Na intercalation energy difference clearly showed that all Li SS phases have enhanced Li preference, compared to the empty FePO$_4$ phase, as ($\Delta G_{Li-Na}^{Li_xFePO_4} - \Delta G_{Li-Na}^{FePO_4}$) are negative (Fig. 4d). Between low-Li SS phases and high-Li SS phases, high-Li SS phases have more negative ($\Delta G_{Li-Na}^{Li_xFePO_4} - \Delta G_{Li-Na}^{FePO_4}$), supporting the conclusion that high-Li SS phases being more effective in promoting Li-ion intercalation competitiveness. We also compared the Li and Na intercalation potential difference of both the empty host and the 50% Li pre-seeded host under 4 C (588 mA/g). As shown in Supplementary Fig. 18, the potential difference ($\Delta V_1$) is 0.23 V for empty host and is 0.27 V ($\Delta V_2$) for L(0.5)$_{4C}$ host. The exhibited Li preference is stronger for L(0.5)$_{4C}$, than the empty host ($\Delta V_2 > \Delta V_1$). Therefore, both experimental and computational results show the same trend of Li preference for Li pre-seeded hosts.

## Effects of particle characteristics on solid solution seeding and Li selectivity

The ion insertion reaction of Li$_x$FePO$_4$ with different particle characteristics has been studied intensively. The formation of solid

solution in olivine $FePO_4$ is a complicated process that is affected by many factors, such as the temperature[18,26], particle size/morphology[32,38–40], applied current[25,28–30,51], and defects[21,41]. The earliest reports have shown that the (de)intercalation of $Li^+$ goes through a phase separation reaction into Li-rich and Li-poor phases at room temperature (20-25 °C)[15,52]. At elevated temperature (>400 °C), single-phase reaction was observed in the whole composition range ($0 < x < 1$ in $Li_xFePO_4$)[26]. In addition, the miscibility gap has been found to reduce with the reduction of particle size, even vanish when the particles reach the critical nano-size region ($d_c \leq 22$ nm)[32,38,39]. Moreover, both computational[25,30] and experimental[28,29] results have demonstrated that, at elevated (de)lithiation rates, phase separation is suppressed and replaced with a solid solution pathway. Particles with different morphologies may also have various response to the same applied current, even with similar particle size. It is demonstrated that platelet particles have a much lower exchange current than ellipsoidal particles, which would increase the active particle population and promote uniform solid-solution domains[29,51]. Importantly, defects play a significant role in controlling the intercalation phase transformation pathway. It is found that particle size can be considered as a good but not sufficient condition to anticipate single phase solid solution formation[21]. Different amounts of non-stoichiometry and cationic mixing could lead to different phase transformation, even with the same particle size. With considerable Li/Fe disorder, solid solution formation in the whole composition range can be realized[21].

We then investigate the effect of particle characteristics on the formation of solid solutions and Li competitiveness. We studied two other particles with various forms. One is commercial $LiFePO_4$ particles (Comm-$LiFePO_4$), which were bought from MTI Corporation (Item Number: Lib-LFPOS21). As shown in Supplementary Fig. 19, the average dimension of the primary ellipsoidal shape particles is ~430 nm. Additionally, the dimension of the secondary particles is ~2.93 μm. As a comparison, Supplementary Fig. 20 shows the original particles (Synthesized-$LiFePO_4$) which we used for all the experiments unless mentioned. The lateral dimension along the long axis is ~1.93 μm, with the [010] channel length ~270 nm. Besides, we synthesized another platelet-like particle (EG-$LiFePO_4$) as shown in Supplementary Fig. 21 with smaller lateral (~97 nm) and thickness dimensions (~0.50 μm). All the particles have dimensions for the migration direction below 1 μm. Both Comm-$LiFePO_4$ particles and EG-$LiFePO_4$ particles follow the same chemical Li extraction process to prepare empty $FePO_4$ hosts (See Methods for more details).

Comm-$FePO_4$, Synthesized-$FePO_4$, and EG-$FePO_4$ showed capacity of 135 mAh/g, 147 mAh/g, and 125 mAh/g under 0.1 C, using 13.5 mA/g, 14.7 mA/g, and 12.5 mA/g respectively (The total capacity is measured by cycling the electrodes in 60 mL 1 M LiCl aqueous solutions between −0.6 V and 0.6 V (vs. Ag/AgCl/KCl (4.0 M)) at room temperature (20-25 °C)). It is worth mentioning that not all particles are good battery quality particles, especially EG-$FePO_4$. Their low delivered capacity can be correlated to the presence of the high level of defects. Supplementary Fig. 22 shows example XRD patterns collected on flat glassy carbon for Comm-$FePO_4$, Synthesized-$FePO_4$, and EG-$FePO_4$ electrodes with 20% Li seeding under 4 C (540 mA/g for Comm-$FePO_4$, 588 mA/g for Synthesized-$FePO_4$, and 500 mA/g for EG-$FePO_4$). We seeded 20% Li for each type of particles and summarized the solid solution fractions in Supplementary Fig. 23a. In the case of Comm-$FePO_4$ ellipsoidal shape particles, the total SS fraction generated is the least (Total SS = 0.234 ± 0.002) with the intercalated Li ions mainly formed fully occupied $LiFePO_4$ phase, as indicated by the pronounced $LiFePO_4$ (020) peak in Supplementary Fig. 22a. And the calculated weighted sum of Li is around 17% from the XRD pattern, close to the 20% Li seeding. The ellipsoidal shape and low defect level of Comm-$FePO_4$ particles could increase the miscibility gap, which can suppress solid solution formation[29,32,38,39,51]. Meanwhile, we achieved the lowest

selectivity with Comm-$FePO_4$ particles ($Li/(Li+Na)_{net} = 0.26 \pm 0.01$, in Supplementary Fig. 23b).

For the platelet-like EG-$LiFePO_4$ particles, we noticed that there is ~23.5% Li remained in the host after chemical Li-extraction process (Supplementary Table 10). The chemical route initially used cannot remove $Li^+$ completely, suggesting the presence of defects in the EG-$FePO_4$ particles which is consistent with their lower capacity. It is worth noting that these trapped Li species are in the form of solid solution phases, as evidenced by the strong solid solution intensity band in Supplementary Fig. 24a. Furthermore, the XRD patterns and the fitted phase fractions didn't change after Li recovery deintercalation (Supplementary Fig. 24 and Supplementary Table 11), suggesting a lack of mobility of these defect-induced trapped Li ions during (de)intercalation. The EG-$FePO_4$ particle shows a slightly higher Li selectivity (Supplementary Fig. 23b) after 20% seeding and without Li seeding ($Li/(Li+Na)_{net} = 0.81 \pm 0.01$ for $L(0.2)_{4C}$-$LN(0.7)_{0.1C}$ and $Li/(Li+Na)_{net} = 0.74 \pm 0.01$ for $L(0)$-$LN(0.7)_{0.1C}$), compared with Li pre-seeded Synthesized-$FePO_4$ particle ($Li/(Li+Na)_{net} = 0.72 \pm 0.03$ for $L(0.2)_{4C}$-$LN(0.7)_{0.1C}$).

For the comparison among the three types of particles, the Li selectivity shows the same trend as the high-Li solid solution fractions for 20% seeded samples (Supplementary Fig. 23). However, the high-Li solid solution fraction to Li selectivity correlation across different particles might not follow the same linear relationship. Therefore, when comparing different particles, morphologies, sizes, and defect levels have to be taken into consideration since they can act together and play a complex role in determining the intercalation pathway and phase formation. Besides solid solution formation, some other aspects may also affect Li/Na selectivity. For example, coherency strain energy has different anisotropies and magnitude when changing from $FePO_4$ to $Li_xFePO_4$ or $Na_xFePO_4$, dependent on the particle size and the particle morphology, which may also be a significant factor for Li competitiveness[40]. Here, we demonstrated that production of Li solid solution phases is an effective strategy to improve the Li selectivity beyond the intrinsic thermodynamic and kinetic material preference to Li. More systematic studies on other structural factors could bring new opportunities in the future to facilitate the Li extraction process.

## Discussion

In summary, both DFT calculation and structural characterization reveal that our model 1D $FePO_4$ host tends to phase separate upon Li and Na ions co-intercalation. Benefiting from the phase separation, we improve the Li ions co-intercalation competitiveness by creating Li SS phases. Li SS phases restrict Na-ion intercalation and enhance Li-ion intercalation preference. We show both experimentally and theoretically that high-Li and low-Li SS phases are not equivalently effective in repelling Na. High-Li SS phases exhibit a strong correlation with Li selectivity enhancement and are more effective in promoting Li competitiveness. Moreover, particle characteristic is a critical factor in determining the solid solution formation, which affects co-intercalation behavior and Li selectivity. In our study, with a similar level of seeding, the solid solution fractions are higher in the platelet particles synthesized but lower for commercial ellipsoidal-shaped particles. Additionally, as the defect level in the particles increases, the trapped remnant Li amount increases in the host. These trapped Li exists as solid solution phases and leads to a higher Li selectivity. However, the influence of material characteristics (including morphologies, sizes, and defects) can be complex and requires future work to reveal their interplays. These insights highlight the importance of manipulating the co-intercalation kinetic pathways for controlling the Li selectivity.

## Methods

### Synthesis of $FePO_4$ microplatelets

To synthesize pristine $LiFePO_4$ microplatelets, a solvothermal method with a mixed water and polyethylene glycol solvent was used, modified

slightly from the previous report[48]. All the operations were done in an $N_2/H_2O$ glovebox (<1 ppm $O_2$) to ensure that all precursors are not exposed to oxygen. 6 mL of 0.2 M $H_3PO_{4(aq)}$ was mixed with 24 ml of polyethylene glycol 400. Afterward, 18 mL of 0.2 M $LiOH_{(aq)}$ was added to produce the creamy-white $Li_3PO_4$ precipitate. This mixture was stirred in an $N_2$ glovebox overnight to remove dissolved oxygen. 1.2 mmol of $FeSO_4 \cdot 7H_2O$ was dried under vacuum in a Schlenk line overnight, while 12 mL of $H_2O$ was stored in the $N_2$ glove box for 24 h to deoxygenate. Next, the deoxygenated $H_2O$ was transferred to the dried $FeSO_4$ powder and stirred for about 10 min, creating a lime-green solution. The $FeSO_4$ solution was transferred to the $Li_3PO_4$ suspension without oxygen exposure, and the entire mixture was transferred to a 100 mL Teflon-lined autoclave. The autoclave was heated to 140 °C for 1 h, then to 210 °C for 17 h and cooled.

After the synthesis was completed, the white $LiFePO_4$ particles were centrifuged three times with deionized water and dried. Carbon coating was conducted by mixing the $LiFePO_4$ with sucrose at a mass ratio of 5:1 ($LiFePO_4$:sucrose) without breaking the primary particles. This sample was heated to 600 °C for 5 h in a tube furnace under flowing Ar to yield the carbon-coated $LiFePO_4$. Surface carbon coating is used to increase the electronic conductivity of $LiFePO_4$ and has proven to be an effective strategy[53,54].

For chemical extraction of Li from carbon-coated $LiFePO_4$, an oxidizing solution was prepared by dissolving 1.36 g of nitronium tetrafluoroborate ($NO_2BF_4$) in 80 mL of acetonitrile. 0.8 g of carbon-coated $LiFePO_4$ powder was immersed into the solution and stirred for 24 h at room temperature (20-25 °C). The powder was then washed several times by acetonitrile and finally dried in a vacuum oven for 12 h. Finally, we will have micro-sized $FePO_4$ platelet particles (Supplementary Fig. 20).

Commercial $LiFePO_4$ particles (Comm-$LiFePO_4$) were bought from MTI Corporation (Item Number: Lib-LFPOS21; Supplementary Fig. 19). We also synthesized a smaller platelet-like $LiFePO_4$ particle by replacing 24 ml polyethylene glycol 400 with 24 ml ethylene glycol in the synthetic process, keeping all the other steps the same. The new formulation gives us smaller $LiFePO_4$ platelet-like particles shown in Supplementary Fig. 21, which are labeled as EG-$LiFePO_4$ particles. See Supplementary Fig. 25 and Supplementary Table 12 for the structure verification (Lebail refinements) of bought Comm-$FePO_4$/$LiFePO_4$ and EG-$FePO_4$/$LiFePO_4$ particles.

## Preparation of electrodes

All $FePO_4$ electrodes were prepared by casting a slurry of $FePO_4$, Super P carbon black (MTI Corporation; Item Number: Lib-SP; average particle size ~40 nm; purity ≥99.5%), and polyvinylidene fluoride (MTI Corporation; Item Number: Lib-PVDF; purity ≥99.5%) with a mass ratio of 80:10:10, in N-methyl-2-pyrrolidone. The electrode slurry was drop cast on a $0.5 \times 1$ cm$^2$ geometrical surface of a porous carbon cloth (ELAT-H, FuelCellEtc, 406 μm in thickness, 80% porosity) current collector of $5 \times 1$ cm$^2$ or a flat glassy carbon disk electrode (5 mm in diameter, 4 mm in thickness) and dried on a hotplate at 100 °C overnight. 3 nm $TiO_2$ was coated onto the $FePO_4$ electrodes to increase the wettability and decrease the contact impedance (See Supplementary Figs. 26, 27, 28, Supplementary Tables 13, 14, and Supplementary Note 5 for more discussions of surface carbon and $TiO_2$ coatings), using atomic layer deposition (ALD) at 100 °C, 0.645 Å/cycle with tetrakis(dimethylamido)titanium (IV) and $H_2O$ as precursors (Savannah G2 Thermal ALD). During tests, the other end of the carbon cloth was connected to a Pt clamp. The active material mass loadings ranged between 7 and 14 mg cm$^{-2}$. To measure the accessible capacity, the electrodes were cycled in 1 M LiCl aqueous solutions at 14.7 mA/g (Supplementary Fig. 29a) between −0.6 V and 0.6 V (vs. Ag|AgCl|KCl (4.0 M)), which delivered a 147 mAh/g capacity. See Supplementary Fig. 29 for detailed aqueous electrochemical energy storage performance evaluation. $FePO_4$ counter electrodes were made with the same

slurry depositing on carbon felt (Alfa Aesar) disks (0.9525 cm diameter × 3.18 mm thickness, around 240 g/m$^2$ in areal weight) by galvanostatically sodiating $FePO_4$ in 1 M $NaCl_{(aq)}$ at a C/20 (7.35 mA/g) rate until reaching a −0.6 V versus Ag/AgCl voltage cutoff. C/N describes the current to (de)intercalate the electrode in Nh. The active material mass loading on the counter electrodes ranged between 60 and 70 mg cm$^{-2}$. The larger mass loading of the counter electrode ensures we have enough ion stock in the counter electrode to avoid side reactions from water splitting and pH fluctuations.

We also evaluated the aqueous electrochemical energy storage performance of Comm-$FePO_4$ and EG-$FePO_4$ samples (Supplementary Figs. 30 and 31). Specifically, Comm-$FePO_4$ and EG-$FePO_4$ were cycled in 1 M LiCl aqueous solutions at 13.5 mA/g and 12.5 mA/g respectively, between −0.6 V and 0.6 V (vs. Ag|AgCl|KCl (4.0 M)), which delivered 135 mAh/g and 125 mAh/g capacity, respectively.

## Electrochemical methods

All electrochemical operations were performed on a Bio-Logic VMP3 workstation using a three-neck round-bottomed flask at room temperature (20-25 °C). $N_2$ (purity > 99.998%) was continuously bubbled into the solution to avoid side reactions caused from dissolved $O_2$[7]. Ag|AgCl|KCl (4.0 M) was used as the reference electrode.

*During seeding process*, $FePO_4$ working electrodes were paired with $LiFePO_4$ counter electrodes for galvanostatic Li-ion intercalation in 60 mL 1 M pure $LiCl_{(aq)}$ with different C rates (0.1 C, 2 C, 4 C, 6 C and 8 C; 1 C is equivalent to 147 mA/g specific current) and capacity range (10%, 20%, 30% and 40%; capacity range always relative to the measured capacity in 1 M LiCl aqueous solutions under 0.1 C, which is 147 mAh/g). For example, $L(0.2)_{4C}$ means seeding 20% of Li (29.4 mAh/g) into $FePO_4$ hosts under 4 C (588 mA/g). After the seeding process, electrodes were rinsed with 1 L of DI water with a flow rate of -0.3 L/min to remove adsorbed Li$^+$ and ready for the intercalation process. Specifically, L(0) means no seeding process.

*During co-intercalation process*, with or without seeding, all the working electrodes, paired with $NaFePO_4$ counter electrodes, would undergo intercalation in 500 mL of synthetic brine solutions (1 mM LiCl and 1 M NaCl mixed solution) until 70% of the total capacity using different intercalation C rates (0.1 C, 0.2 C, and 0.5 C; 0.1 C equals to 14.7 mA/g). For example, $L(0.2)_{4C}$-$LN(0.7)_{0.1C}$ means, after 4C-20% seeding process (588 mA/g−29.4 mAh/g), the intercalation was carried out under 0.1 C (14.7 mA/g) until 70% of capacity (102.9 mAh/g) was used.

*During recovery process*, after finishing the Li extraction in Na-dominated solutions, the electrode was first rinsed in three different 60 mL DI water for 30 min with continuous $N_2$ bubbling to remove excess adsorbed cations. The electrode was then de-intercalated in 30 mM $NH_4HCO_3$ solution with a constant current of C/30 (4.9 mA/g), using a graphite rod (Sigma-Aldrich, 99.995%, 10 cm length × 6 mm diameter) as the counter electrode and Ag|AgCl|KCl (4.0 M) as the reference electrode. The solution before and after the deintercalation process was collected for ICP-MS for Li and Na concentration measurement. We measure Li and Na concentration in the recovery solution and make sure the total ion amount measured matches the electrochemical deintercalation capacity with ~5% error tolerance.

See Supplementary Fig. 32 for the example electrochemical extraction cycle curves, including Li extraction and recovery steps. See Supplementary Fig. 33 for the detailed images of the cells used in the seeding, co-intercalation, and recovery processes. All the electrochemical operations were performed at room temperature (20-25 °C) with $N_2$ (purity > 99.998%) continuously bubbled into the solution from the inlet. Specifically, no climatic/environmental chamber is used.

*Electrochemical impedance spectroscopy (EIS)* was conducted in 60 mL 1 M LiCl aqueous solution under open air at room temperature (20-25 °C), with a graphite rod as the counter electrode and Ag|AgCl|

KCl (4.0 M) as the reference electrode. The applied signal was potentiostatic, with working potential set at open circuit voltage. The frequency ranged from 200 kHz to 100 mHz with 6 data points per decade of frequency at a 10 mV amplitude. Before each frequency, the measurement would wait for a 0.1 period.

## Indicators for Li extraction performance

Two different Li/(Li+Na) ratios are reported here. One is Li/(Li+Na)$_{total}$, which denotes the ratio of total Li$^+$ in the recovery solution, and another one is Li/(Li+Na)$_{net}$, which subtracts the contributions from the seeded-Li. For example, the tested Li/(Li+Na)$_{total}$ from ICP-MS results for L(0.2)$_{4C}$-LN(0.7)$_{0.1C}$ is 0.802. Therefore, the calculated Li/(Li+Na)$_{net}$ should be:

$$Li/(Li+Na)_{net} = \frac{Li/(Li+Na)_{total} \cdot Total\ used\ capacity - Seeded\ Li\ capacity}{Co - intercalated\ capacity}$$

$$= \frac{0.802 \times 0.7 - 0.2}{(0.7 - 0.2)} = 0.723$$

Another indicator is the Li selectivity, which is defined by the following equation:

$$Li_{selectivity} = \frac{([Li]/[Na])_{final}}{([Li]/[Na])_{initial}}$$

## XRD characterization

To prepare the seeded electrodes with SS maintained in the structure, we quickly disassembled the electrodes from the beaker cells in air, rinsed them with excess DI water to remove the adsorbed ions, dried the electrodes under vacuum for 20 min, and then sent for XRD measurements. The disassembly process was completed within 2 min of stopping the current. By rapidly disassembling the electrode and removing the electrolyte, we minimize inter-particle Li transport. XRD was carried out on Rigaku MiniFlex 600 diffractometer, using Cu Kα radiation (Kα 1: 1.54059 Å; Kα 2: 1.54441 Å; Kα 12 ratio: 0.4970). The tube voltage and the current used were 40 kV and 15 mA. Diffractograms were recorded with a 0.01° step width and a 5°/min speed. Rietveld refinement was executed on synthesized pristine LiFePO$_4$ and FePO$_4$ microplatelets using GSAS-II software (Supplementary Fig. 4).

## ICP-MS characterization

3% HNO$_{3(aq)}$ was used as the diluting matrix for all the Li recovery solutions. Besides, the chemically Li-extracted EG-FePO$_4$ hosts were first washed with distilled water 3–5 times, then digested with aqua regia solution for three days to ensure complete dissolution. The resulting supernatant was diluted with 3% HNO$_3$ for later ICP-MS measurement. All the measurements used either Thermo iCAP Q ICP-MS or Thermo iCAP RQ ICP-MS.

## SEM characterization

Scanning electron microscopy (SEM, Zeiss Merlin) was performed at the accelerating voltage of 10 kV.

## STEM-EDS characterization

STEM images were acquired using JEOL ARM 200F equipped with a cold field emission source operated at 200 kV. STEM EDS mapping was acquired using an Oxford X-Max 100TLE windowless SDD detector equipped with JEOL ARM 200F.

## SEND characterization

Scanning electron nanodiffraction patterns were acquired using a Themis Z S/TEM (Thermo Scientific, Waltham, USA). The microscope was operated in the μProbe STEM mode with an acceleration voltage of 300 kV. The electron probe focused on the sample had a semi-

convergence angle of 0.46 mrad, and a probe size of 1.8 nm in FWHM. For strain mapping, camera length was set at 360 mm so that in each diffraction pattern, the positions of about 40 diffraction peaks can be measured using circular Hough transform method to fit a 2D reciprocal lattice. Diffraction patterns were recorded using a CMOS camera (Ceta, Thermo Scientific) at the resolution of 1024 × 1024 pixels and 0.1 s exposure time per diffraction pattern. The scan was over an area of 600 × 400 nm$^2$ with a step size of 10 nm. The lattice parameters and measurement error are converted from diffraction peaks and uncertainty of peak detection, respectively, following our previous works[55,56].

## Reporting summary

Further information on research design is available in the Nature Research Reporting Summary linked to this article.

## Data availability

The authors declare that all relevant data are included in the paper and Supplementary Information files and are available from the corresponding author upon reasonable request.

## Code availability

The python code used to deconvolute XRD patterns is available from the corresponding author upon reasonable request.

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

## Acknowledgements

We thank Alexander S. Filatov (University of Chicago) for the set-up of Multi-purpose XRD. This work was supported by the Pritzker School of Molecular Engineering at the University of Chicago. This work was partially supported by the University of Chicago Materials Research Science and Engineering Center, which is funded by the National Science Foundation under award number DMR-2011854. This work made use of instruments in the Electron Microscopy Core, Research Resources Center in University of Illinois at Chicago.

## Author contributions

C.L. and G.Y. conceived the idea and designed the experiment. G.Y. performed all the experimental work and analyzed the experimental data. G.K. carried out and interpreted the DFT calculations. R.Y. and W.C. conducted SEND characterization. E.H. generated the code for Gaussian

fitting. F.S. conducted STEM imaging and elemental characterization with EDS. Y.H. conducted SEM imaging. Q.C. and J.Z. supervised the SEND characterization. W.C. supervised the DFT calculations. C.L. supervised the work. All authors analyzed the data and co-wrote the paper.

## Competing interests

The authors declare no competing interests.
