## [Peer Review File · Nature Communications]

REVIEWER COMMENTS

Reviewer #1 (Remarks to the Author):

The submitted manuscript to Nature Communications with the title "High-Li solid solution phases promoting Li co-intercalation competitiveness in FePO₄" tries to elucidate the mechanisms behind the Li/Na selectivity in the olivine LiFePO₄. This material has been extensively studied as Li selective electrode for lithium recovery methods, yet a deep structural study was missed, therefore, this article fills this gap, however, before its publication some modifications are required.

The reference list used by the authors is complete, no more citations are needed, however, in some parts of the article a comparison with results previously published is missed, e. g. the values of selectivity (equivalent to purity in some references) are in general lower than published by other authors (see ref. 9). This point does not decrease the quality of the study developed here, yet a comparison and potential explanation will reinforce the article's discussion.

The meaning of total capacity is not clear. I guess is the capacity delivered during the LiFePO₄ pre-oxidation step, different than the theoretical capacity.

A surprising and unexplained characteristic of the method followed is the election of just 70% of the total capacity as limit during the Li recovery. Why this value? This means that 30% of the available Li sites are not used, then the capacity of the material to transfer Li from diluted solution is more limited. In theory, a high % will increase the fraction of high-Li SS phases, increasing the selectivity.

A discussion of the pros and cons of using 40% prelithiated FePO₄ should be added. The benefits in terms of selectivity are clear, however, just 30% of the material is used to transfer Lithium from the seed solution to the recovery one, this will require the cycle repetition to obtain the same amount of lithium transferred.

The low-Li SS phases remain almost constant during the whole seeding range (Fig. 2d). Why does this fraction remain constant? And how can affect the value of Li selectivity? Why the error bar in seeding 0.2 Fig. 2d is much larger than in the rest of the seeding values?

Page 8, lines 201-204. Why a seeding equal to 0.2 was selected for the C effect study? A seeding equal to 0.4 seems more adequate since the selectivity obtained with this value was higher.

How was measured the figure 3f? was there a seeding step between cycles? Was the electrolyte recovered and measured each cycle? If the answer to these questions is affirmative. I think a new experiment without seeding step for the second cycle is required to elucidate if the improvement of selectivity works just for one cycle or it remains upon cycling.

In the preparation of electrode section (page 13 lines 346-349), there is described the coating of the particle with TiO₂, however, there is no mention in the manuscript about this coating.

As suggested by nature communications, I sign this review:

Dr. Rafael Trócoli

Institut de Ciència de Materials de Barcelona (CSIC)

Campus UAB E-08193 Bellaterra

Catalonia (SPAIN)

Reviewer #2 (Remarks to the Author):

The article presents innovative experiments and interesting results, which may have potential for translation into practical application for selective extraction of lithium ions from mixed brines. It is very well written overall. A few typos or minor corrections may be needed (please see annotated copy).

A few clarifications are requested as follows:

1- The overall performance of the materials as battery materials, in terms of absolute capacity as function of C-rate would be very informative. The particles have been coated with TiO₂ by ALD and with carbon from sugar pyrolysis, both of which will impact the overall electrochemical performance. That information will provide some benchmark for the "quality" of the materials in terms of battery material performance.

2- If such information, mentioned in 1 above, has been gathered for Na-intercalation in Na-ion battery types, it would be useful too.

3- There are several normalisations, one from the XRD peak intensities, another from combined total Li+Na content, and perhaps another from the intercalation "capacity" (in synthetic brine solution). This intercalation capacity is normalised at 0.7, as shown in Figure S8, but remains slightly ambiguous, without a clear display of the full absolute capacities at different C-rates. The display of absolute capacities would add clarity to the information.

4- Is the capacity 70% of the measured capacity at 0.1C? Is it 70% of the measured capacity at the selected, fixed C/n rate? I am not sure how different the rate capability in synthetic brine solution is to be expected compared to the C-rate in an actual battery; nanoscale LFP at 5C in a Li-ion battery typically displays less than 70% of the capacity at 0.1C, therefore, it seems important to show clearly and unambiguously the rate capability in the brine.

5- Some brief contrast of rate capability in standard battery cell format and that for synthetic brine would be of interest. Are the best performing battery materials also the best for Li-extraction?

6- A few comments have been made on the annotated copy of the word format file. Some comments refer to the ranges of x values in brackets, which I found confusing or may have a typo.

I recommend publication after addressing the abovementioned clarifications.

Reviewer #3 (Remarks to the Author):

The authors present interesting work that is aimed to shed more light on the role of pre-formation of solid-solution in olivine FePO₄ electrode host material that is applied in the emerging approach of selective electrochemical (Lithium) ion extraction from geothermal sources.

It is exciting to see how comprehensive knowledge gained in one field of electrochemistry (battery energy storage systems) is utilized in another one (element extraction from natural sources).

The authors indeed conducted a systematic study and provide quite extensive results that do rather well support their proposed mechanism of the promoted enhanced ion insertion selectivity in the host FePO₄ (FP) material.

Nonetheless, there are still some missing parts in the puzzle.

A) One important aspect missing is the effect of FP particle size (lateral and thickness dimensions). The authors show SEM image of the "parent" LiFePO_4 (LFP) particles (Figure S13) where it can be seen that the micrometer-sized particles indeed have platelet-like morphology. It can be deduced from the image that the particles somewhat vary in the thickness with the thick ones showing thicknesses of about 400 nm or probably even more in some cases. The important question is how do the particle dimensions (especially the platelet thickness governing the [010] 1D channel length) impact: i) solid solution formation during the high-rate Lithium "seeding" step, and ii) the consequential ion insertion Li/Na selectivity enhancement. Experimental investigation of this aspect should be provided.

B) We know that Li-ion battery-grade LFP materials have to be made of LFP crystallites of sub-micron dimensions in order to provide needed sufficient rate performance. At the same time it was found (e.g. ref. R1 and R2) that in these types of materials the total amount of observed solid-solution is rather small even at very high rates of lithiation and the observed relaxation time of the formed non-equilibrium solid-solution was found to be short (in the order 10 seconds or several minutes). On the other hand, more lately there were reports (e.g. ref. R3, ref. 44 in this manuscript) showing results with partially lithiated platelet LFP ($\text{Li}_{0.5}\text{FePO}_4$) that exhibited very large fractions of solid-solution (even larger than 50%). It appears that in the current work the authors have synthesized very similar platelet LFP particles and moreover the solid-solution fractions obtained during high-rate "seeding" step are comparable to those found in the work of Li et al. (ref. R3, ref. 44 in this manuscript). So it seems that this type of large/thick LFP particles exhibit high solid-solution fractions after partial high-rate lithiation in both organic and water-based electrolytes. Time evolution of a diffusion process is strongly dependent upon the medium dimension (L) wherein it is let to diffuse and in general the characteristic propagation time is proportional to $L \times L = L^2$. Thus surely the formation and relaxation of the non-equilibrium Li solid solution phase(s) has to be strongly dependent upon the particle dimensions. The authors are encouraged to elaborate this point. Preferable would be to extend the experimental study to at least one (notably) different particle size/thickness. For example they could employ some battery-grade LFP material with good high C-rate performance.

C) How does FP crystal size/thickness affect the possible practical rate of the process of Li extraction from Na-Li brine mixtures?

D) Based on the answering the question of the particle size/thickness effect in upper points A), B) and C) the authors should be able to provide guidelines for the most suitable FP host material particle size (thickness) that would enable most selective and most efficient Li extraction from mixed Na-Li salt solutions.

E) How does the total amount of solid-solution fraction depend upon rate of lithiation during the Li pre-seeding step? Figure S7 shows relative amounts of the selected solid-solution phase fractions but it does not show the total (absolute) amount of the sum of all the solid-solution compositions. It is not clear for example what is the sum of all solid-solution compositions for C/10 pre-lithiation. At this (low) rate of lithiation the expected solid-solution formation is expected to be much lower compared to high C-rates. At very low rates of partial lithiation the solid-solution formation should tend to very small values since the active particle population should tend to zero when current density (C-rate) goes to zero. At least this is the current understanding of the topic for particle population lithiation in Li ion battery systems with organic electrolytes.

F) It is not clearly described and explained by the authors what are the reasons for the

statement that high-rate Li pre-seeded FP particles would preferentially insert Na at the edges of a particles (Scheme in Figure 2a and Scheme S1). Should we not expect that the corresponding pre-seeded Li solid-solution be formed somehow randomly within a particle? Please do explain this your hypothesis more in a detail. Do you have any experimental data (EDS elemental mapping, local electron diffraction, SEND) to support it?

G) Based on the DFT results that you show in Figure 4d what should one expect in the voltage curves V vs Ag/AgCl RE? For example, let say that we pre-lithiate FP up to $\text{Li}_{0.5}\text{FePO}_4$ and afterwards: i) continue lithiation with low rate (e.g. $-C/100$), or ii) wash the electrode and transfer it in Na-system (pure Na-electrolyte) and perform intercalation of Na at low rate. Should we observe a distinct voltage step to lower voltage when switching from Li to Na intercalation? Do you have any experimental confirmation in regard of this? Please provide some additional explanation of the DFT results in Figure 4d – e.g. can be the difference in ΔG values directly translated in the difference in the corresponding voltage curves?

H) How it would behave FP host material with very small particles – let say nano-LFP that was shown to have intrinsic solid-solution (de)-lithiation mechanism in the whole Lithium compositional span? Would be nano-LFP the most- or the least-suitable material for selective Li extraction from Li-Na solution mixtures?

References:

R1: Liu, H. et al., Capturing metastable structures during high-rate cycling of LiFePO_4 nanoparticle electrodes. *Science* 344 (6191), 1451–1452 (2014).

R2: Zhang, X. et al., Rate-induced solubility and suppression of the first-order phase transition in olivine LiFePO_4 . *Nano Letters* 14, 2279–2285 (2014).

R3: Li, Y. et al., Fluid-enhanced surface diffusion controls intraparticle phase transformations. *Nature Materials* (2018).

Response to reviewers' comments:

We would like to thank the reviewers for their efforts in reviewing our manuscript. We have revised our manuscript accordingly. Below please find our point-by-point response to the comments received. The major changes in our revised manuscript have been marked in Red.

Reviewer #1:

The submitted manuscript to Nature Communications with the title “High-Li solid solution phases promoting Li co-intercalation competitiveness in FePO₄” tries to elucidate the mechanisms behind the Li/Na selectivity in the olivine LiFePO₄. This material has been extensively studied as Li selective electrode for lithium recovery methods, yet a deep structural study was missed, therefore, this article fills this gap, however, before its publication some modifications are required.

1. The reference list used by the authors is complete, no more citations are needed, however, in some parts of the article a comparison with results previously published is missed, e. g. the values of selectivity (equivalent to purity in some references) are in general lower than published by other authors (see ref. 9). This point does not decrease the quality of the study developed here, yet a comparison and potential explanation will reinforce the article's discussion.

Thanks for the valuable comment. We agree with the reviewer that a summary of the performances from the literature and an explanation of selectivity dependence is helpful to the readers. We compared our results with those previously published in **Table R1 also in Supplementary Table 6**. As can be seen from the table, there are three main differences for the experimental conditions: 1. the concentration and molar ratio of Li ; 2. the current density; and 3. the used capacity. In our study, we have showed that the current density and used capacity will affect the selectivity (**Figure 3** in the manuscript). In the table, we marked all these conditions to help clarify the differences in selectivity. Direct comparison is difficult. But in general, the Li selectivity will decrease with increasing current density and increasing capacity used. The LiFePO₄ particle size and morphology have complex effect to the selectivity, and we discussed some aspects of the particle morphology effect in page 12-13 in our revised manuscript. Additionally, electrochemical methods used for intercalation also affect the selectivity (Joule 2020, 4, 1459; Ref 5 below).

For our control samples without the seeding process, our used capacities are among the largest. Among our samples, for the Li seeded ones, the selectivity is high (e.g., L(0.4)_{8C}-LN(0.7)_{0.1C}). We only identified one reference with similar experimental conditions to our control experiment. In Ref1, constant current 22.2 mA/g, capacity used 44.4 mAh/g, initial Li:Na 5 mM: 5 M (Li to Na of 1:1000) in Table R1, the reported selectivity is 5.6×10^2 . Our control using smaller constant current of 14.7 mA/g with larger capacity of 58.8 mAh/g showed selectivity of 2.05×10^3 .

We added **Table R1** as **Supplementary Table 6** in the SI.

Table R1 Li selectivity over Na with FePO₄ in the literature and this work.

Reference	Constant current during extraction (mA/g)	Capacity used during extraction (mAh/g)	Initial Li:Na (at%)	Recovered Li:Na (at%)	Selectivity	
LiFePO ₄ ¹	22.2	44.4	50 mM:5 M (1:100)	5.7:1	5.70 × 10 ²	
		44.4	5 mM:5 M (1:1000)	1:1.8	5.60 × 10 ²	
		44.4	0.5 mM:5 M (1:10000)	1:4.0	2.50 × 10 ²	
LiFePO ₄ ²	62.5	62.5	0.1 M:0.1 M (1:1)	225.0:1	2.25 × 10 ²	
		62.5	1 M:1 M (1:1)	17250.0:1	1.73 × 10 ⁴	
		62.5	42 mM:793 mM (1:19; Atacama)	9250.0:1	1.76 × 10 ⁵	
		62.5		106.5:1	2.02 × 10 ³	
62.5	3.1:1	59				
LiFePO ₄ ³	0.184	0.66 (~215 mins operation)	5 mM:0.5 M (1:100)	2.6:1 (dissolved solid)	2.60 × 10 ²	
LiFePO ₄ w/ Polydopamine coating ³		0.66 (~215 mins operation)		43.3:1 (dissolved solid)	4.33 × 10 ³	
LiFePO ₄ ⁴	Chemical reaction, using potassium persulfate as the oxidant	*45.3 mg Li/g FePO ₄	60 mM:6 M (1:100)	39.2:1 (dissolved solid)	3.92 × 10 ³	
		*44.8 mg Li/g FePO ₄	60 mM:3 M (1:50)	49.1:1 (dissolved solid)	2.46 × 10 ³	
		*35.1 mg Li/g FePO ₄	60 mM:0.6 M (1:10)	38.4:1 (dissolved solid)	3.84 × 10 ²	
		*45.7 mg Li/g FePO ₄	60 mM:4.6 M (1:77)	61.0:1 (dissolved solid)	4.70 × 10 ³	
		*46.4 mg Li/g FePO ₄	200 mM:3 M (1:15)	370.0:1 (dissolved solid)	5.55 × 10 ³	
LiFePO ₄ w/ TiO ₂ coating ⁵	34 (constant current)	51	0.025 mM:0.47 M (1:18500, Seawater)	1:2.18	8.49 × 10 ³	
	17 (P _{10s} -equivalent current density)	51	0.025 mM:0.47 M (1:18500, Seawater)	1:1.01	1.83 × 10 ⁴	
	17 (P _{1s} -equivalent current density)	51	0.025 mM:0.47 M (1:18500, Seawater)	1:1.11	1.67 × 10 ⁴	
	10.625 (P _{10s} R _{2s} -equivalent current density)	53.125	0.025 mM:0.47 M (1:18500, Seawater)	1.01:1	1.87 × 10 ⁴	
	17 (P _{10s} -equivalent current density)	51	0.025 mM:0.47 M (1:18500, Li-Na binary)	1:1	1.85 × 10 ⁴	
	17 (P _{10s} -equivalent current density)	51	0.235 mM:0.47 M (1:2000, Li-Na binary)	16.5:1	3.30 × 10 ⁴	
	17 (P _{10s} -equivalent current density)	51	1.88 mM:0.47 M (1:250, Li-Na binary)	1:0 (Na below detection limit)	n.a.	
This work	Constant current during extraction (mA/g)	Capacity used during extraction (mAh/g)	Total capacity used (mAh/g)	Initial Li:Na (at%)	Recovered Li:Na (at%)	Selectivity
	(147 mA/g = 1 C)					
L(0)-LN(0.1)0.1C	14.7	14.7	14.7	1 mM:1 M (1:1000)	11.0:1	1.10 × 10 ⁴
L(0)-LN(0.2)0.1C	14.7	29.4	29.4		5.4:1	5.41 × 10 ³
L(0)-LN(0.3)0.1C	14.7	44.1	44.1		3.7:1	3.66 × 10 ³
L(0)-LN(0.4)0.1C	14.7	58.8	58.8		2.0:1	2.05 × 10 ³
L(0)-LN(0.7)0.01C	1.47	102.9	102.9		13.6:1	1.36 × 10 ⁴
L(0)-LN(0.7)0.1C	14.7	102.9			1.7:1	1.68 × 10 ³
L(0)-LN(0.7)0.2C	29.4	102.9			1.1:1	1.12 × 10 ³
L(0)-LN(0.7)0.5C	73.5	102.9			0.4:1	4.15 × 10 ²
L(0.1)4C-LN(0.7)0.1C	14.7	88.2			1.6:1	1.58 × 10 ³
L(0.2)4C-LN(0.7)0.1C	14.7	73.5			2.6:1	2.61 × 10 ³
L(0.3)4C-LN(0.7)0.1C	14.7	58.8			3.5:1	3.50 × 10 ³
L(0.4)4C-LN(0.7)0.1C	14.7	44.1			6.0:1	6.00 × 10 ³
L(0.2)4C-LN(0.7)0.2C	29.4	73.5			1.6:1	1.62 × 10 ³
L(0.2)4C-LN(0.7)0.5C	73.5	73.5			0.6:1	6.50 × 10 ²
L(0.2)0.1C-LN(0.7)0.1C	14.7	73.5			1.9:1	1.94 × 10 ³
L(0.2)2C-LN(0.7)0.1C	14.7	73.5			2.6:1	2.64 × 10 ³
L(0.2)6C-LN(0.7)0.1C	14.7	73.5			2.7:1	2.72 × 10 ³
L(0.2)8C-LN(0.7)0.1C	14.7	73.5			3.3:1	3.26 × 10 ³
L(0.4)0.1C-LN(0.7)0.1C	14.7	44.1			2.7:1	2.74 × 10 ³
L(0.4)2C-LN(0.7)0.1C	14.7	44.1			4.4:1	4.38 × 10 ³
L(0.4)6C-LN(0.7)0.1C	14.7	44.1			9.0:1	9.05 × 10 ³
L(0.4)8C-LN(0.7)0.1C	14.7	44.1			14.8:1	1.48 × 10 ⁴

(*: 46.4 mg Li/g FePO₄ is equivalent to 170 mAh/g capacity used)

References:

1. M. Pasta, A. Battistel, F. La Mantia, Energy Environ. Sci. 2012, 5, 9487.
2. R. Trocoli, A. Battistel, F. La Mantia, Chemistry-a European Journal 2014, 20, 9888.
3. J.-S. Kim, Y.-H. Lee, S. Choi, J. Shin, D. Cuong, J. Choi, Environmental science & technology 2015, 49.
4. N. Intaranont, N. Garcia-Araez, A. L. Hector, J. A. Milton, J. R. Owen, Journal of Materials Chemistry A 2014, 2, 6374.
5. C. Liu, Y. B. Li, D. C. Lin, P. C. Hsu, B. F. Liu, G. B. Yan, T. Wu, Y. Cui, S. Chu, Joule 2020, 4, 1459.

2. The meaning of total capacity is not clear. I guess is the capacity delivered during the LiFePO_4 pre-oxidation step, different than the theoretical capacity.

We thank the reviewer for the comment. We made clarifications to the capacities mentioned in the manuscript. The total capacity we refer to is 147 mAh/g, measured by cycling the electrodes in 1 M LiCl aqueous solutions at 14.7 mA/g (equivalent to 0.1C defined in our case) between -0.6 V and 0.6 V (vs. Ag/AgCl) (**Figure R1**). The capacity delivered at 73.5 mA/g (0.5C) and 588 mA/g (4C) in 1M LiCl are 136 mAh/g and 101 mAh/g. We added the capacity measurement in the SI and included **Figure R1** into **Supplementary Figure 23**.

Figure R1 Electrochemical cycling of the FePO_4 electrodes in 1 M LiCl aqueous solution under different current densities/C rates. 14.7 mA/g equals a rate of 0.1C, while 147 mA/g equals 1C.

3. A surprising and unexplained characteristic of the method followed is the election of just 70% of the total capacity as limit during the Li recovery. Why this value? This means that 30% of the available Li sites are not used, then the capacity of the material to transfer Li from diluted solution is more limited. In theory, a

high % will increase the fraction of high-Li SS phases, increasing the selectivity.

Thanks for the great question. The total capacity we used here is always based on the capacity measured in 1 M LiCl aqueous solutions at 14.7 mA/g (**Figure R1**), which is 147 mAh/g. However, as shown in **Figure R2a**, we could only get 134 mAh/g capacity within the selected water safety window in 1 mM:1 M LiCl:NaCl mixed solution, which is 91% of the 147 mAh/g capacity. The highest co-intercalation rate we used throughout the paper is 0.5C (**Figure 3d** in the manuscript), which further decreased the accessible capacity. Therefore, as shown in **Figure R2b**, we set a limit of 70% of capacity (102.9 mAh/g) to prevent side reactions from damaging the structure under high polarization voltage and use a slower C rate (4.9 mA/g) to fully recover the intercalated ions. Moreover, the used 70% of capacity (102.9 mAh/g) is already larger than most of the reported capacity range, as shown in **Table R1**.

Indeed, a large seeding range will increase the fraction of high-Li SS phases, thus increasing the selectivity. For example, the performance in the 40% seeding case is better than that in the 20% seeding case (**Figure 3b** in the manuscript). However, a larger intercalation range in the Li extraction step will decrease the Li selectivity due to the increased overpotential at the enlarged depth of discharge, which promotes the intercalation of Na⁺ (**Figure 3c** in the manuscript).

We integrated **Figure R2a** into **Supplementary Figure 23** and included **Figure R2b** as **Supplementary Figure 25** in the SI.

Figure R2 (a) Electrochemical cycling of the electrodes in 1 M LiCl and 1 mM LiCl: 1 M NaCl aqueous solutions under 0.1C (14.7 mA/g). (b) 0.1C (14.7 mA/g) intercalation curve in 1 mM LiCl: 1 M NaCl aqueous solution and C/30 (4.9 mA/g) de-intercalation curve in 30 mM NH₄HCO₃ recovery solution, with the use of 70% of the total capacity (102.9 mAh/g).

4. A discussion of the pros and cons of using 40% prelithiated FePO₄ should be added. The benefits in terms of selectivity are clear, however, just 30% of the material is used to transfer Lithium from the seed solution

to the recovery one, this will require the cycle repetition to obtain the same amount of lithium transferred.

Thanks for the reviewer's suggestion. We do have the pros and cons of using 40% prelithiated FePO₄ hosts. For the benefit, with 40% seeding under 4C, the achieved Li/(Li+Na)_{net} is 0.86 ± 0.01, which is higher than the 20% seeding case (0.72 ± 0.03).

For the drawbacks, as brought out by the reviewer, we can only have 30% of the capacity to transfer the lithium from the seed solution, which requires more cycle repetition to obtain the same amount of lithium transferred. For example, comparing 20% seeding and 40% seeding cases, to obtain the same amount of transferred lithium, we need 1.4 times the extraction cycle number of the 20% seeding.

The value of investigating different seeding range in this paper is to draw the correlation between Li selectivity to solid solution phase fraction of Li-seed phases. We think this finding will guide us to design better electrodes to achieve higher high-Li SS phases at smaller seeding capacity that we can use much less capacity during seeding.

We added the discussion of this point in the paper as

“Specifically, the Li/(Li+Na)_{net} ratio increased from 0.61 ± 0.01 to 0.86 ± 0.01 from 10% to 40% seeding. With 40% seeding, we achieved ~ 3.8 fold increase of Li selectivity to 6.0 × 10³, comparing to the empty host without seeding. However, we inevitably need more cycle repetition to obtain the same amount of lithium transferred with a Li pre-seeded host. Further improvement of the structural response of the FePO₄ electrode could improve the capacity usage”

5. The low-Li SS phases remain almost constant during the whole seeding range (Fig. 2d). Why does this fraction remain constant? And how can affect the value of Li selectivity?

We thank the reviewer for the great question. At this stage, we think the correlation between the low-Li SS phases to the Li selectivity is not obvious. As shown in **Figure 3e**, the coefficient of determination (COD) is only 0.01. However, we do not fully understand why the low-Li SS phase fraction did not change much for all the seeding ranges. We do see that even the total phase fractions of low-Li SS phases remain constant at around 13%, the compositions for each of three components (Li_{0.125}FP, Li_{0.250}FP, and Li_{0.375}FP) are changing under different seeding ranges (**Figure 2d**). The formation of SS phases is a complicated process that involves intercalation, solid diffusion and cross channel surface rearrangement. We think this process is related to particle size, morphology, surface chemistry and solution composition. An in situ XRD monitoring the phase evolution can be helpful and we will investigate this in our future work.

6. Why the error bar in seeding 0.2 Fig. 2d is much larger than in the rest of the seeding values?

We thank the reviewer for pointing this out. We repeated more times the 20% seeding condition and have a better statistic (**Figure R3**). We updated the **Figure 2d and Supplementary Figure 6** in the manuscript.

Figure R3 Averaged accumulative SS phase fractions of L(0.1/0.2/0.3/0.4)_{4C}. (Error bars representing the standard deviation of three replicate measurements.)

7. Page 8, lines 201-204. Why a seeding equal to 0.2 was selected for the C effect study? A seeding equal to 0.4 seems more adequate since the selectivity obtained with this value was higher.

Thanks for the reviewer's suggestion. We were thinking about using a smaller seeding range therefore a larger capacity for the further Li extraction, so we picked 20% seeding for the C rate effect study. We agree with the reviewer that a higher seeding range could be more interesting for studying the effects of C rates. Therefore, we carried out the C effect study with 40% seeding, as shown in **Figure R4**. We did see a clearer trend than the 20% seeding case that as the seeding C rate increases, the Li selectivity increases. It is also worth noting that the L(0.4)_{8C}-LN(0.7)_{0.1C} case achieved the highest selectivity (1.48×10^4) in this work (**Table R1**).

We included **Figure R4** into **Supplementary Figure 7** and added the following text into the revised manuscript

Figure R4 $Li/(Li+Na)_{total}$ and $Li/(Li+Na)_{net}$ under different seeding C rates with the same 40% seeding range $L(0.4)_{0.1/2/4/6/8C}$.

“We also tested the electrodes with 40% of seeding under different seeding C rates. As shown in Supplementary Figure 7c, comparing to 20% seeding, we see a clearer trend that increasing the seeding C rate would lead to better Li selectivity. For the $L(0.4)_{8C}-LN(0.7)_{0.1C}$ case, we achieved the highest Li selectivity ($Li_{selectivity} = 1.48 \times 10^4$) in this work (Supplementary Table 6). The applied high seeding current density induces concurrent, non-mosaic intercalation in the porous electrode⁴⁸.”

8. How was measured the figure 3f? was there a seeding step between cycles? Was the electrolyte recovered and measured each cycle? If the answer to these questions is affirmative. I think a new experiment without seeding step for the second cycle is required to elucidate if the improvement of selectivity works just for one cycle or it remains upon cycling.

For **Figure 3f**, each cycle has Li seeding, Li extraction, and Li recovery steps. We did measure the Li selectivity for each cycle. To address the question, we conducted the following two cycling tests. As shown in **Figure R5**, we skip the Li seeding step from the 2nd cycle and use either 70% or 50% of capacity for further Li extraction (70% represents the same total capacity and 50% represents the same Li extraction capacity). Obviously, the recovered $Li/(Li+Na)$ decreases for the following three cycles, which indicates that the seeding effect only works for one cycle. However, it would be very interesting if we could maintain the selectivity upon cycling by only seeding once. With the total $Li/(Li+Na)$ dropping from 0.81 ± 0.01 to 0.56 ± 0.02 for 70% case (**Figure R5a**) after four extraction cycles, we observed slight Li selectivity decrease. This is worth further investigation to check the effect of the Na co-intercalation fraction on the stability of the $FePO_4$ host.

Figure R5 $\text{Li}/(\text{Li}+\text{Na})_{\text{net}}$ for the multi-intercalation stability test using 20% Li seeding for the 1st cycle and skipping Li seeding for the following cycles. (a) 70% capacity usage from the 2nd cycle, $\text{L}(0)\text{-LN}(0.7)_{0.1\text{C}}$. (b) 50% capacity usage from the 2nd cycle, $\text{L}(0)\text{-LN}(0.5)_{0.1\text{C}}$.

We included **Figure R5** as **Supplementary Figure 10** and added the following text into the revised manuscript:

“Meanwhile, the $\text{L}(0.2)_{4\text{C}}$ seeding condition was tested for multiple cycles on one electrode. Each cycle has Li seeding, Li extraction, and Li recovery steps. We measured the Li selectivity for each cycle. As shown in Figure 3f, the $\text{Li}/(\text{Li}+\text{Na})_{\text{net}}$ is maintained at ~ 0.73 , proving the effect of seeding to improve Li selectivity as well as cycle stability. Additionally, we conducted two more cycling tests. Shown in Supplementary Figure 10, we skip the Li seeding step from the 2nd cycle and use either 70% or 50% of capacity for further Li extraction. A decrease of the recovered $\text{Li}/(\text{Li}+\text{Na})$ was observed without seeding from the 2nd cycle, indicating that the seeding effect can only work for one cycle.”

9. In the preparation of electrode section (page 13 lines 346-349), there is described the coating of the particle with TiO_2 , however, there is no mention in the manuscript about this coating.

We thank the reviewer for pointing this out. We followed our previous work for the coating of 3nm TiO_2 ¹. We would like to clarify that the TiO_2 was coated after the electrode fabrication, not on each particle. We only do carbon coating for each particle. The TiO_2 layer is the outer surface of the electrode interfacing with the electrolyte. In this case, the electron conduction is still from the carbon current collector, to activated carbon, and to carbon-coated FePO_4 . TiO_2 is not conducting electrons but conducting Li^+ . The main role of TiO_2 is to improve the solid-liquid interface contact. First, the coating of TiO_2 does not affect the battery material performance. As can be seen from **Figure R6**, the TiO_2 -Carbon-coated FePO_4 electrode delivered a similar specific capacity (147 mAh/g) with only Carbon-coated FePO_4 electrode (151 mAh/g), while the Bare- FePO_4 electrode without either coating delivered a much worse specific capacity (120 mAh/g; Note: bare particles were annealed under the same condition without mixing with sucrose). Surface

carbon coating is often used to increase the electrical conductivity of LiFePO_4 and has proven to be an effective strategy^{2,3}.

Figure R6 Electrochemical cycling of Bare, Carbon-coated and TiO_2 -Carbon-coated FePO_4 electrodes in 1 M LiCl aqueous solution under 0.1C (14.7 mA/g). 147 mA/g equals a rate of 1C.

For the role of TiO_2 coating, as shown in **Figure R7a**, without TiO_2 coating, the carbon-coated FePO_4 together with carbon cloth substrate is very hydrophobic. After coating the electrode with TiO_2 , the 10 μL water immediately infiltrated the electrode surface. Several pre-wetting steps can be avoided using TiO_2 -Carbon-coated FePO_4 electrodes. We are aware that surface coating may affect the Li^+ diffusivity. We found in the literature⁴ that beta- TiO_2 has similar Li^+ diffusivity compared to LiFePO_4 . Our ALD coated TiO_2 is amorphous. It was shown in other work that amorphous TiO_2 has Li^+ diffusivity on the same order of magnitude as crystalline TiO_2 ⁵. Therefore, we find TiO_2 to be a good candidate for coating compared to other common ALD materials such as ZrO_2 , Al_2O_3 , etc., which showed much higher Li diffusion barriers. We also conducted EIS in 1M LiCl aqueous solution. **Figure R7b** shows the Nyquist plots for the carbon-coated FePO_4 wo/w the TiO_2 coating with the equivalent circuit shown in the inset. The resistor R_s corresponds to the electrolyte resistance. The resistors R_1 and R_2 paralleled with the constant phase element (CPE) account for the contact impedance and charge transfer impedance, respectively. The ion diffusion in the host material is described with the Warburg element (Z_w). As shown in **Figure R7b**, the simulated data from the equivalent circuit well fit the impedance data for both electrodes. The values for the different resistances obtained from fitting are listed in **Table R2**. As shown in **Table R2**, the electrolyte resistance (R_s) and charge transfer impedance (R_2) are almost the same for the carbon-coated FePO_4 wo/w the TiO_2 coating. The contact impedance (R_1) of the electrode without TiO_2 coating is more than double the value of the electrode with TiO_2 coating. Therefore the TiO_2 coating can reduce the contact resistance as well.

Figure R7 (a) Photographic image showing 10 μL water dropped on the surface of the carbon-coated FePO_4 electrode wo/w 3nm TiO_2 coating. (b) Nyquist plots for the electrode wo/w 3nm TiO_2 obtained by electrochemical impedance spectroscopy (EIS) tests in 1 M LiCl aqueous solution with the frequency ranging from 200 kHz to 100 mHz at a 10 mV amplitude. The dot-dashed lines are the fitting curves by using the equivalent circuit, which is shown as the inset and consists of a resistor (R_s), a resistor (R_1) paralleled with a constant phase element (CPE), and a CPE parallel with a resistor (R_2) which is connected with a Warburg element (Z_w) in series.

Table R2 Electrode resistances for the carbon-coated FePO_4 electrode wo/w 3nm TiO_2 coating obtained from equivalent circuit fitting of EIS results.

	wo TiO ₂	w TiO ₂
R _s	5.2	5.5
R ₁	21.1	9.7
R ₂	19.3	22.6

We also evaluated the effect of TiO₂ coating on the seeding process. **Figure R8** shows the XRD patterns of the carbon-coated FePO₄ electrode wo/w TiO₂ coating after 20% Li seeding under 4C. There is little difference for both obtained and fitted patterns. The calculated solid solution fraction (SSF) for the electrode without or with TiO₂ coating are 0.429 and 0.428, respectively, which is similar. And the recovered Li/(Li+Na) ratios for the following Li extraction step are almost the same (**Table R3**). Without TiO₂, Li/(Li+Na)_{net} is 0.73 ± 0.01, and with TiO₂, Li/(Li+Na)_{net} is 0.74 ± 0.01. Therefore, the above results show that the TiO₂ coating will not affect SSF generated in the seeding process, as well as the following Li extraction performance. Long term corrosion resilience can be a benefit of TiO₂ coating. We will investigate this in our future study.

Figure R8 XRD of the carbon-coated FePO₄ electrode wo/w 3nm TiO₂ coating after 20% Li seeding under 4C (Dotted lines: obtained patterns; Solid lines: fitted patterns).

Table R3 Recovered Li/(Li+Na)_{net} ratios of the carbon-coated FePO₄ electrode wo/w 3nm TiO₂ coating using 20% Li seeding (Error bars representing the standard deviation of three replicate measurements).

	L(0.2) _{4C} -LN(0.7) _{0.1C}	
	wo TiO ₂	w TiO ₂
Li/(Li+Na) _{net}	0.73 ± 0.01	0.74 ± 0.01

We have added the discussions of surface carbon and TiO₂ coatings in our revised Supplementary Materials accordingly.

References:

1. C. Liu, Y. B. Li, D. C. Lin, P. C. Hsu, B. F. Liu, G. B. Yan, T. Wu, Y. Cui, S. Chu, *Joule* 2020, 4, 1459.
 2. M. Herstedt, M. r. Stjerndahl, A. Nyttén, T. r. Gustafsson, H. k. Rensmo, H. Siegbahn, N. Ravet, M. Armand, J. O. Thomas, K. Edström, *Electrochemical and Solid-State Letters* 2003, 6, A202.
 3. J. Wang, X. Sun, *Energy Environ. Sci.* 2012, 5, 5163.
 4. S. Xu, R. M. Jacobs, H. M. Nguyen, S. Hao, M. Mahanthappa, C. Wolverton, D. Morgan, *Journal of Materials Chemistry A* 2015, 3, 17248.
 5. Y.-M. Lin, P. R. Abel, D. W. Flaherty, J. Wu, K. J. Stevenson, A. Heller, C. B. Mullins, *The Journal of Physical Chemistry C* 2011, 115, 2585.
-

Reviewer #2:

The article presents innovative experiments and interesting results, which may have potential for translation into practical application for selective extraction of lithium ions from mixed brines. It is very well written overall. A few typos or minor corrections may be needed (please see annotated copy).

We appreciate the positive comments and valuable suggestions from the reviewer. Below we provided point-to-point replies to the reviewer's questions.

A few clarifications are requested as follows:

1- The overall performance of the materials as battery materials, in terms of absolute capacity as function of C-rate would be very informative. The particles have been coated with TiO₂ by ALD and with carbon from sugar pyrolysis, both of which will impact the overall electrochemical performance. That information will provide some benchmark for the “quality” of the materials in terms of battery material performance.

We thank the reviewer for bringing this point out and we totally agree with the reviewer that battery material performance analysis is needed as a reference point for future comparisons between electrodes.

First, we would like to clarify that the 3 nm TiO₂ was coated after the electrode fabrication, not on each particle. Only the carbon was coated for each particle by sugar pyrolysis. In brief, we followed our previous work for the coating of 3nm TiO₂¹. Surface carbon coating is often used to increase the electrical conductivity of LiFePO₄ and has proven to be an effective strategy as mentioned by the reviewer^{2,3}. We show below the detailed evaluations:

1.1 The capacity comparison among Bare, Carbon-coated, and TiO₂-Carbon-coated FePO₄ electrodes is summarized in **Figure R6**. Under 0.1C (14.7 mAh/g), the TiO₂-Carbon-coated FePO₄ electrode delivered a similar specific capacity (147 mAh/g) with only Carbon-coated FePO₄ electrode (151 mAh/g), while the Bare-FePO₄ electrode delivered a much worse specific capacity (120 mAh/g; Note: bare particles were annealed under the same condition without mixing with sucrose). Surface carbon coating indeed help increase the capacity of LiFePO₄ electrodes.

Figure R6 Electrochemical cycling of Bare, Carbon-coated and TiO₂-Carbon-coated FePO₄ electrodes in 1 M LiCl aqueous solution under 0.1C (14.7 mA/g). 147 mA/g equals a rate of 1C.

1.2 The main role of TiO₂ is to increase the wettability of carbon-coated FePO₄ electrodes and decrease the contact impedance. To verify the improved solid-liquid interface contact by TiO₂ coating, as shown in **Figure R7a**, the carbon-coated FePO₄ together with carbon cloth substrate is very hydrophobic. After coating the electrode with TiO₂, the 10 μ L water immediately infiltrated the electrode surface. Several pre-wetting steps can be avoided using TiO₂-Carbon-coated FePO₄ electrodes. We are aware that surface coating may affect the Li⁺ diffusivity. We found in the literature⁴ that beta-TiO₂ has similar Li⁺ diffusivity compared to LiFePO₄. Our ALD coated TiO₂ is amorphous. It was shown in other work that amorphous TiO₂ has Li⁺ diffusivity on the same order of magnitude as crystalline TiO₂⁵. Therefore, we find TiO₂ to be a good candidate for coating compared to other common ALD materials such as ZrO₂, Al₂O₃, etc., which showed much higher Li diffusion barriers. We also conducted EIS in 1M LiCl aqueous solution. **Figure R7b** shows the Nyquist plots for the carbon-coated FePO₄ wo/w the 3 nm TiO₂ coating with the equivalent circuit shown in the inset. The resistor R_s corresponds to the electrolyte resistance. The resistors R₁ and R₂ paralleled with the constant phase element (CPE) account for the contact impedance and charge transfer impedance, respectively. The ion diffusion in the host material is described with the Warburg element (Z_w). As shown in **Figure R7b**, the simulated data from the equivalent circuit well fit the impedance data for both electrodes. The values for the different resistances obtained from fitting are listed in **Table R2**. As shown in Table R2, the electrolyte resistance (R_s) and charge transfer impedance (R₂) are almost the same for the carbon-coated FePO₄ wo/w the TiO₂ coating. The contact impedance (R₁) of the electrode without TiO₂ coating is more than double the value of the electrode with TiO₂ coating. Therefore the TiO₂ coating can reduce the contact resistance as well.

Figure R7 (a) Photographic image showing 10 μ L water dropped on the surface of the carbon-coated FePO₄ electrode wo/w 3nm TiO₂ coating. (b) Nyquist plots for the electrode wo/w 3nm TiO₂ obtained by electrochemical impedance spectroscopy (EIS) tests in 1 M LiCl aqueous solution with the frequency ranging from 200 kHz to 100 mHz at a 10 mV amplitude. The dot-dashed lines are the fitting curves by using the equivalent circuit, which is shown as the inset and consists of a resistor (R_s), a resistor (R_1) paralleled with a constant phase element (CPE), and a CPE parallel with a resistor (R_2) which is connected with a Warburg element (Z_w) in series.

Table R2 Electrode resistances for the carbon-coated FePO₄ electrode wo/w 3nm TiO₂ coating obtained from equivalent circuit fitting of EIS results.

	wo TiO ₂	w TiO ₂
R _s	5.2	5.5
R ₁	21.1	9.7
R ₂	19.3	22.6

1.3 We further evaluated the effect of TiO₂ coating on the seeding process. **Figure R8** shows the XRD patterns of the carbon-coated FePO₄ electrode wo/w TiO₂ coating after 20% Li seeding under 4C. There is little difference for both obtained and fitted patterns. The calculated solid solution fraction (SSF) for the electrode without or with TiO₂ coating are 0.429 and 0.428, respectively, which is similar. And the recovered Li/(Li+Na) ratios for the following Li extraction step are almost the same (**Table R3**). Without TiO₂, Li/(Li+Na)_{net} is 0.73 ± 0.01, and with TiO₂, Li/(Li+Na)_{net} is 0.74 ± 0.01. Therefore, the above results show that the TiO₂ coating will not affect SSF generated in the seeding process, as well as the following Li extraction performance. Long term corrosion resilience can be a benefit of TiO₂ coating. We will investigate this in our future study.

Figure R8 XRD of the carbon-coated FePO₄ electrode wo/w 3nm TiO₂ coating after 20% Li seeding under 4C (Dotted lines: obtained patterns; Solid lines: fitted patterns).

Table R3 Recovered Li/(Li+Na)_{net} ratios of the carbon-coated FePO₄ electrode wo/w 3nm TiO₂ coating using 20% Li seeding (Error bars representing the standard deviation of three replicate measurements).

L(0.2) _{4C} -LN(0.7) _{0.1C}		
	wo TiO ₂	w TiO ₂
Li/(Li+Na) _{net}	0.73 ± 0.01	0.74 ± 0.01

Besides coating, we also tested the rate ability and cyclic voltammetry (CV) of TiO₂-carbon-coated FePO₄ electrodes both in 1 M LiCl(aq) solution and 1 M NaCl(aq) solution. **Figure R9a** shows the rate capability of FePO₄ in 1 M LiCl(aq) solution. Specifically, under a 14.7 mA/g (0.1C) current density, we acquired a 147 mAh/g specific capacity. The specific capacity decreased to 101 mAh/g under a 588 mA/g (4C) current density. We also measured the capacity of FePO₄ for Na intercalation. At 14.7 mA/g, the Na intercalation capacity is 88 mAh/g and at 588 mA/g, the capacity is 39 mAh/g, which are both smaller than Li intercalation capacities. For the cycling performance in 1 M NaCl(aq) solution, as shown in **Figure R9b**, two plateaus are found for the charging process, and only one plateau is observed for the discharging process. This asymmetric behavior is due to the formation of the Na_{0.7}FePO₄ intermediate phase during the charging process^{6,7}. It is also consistent with the CV results we see in **Figure R9c**. The CV scan of FePO₄ in 1 M LiCl(aq) solution shows a pair of symmetric anodic and cathodic peaks, with half-wave potential ($E_{1/2} = 0.208$ V vs. Ag/AgCl) close to the thermodynamic value (3.45 V vs. Li/Li⁺ = 0.213 V vs. Ag/AgCl)^{8,9}. In 1 M NaCl(aq) solution, two well-defined current peaks are found for the anodic process, and only one current peak is observed for the cathodic scan. Besides, the value of the current peak in LiCl(aq) is more than three times higher than that in NaCl(aq) at the same scan rate, which demonstrates the fast kinetics of Li⁺ insertion and extraction in FePO₄ hosts.

Figure R9 Electrochemical cycling of the FePO₄ electrodes in (a) 1 M LiCl aqueous solution and (b) 1 M NaCl aqueous solution under different current densities/C rates. 14.7 mA/g equals a rate of 0.1C, while 147 mA/g equals 1C. (c) Cyclic voltammetry (CV) tests for the FePO₄ host in 1 M LiCl/NaCl aqueous solution at a 0.03 mV/s scan rate.

We have added the discussions of surface carbon and TiO₂ coatings in our revised Supplementary Materials accordingly and included **Figure R9** into the **Supplementary Figure 23** in the SI.

References:

1. C. Liu, Y. B. Li, D. C. Lin, P. C. Hsu, B. F. Liu, G. B. Yan, T. Wu, Y. Cui, S. Chu, *Joule* 2020, 4, 1459.
2. M. Herstedt, M. r. Stjerndahl, A. Nyttén, T. r. Gustafsson, H. k. Rensmo, H. Siegbahn, N. Ravet, M. Armand, J. O. Thomas, K. Edström, *Electrochemical and Solid-State Letters* 2003, 6, A202.
3. J. Wang, X. Sun, *Energy Environ. Sci.* 2012, 5, 5163.

4. S. Xu, R. M. Jacobs, H. M. Nguyen, S. Hao, M. Mahanthappa, C. Wolverton, D. Morgan, *Journal of Materials Chemistry A* 2015, 3, 17248.
5. Y.-M. Lin, P. R. Abel, D. W. Flaherty, J. Wu, K. J. Stevenson, A. Heller, C. B. Mullins, *The Journal of Physical Chemistry C* 2011, 115, 2585.
6. P. Moreau, D. Guyomard, J. Gaubicher, F. Boucher, *Chemistry of Materials* 2010, 22, 4126.
7. J. C. Lu, S. C. Chung, S. Nishimura, A. Yamada, *Chemistry of Materials* 2013, 25, 4557.
8. A. K. Padhi, K. S. Nanjundaswamy, J. B. Goodenough, *Journal of the Electrochemical Society* 1997, 144, 1188.
9. Y. Zhu, Y. Xu, Y. Liu, C. Luo, C. Wang, *Nanoscale* 2013, 5, 780.

2- If such information, mentioned in 1 above, has been gathered for Na-intercalation in Na-ion battery types, it would be useful too.

Thanks for the suggestion. The Na-intercalation performance has been added and the discussions are shown above in **Point #1**.

3- There are several normalisations, one from the XRD peak intensities, another from combined total Li+Na content, and perhaps another from the intercalation “capacity” (in synthetic brine solution). This intercalation capacity is normalised at 0.7, as shown in Figure S8, but remains slightly ambiguous, without a clear display of the full absolute capacities at different C-rates. The display of absolute capacities would add clarity to the information.

We thank the reviewer for the suggest and we made clarifications on these normalizations. The total capacity we refer to in the manuscript is 147 mAh/g, measured by 1st cycle (de) intercalation of the electrodes in 1 M LiCl aqueous solution at 14.7 mA/g (**Figure R9a**) between -0.6 V and 0.6 V (vs. Ag/AgCl). The normalization of XRD peaks is explained in the method section. The Li/(Li+Na) is explained in the method as well. In brief, we measure Li and Na concentration in the recovery solution and make sure the total ion amount measured matches the electrochemical deintercalation capacity with ~ 5% error tolerance. Li/(Li+Na) ratio is calculated based on their concentrations in the recovery solution.

4- Is the capacity 70% of the measured capacity at 0.1C? Is it 70% of the measured capacity at the selected, fixed C/n rate? I am not sure how different the rate capability in synthetic brine solution is to be expected compared to the C-rate in an actual battery; nanoscale LFP at 5C in a Li-ion battery typically displays less than 70% of the capacity at 0.1C, therefore, it seems important to show clearly and unambiguously the rate capability in the brine.

We agree with the reviewer that the rate capability of FePO₄ electrodes in aqueous solution needs to be clarified, especially whether 4C seeding of 40% is possible has to be shown without any ambiguity. The 70% capacity is to the measured capacity at 0.1C in 1 M LiCl aqueous solution, which is 147 mAh/g (**Figure R9a**). We do see a decrease in the capacity at a higher C rate in 1 M LiCl aqueous solution. For example,

the capacity is only 101 mAh/g under 4C, only around 68% of the capacity at 0.1C. The delivered capacity at 4C can allow us to seed 40% of Li before the Li extraction step. Moreover, indeed, the rate capability would be different in the synthetic brine solution. As shown in **Figure R2 below**, within the selected water safety window, we could only get 134 mAh/g capacity in 1 mM:1 M LiCl:NaCl mixed solution, which is 91% of the 147 mAh/g capacity achieved in 1 M LiCl aqueous solution. The highest co-intercalation rate in mixed brines we used is 0.5C (Figure 3d in the manuscript), which would further decrease the accessible capacity. Therefore, as shown in **Figure R2b**, we set a limit of 70% of capacity (102.9 mAh/g) to prevent side reactions from damaging the structure under high polarization voltage and use a slower C rate (4.9 mA/g) to fully recover the intercalated ions.

we revised this information to make it clearer in the methods part. The highest C rate we used during Li extraction in mixed brines is only 0.5C (**Table R1**). For most of our cases, we used 0.1C as our extraction C rate. The 4C is only used for Li-seeding which is done in 1M LiCl solutions.

Figure R2 (a) Electrochemical cycling of the electrodes in 1 M LiCl and 1 mM LiCl: 1 M NaCl aqueous solutions under 0.1C (14.7 mA/g). (b) 0.1C (14.7 mA/g) intercalation curve in 1 mM LiCl: 1 M NaCl aqueous solution and C/30 (4.9 mA/g) de-intercalation curve in 30 mM NH_4HCO_3 recovery solution, with the use of 70% of the total capacity (102.9 mAh/g).

We have added the discussions in our revised Supplementary Information accordingly.

5- Some brief contrast of rate capability in standard battery cell format and that for synthetic brine would be of interest. Are the best performing battery materials also the best for Li-extraction?

We thank the reviewer for the suggestion. We cycled the electrodes in 1 M LiClO_4 1:1 (v/v) ethylene carbonate and dimethyl carbonate electrolyte. The measured capacity was 143 mAh/g at 14.3 mA/g (equivalent to 0.1C) current density between 2V and 4V (vs. Li/Li^+) (**Figure R10**). The capacity delivered

at 71.5 mA/g (0.5C) and 572 mA/g (4C) in 1M LiClO₄ are 121 mAh/g and 85 mAh/g respectively. We included **Figure R10** into **Supplementary Figure 23**.

The two fields of battery and element-extraction do share many similar criteria for electrode evaluations, such as the rate capability, structure stability, and delivered capacity. In these aspects, we believe that better capacity, rate capability, stability of battery materials, also benefit the performance of lithium extraction. However, we may need to adopt a completely different strategy in some respects. For example, in the battery field, people tend to choose bigger particle (or nanoparticle cluster) size to achieve better interfacial stability, higher tapped density as well as volumetric energy density^{1,2}. However, in the case of Li extraction with FePO₄ particles, based on our finding on the SS formation effect on Li selectivity, smaller particles, such as nano-FePO₄, could have better intrinsic solid-solution (de)lithiation mechanism. Another example is that we may need different surface modifications for the materials. Since most of the natural sources are water-based solutions, in long-term we need a good hydrophilic anti-corrosion layer to induce a good contact and help suppress the structure deterioration caused by the aqueous solution. This could also be different from the current Li-ion battery systems with organic electrolytes.

Figure R10 Electrochemical cycling of the FePO₄ electrodes in the organic solution. The electrolyte was 1 M LiClO₄ dissolved in a 1:1 (v/v) mixture of ethylene carbonate (EC) and dimethyl carbonate (DMC). (143 mA/g equals a rate of 1C.)

References:

1. J. Kim, H. Lee, H. Cha, M. Yoon, M. Park, J. Cho, *Adv. Energy Mater.* 2018, 8, 1702028.
2. H. Li, J. Li, X. Ma, J. R. Dahn, *Journal of The Electrochemical Society* 2018, 165, A1038.

6- A few comments have been made on the annotated copy of the word format file. Some comments refer to the ranges of x values in brackets, which I found confusing or may have a typo.

We appreciate the revision from the reviewer. We have modified our manuscript accordingly. Specifically, the notation “ Li_xFePO_4 , $x = [0.500:0.125:0.875]$ ” means the group of $\text{Li}_{0.500}\text{FePO}_4$, $\text{Li}_{0.625}\text{FePO}_4$, $\text{Li}_{0.750}\text{FePO}_4$ and $\text{Li}_{0.875}\text{FePO}_4$, with “0.125” in the middle means the change step.

I recommend publication after addressing the abovementioned clarifications.

Reviewer #3:

The authors present interesting work that is aimed to shed more light on the role of pre-formation of solid-solution in olivine FePO_4 electrode host material that is applied in the emerging approach of selective electrochemical (Lithium) ion extraction from geothermal sources.

It is exciting to see how comprehensive knowledge gained in one field of electrochemistry (battery energy storage systems) is utilized in another one (element extraction from natural sources).

The authors indeed conducted a systematic study and provide quite extensive results that do rather well support their proposed mechanism of the promoted enhanced ion insertion selectivity in the host FePO_4 (FP) material.

We appreciate the reviewer's positive comments and valuable feedback on our manuscript. We have added the discussions of particle size effects and modified our manuscript accordingly. Below please see our detailed point-to-point replies to all the questions.

Nonetheless, there are still some missing parts in the puzzle.

A) One important aspect missing is the effect of FP particle size (lateral and thickness dimensions). The authors show SEM image of the “parent” LiFePO_4 (LFP) particles (Figure S13) where it can be seen that the micrometer-sized particles indeed have platelet-like morphology. It can be deduced from the image that the particles somewhat vary in the thickness with the thick ones showing thicknesses of about 400 nm or probably even more in some cases. The important question is how do the particle dimensions (especially the platelet thickness governing the [010] 1D channel length) impact: i) solid solution formation during the high-rate Lithium “seeding” step, and ii) the consequential ion insertion Li/Na selectivity enhancement. Experimental investigation of this aspect should be provided.

We thank the reviewer for this great question. In order to investigate the effects of particle size, we systematically studied two more types of particles with different sizes.

A.1 Morphology and structure verification. The first added particle is commercial LiFePO_4 (Comm- LiFePO_4) bought from *MTI Corporation*, as shown in **Figure R11a**; they have spherical shapes, with radii ranging from 1 μm to 4 μm . **Figure R11b** shows the particles (Synthesized- LiFePO_4) we mainly used in the manuscript, synthesized with mixed water and polyethylene glycol 400 as the solvent. Synthesized- LiFePO_4 particles share a similar lateral dimension (1 μm ~ 4 μm) with Comm- LiFePO_4 particles but have a smaller dimension in thickness (< 500 nm). Besides, we synthesized a new platelet-like particle (EG- LiFePO_4) shown in **Figure R11c**. As for the synthesis, instead of 24 mL polyethylene glycol 400, we used 24 mL ethylene glycol, keeping all the other parameters the same in the original recipe. The new recipe gives us smaller particles both in lateral (< 1 μm) and thickness dimensions (< 150 nm). For comparison, both Comm- LiFePO_4 particles and EG- LiFePO_4 particles have carbon coatings and follow the same chemical extraction step as Synthesized- LiFePO_4 particles, described in the Methods part of the manuscript, to produce Comm- FePO_4 particles and EG- FePO_4 particles. We also did the 3nm TiO_2 coating for the assembled electrodes. Since there may be slight differences in the lattice parameters of particles with

different shapes or dimensions, and different lattice parameters will affect our deconvolution results of XRD patterns; thus, we conducted LeBail refinements for commercial $\text{LiFePO}_4/\text{FePO}_4$ and EG $\text{LiFePO}_4/\text{FePO}_4$ particles to determine the exact lattice parameters for these two particles. The refinement results are shown in **Figure R12**, with summarized lattice parameters listed in **Table R4**. We then follow the steps described in supplementary materials to deconvolute solid-solution fractions from diffraction patterns with these acquired lattice parameters. More details will be introduced below in A.3 *Solid solution formation and Li selectivity*.

Figure R11 SEM images of (a) **Comm- LiFePO_4** particles, (b) **Synthesized- LiFePO_4** platelet-like particles, and (c) **EG- LiFePO_4** platelet-like particles.

Figure R12 Le Bail refinement of (a) Comm-LiFePO₄ particles, (b) chemical-extracted Comm-LiFePO₄ particles, (c) EG-LiFePO₄ particles, and (d) chemical-extracted EG-FePO₄ particles.

Table R4 Summarized lattice parameters for all three particles.

	Comm-LiFePO ₄	Synthesized-LiFePO ₄	EG-LiFePO ₄		Comm-FePO ₄	Synthesized-FePO ₄	EG-FePO ₄
a (Å)	10.321	10.347	10.343	a (Å)	9.818	9.819	9.829
b (Å)	6.005	6.007	5.995	b (Å)	5.792	5.798	5.794
c (Å)	4.691	4.700	4.700	c (Å)	4.781	4.785	4.779
α				α			
β		90		β		90	
γ				γ			

A.2 Battery material performance verification. We then compared the battery material performance of these three different particles by electrochemical cycling of the electrodes in 1 M LiCl aqueous solutions under

0.1C (13.5 mA/g for Comm-FePO₄; 14.7 mA/g for Synthesized-FePO₄; 12.6 mA/g for EG-FePO₄) between -0.6 V and 0.6 V (vs. Ag/AgCl). As shown in **Figure R13**, interestingly, within the selected operating window, Comm-FePO₄ could only deliver 135 mAh/g capacity during intercalation, together with a decreased capacity during the de-intercalation process. For the synthesized EG-FePO₄ particles, we could only achieve 126 mAh/g of specific capacity under 0.1C (12.6 mA/g). The existence of defects may be the reason for the decreased capacity of commercial FePO₄ and synthesized EG-FePO₄ particles. However, we think this point might be out of the scope of this paper.

Figure R13 Electrochemical cycling of Comm-FePO₄ (13.5 mA/g), Synthesized-FePO₄ (14.7 mA/g), and EG-FePO₄ (12.6 mA/g) electrodes in 1 M LiCl aqueous solutions between -0.6 V and 0.6 V (vs. Ag/AgCl).

A.3 Solid solution formation and Li selectivity. Finally, we investigated the solid solution formation during the high C-rate Li-seeding step and the consequential Li extraction performance. **Figure R14** shows example XRD patterns for Comm-FePO₄, Synthesized-FePO₄, and EG-FePO₄ electrodes with 20% Li seeding under 4C. We tested three parallel electrodes for each type of particles and summarized the solid solution fractions in **Figure R15a**. Obviously, the total solid solution fraction generated in the Comm-FePO₄ electrode is much smaller (Total SS = 0.22 ± 0.009) than that in the Synthesized-FePO₄ (Total SS = 0.43 ± 0.001) or EG-FePO₄ electrode (Total SS = 0.43 ± 0.009). Besides, in the case of Comm-FePO₄ electrode, the intercalated Li⁺ ions mainly formed fully occupied LiFePO₄ phase, as indicated by the pronounced LiFePO₄ (020) peak in **Figure R14a**. Although the EG-FePO₄ electrode has similar total solid solution fractions with Synthesized-FePO₄, the XRD patterns are quite different, as well as the compositions of these solid solutions (**Figure R14b, R14c, and Figure R15a**). Specifically, most of the SS phases generated in the EG-FePO₄ electrode are high-Li SS (0.42 ± 0.013), with low-Li SS being only ~ 3% of the total SS phases. In contrast, for the Synthesized-FePO₄ electrode, low-Li SS phases occupy more than one-third of the total SS phases. This indeed shows that the particle shape and size have significant influences on the formation of solid solution and on the fractions of each SS phase. Particles with smaller dimensions are easier to form solid solutions under similar high C rates, which is consistent with the findings that Li

solubility increases with decreasing particle size due to the increased interfacial energy per unit volume¹⁻³. We would like to perform further study to have a systematic investigation of size effects on the seeding process, which we think could be a very interesting follow up work.

We further tested the selectivity of the three particles, with or without initial Li seeding. As shown in **Figure R15b**, the dark gray squares represent the recovered Li/(Li+Na) ratios for the empty hosts without initial Li seeding, using 70% of the total capacity (L(0)-LN(0.7)_{0.1C}, 0.7 is normalized by the capacity at 0.1C 1M LiCl solution for each type of particle); while the blue squares represent the recovered Li/(Li+Na)_{net} ratios for the 20% pre-seeded hosts under 4C, using the remaining 50% of capacity for Li extraction (L(0.2)_{4C}-LN(0.7)_{0.1C}). First of all, we observed improved selectivity with the Li-seeded hosts for all these three different particles, demonstrating the reliability of the seeding method. In addition, we do find that as the size decreases, or more accurately, the particle [010] dimension decreases, the selectivity of lithium extraction increases for both seeded and empty hosts. For the smallest EG-FePO₄ platelet-like particles, the recovered Li ratio Li/(Li+Na)_{net} is 0.77 ± 0.01 for L(0)-LN(0.7)_{0.1C} and 0.83 ± 0.01 for L(0.2)_{4C}-LN(0.7)_{0.1C} comparing to 0.63 ± 0.01 for L(0)-LN(0.7)_{0.1C} and 0.72 ± 0.03 for L(0.2)_{4C}-LN(0.7)_{0.1C} of larger Synthesized-FePO₄ particles. Even though it looks like that the effect of Li-seeding on Li selectivity improvement is weakened since ratio difference is smaller for the small-sized EG-FePO₄, we would like to point to this as a beneficial “intrinsic seeding” process. The inherent solid-solution (de)-lithiation mechanism of small particles allows Li-seeding in 1:1000 Li to Na mixed brines during the Li extraction step, which leads to empty host having high Li selectivity without the seeding step. Overall, the study of these three particles again proves our conclusion that the high-Li SS phases could promote Li co-intercalation competitiveness in FePO₄ (**Figure R15**).

Figure R14 Example XRD patterns of (a) Comm-FePO₄, (b) Synthesized-FePO₄, and (c) EG-FePO₄ electrodes after 20% of Li seeding under 4C. For Comm-FePO₄, 135 mA/g equals a rate of 1C; for Synthesized-FePO₄, 147 mA/g equals a rate of 1C; for EG-FePO₄, 126 mA/g equals a rate of 1C (Dotted lines: obtained patterns; Solid lines: fitted patterns).

Figure R15 (a) Total SS, Low-Li SS, and High-Li SS fractions for the three electrodes under the same seeding rate 4C (540 mA/g for Comm-FePO₄; 588 mA/g for Synthesized-FePO₄; 504 mA/g for EG-FePO₄) with the 20% seeding range L(0.2)_{4C}. (b) Recovered Li/(Li+Na)_{net} of the three electrodes until 70% of the total capacity used under 0.1C, with either initial 20% of Li seeding under 4C (L(0.2)_{4C}-LN(0.7)_{0.1C}) or without any initial seeding process (L(0)-LN(0.7)_{0.1C}). (Error bars represent the standard deviation of three replicate measurements.)

References:

1. N. Meethong, H.-Y. S. Huang, W. C. Carter, Y. M. Chiang, *Electrochemical and Solid State Letters* 2007, 10.
2. G. Kobayashi, S.-i. Nishimura, M.-S. Park, R. Kanno, M. Yashima, T. Ida, A. Yamada, *Adv. Funct. Mater.* 2009, 19, 395.
3. H. Liu, F. C. Strobridge, O. J. Borkiewicz, K. M. Wiaderek, K. W. Chapman, P. J. Chupas, C. P. Grey, *Science* 2014, 344, 7.

We included all the above figures/tables into the Supporting Materials and added the following text into the revised manuscript:

“Besides the (de)intercalation rates, the formation of the Li solid solution phases depends strongly on the particle dimensions. Particles with smaller dimensions are easier to form solid solutions under the same global C rates since Li solubility increases with decreasing particle size due to the increased interfacial energy per unit volume^{28,37-39}. Specifically, we studied two other particles with different dimensions. One is commercial LiFePO₄ particles (Comm-LiFePO₄, *MTI Corporation*), as shown in Supplementary Figure 15a. The Comm-LiFePO₄ particles have spherical shapes, with wide radius distributions. The radii mainly range from 1 μm to 4 μm. As comparison, Supplementary Figure 15b shows the original particles (Synthesized-LiFePO₄) we discussed for all the experiments unless mentioned. Synthesized-LiFePO₄ particles were synthesized with mixed water and polyethylene glycol 400 as the solvent, which share a similar lateral dimension (1 μm ~ 4 μm) with Comm-LiFePO₄ particles but have a smaller thickness (< 500

nm). Besides, we synthesized a new platelet-like particle (EG-LiFePO₄) as shown in Supplementary Figure 15c with smaller lateral (< 1 μm) and thickness dimensions (< 150 nm). Both Comm-FePO₄ particles and EG-FePO₄ particles follow the same preparation methods as Synthesized-FePO₄ host (See Methods for more details).

Supplementary Figure 16 shows example XRD patterns for Comm-FePO₄, Synthesized-FePO₄, and EG-FePO₄ electrodes with 20% Li seeding under 4C (1C for Comm-FePO₄, Synthesized-FePO₄, and EG-FePO₄ is 135 mA/g, 147 mA/g, and 126 mA/g, respectively; See Supplementary Materials for capacity evaluation). We seeded 20% Li for each type of particles and summarized the solid solution fractions in Supplementary Figure 17a. Obviously, the total solid solution fraction generated in the Comm-FePO₄ electrode is much smaller (Total SS = 0.22 ± 0.009) than that in the Synthesized-FePO₄ (Total SS = 0.43 ± 0.001) or EG-FePO₄ electrode (Total SS = 0.43 ± 0.009). Besides, in the case of Comm-FePO₄ electrode, the intercalated Li⁺ ions mainly formed fully occupied LiFePO₄ phase, as indicated by the pronounced LiFePO₄ (020) peak in Supplementary Figure 16a. Although the EG-FePO₄ electrode has similar total solid solution fractions with Synthesized-FePO₄, most of the SS phases generated in the EG-FePO₄ electrode are high-Li SS (0.42 ± 0.013), with low-Li SS being only ~ 3% of the total SS phases (Supplementary Figure 16b, Supplementary Figure 16c, and Supplementary Figure 17a). In contrast, for the Synthesized-FePO₄ electrode, low-Li SS phases occupy more than one-third of the total SS phases. Particles with smaller dimensions can form solid solutions more easily under the same global C rates with larger high-Li SS fractions.

We further tested the Li selectivity of the three particles, with or without initial Li seeding. As shown in Supplementary Figure 17b, first of all, we observed improved Li selectivity with the Li-seeded hosts for all the three different particles, validating the effectiveness of the seeding method. In addition, we find that as the dimension decreases, the Li selectivity increases for both seeded and empty hosts. It is also worth noting that, for the smallest EG-FePO₄ platelet-like particles, the recovered Li/(Li+Na) ratio difference between L(0)-LN(0.7)_{0.1C} (Li/(Li+Na)_{net} = 0.77 ± 0.01) and L(0.2)_{4C}-LN(0.7)_{0.1C} (Li/(Li+Na)_{net} = 0.83 ± 0.01) hosts is smaller than that of Synthesized-FePO₄ platelet-like particles (Li/(Li+Na)_{net} = 0.63 ± 0.01 for L(0)-LN(0.7)_{0.1C}; Li/(Li+Na)_{net} = 0.72 ± 0.03 for L(0.2)_{4C}-LN(0.7)_{0.1C}). This indicates that the inherent solid-solution (de)-lithiation mechanism of small particles stimulate a “intrinsic seeding” process that without the seeding step, high-Li SS phases can be generated in the Li extraction step in 1: 1000 Li to Na mixed solution. Overall, the increase in high-Li SS fraction together with improved Li selectivity (Supplementary Figure 17) among these three particles again proves our conclusion that the high-Li SS phases could promote Li co-intercalation competitiveness in FePO₄.”

B) We know that Li-ion battery-grade LFP materials have to be made of LFP crystallites of sub-micron dimensions in order to provide needed sufficient rate performance. At the same time it was found (e.g. ref. R1 and R2) that in these types of materials the total amount of observed solid-solution is rather small even at very high rates of lithiation and the observed relaxation time of the formed non-equilibrium solid-solution was found to be short (in the order 10 seconds or several minutes). On the other hand, more lately there were reports (e.g. ref. R3, ref. 44 in this manuscript) showing results with partially lithiated platelet LFP (Li_{0.5}FePO₄) that exhibited very large fractions of solid-solution (even larger than 50%). It appears that in the current work the authors have synthesized very similar platelet LFP particles and moreover the solid-solution fractions obtained during high-rate “seeding” step are comparable to those found in the work of Li

et al. (ref. R3, ref. 44 in this manuscript). So it seems that this type of large/thick LFP particles exhibit high solid-solution fractions after partial high-rate lithiation in both organic and water-based electrolytes. Time evolution of a diffusion process is strongly dependent upon the medium dimension (L) wherein it is let to diffuse and in general the characteristic propagation time is proportional to $L \times L = L^2$. Thus surely the formation and relaxation of the non-equilibrium Li solid solution phase(s) has to be strongly dependent upon the particle dimensions. The authors are encouraged to elaborate this point. Preferable would be to extend the experimental study to at least one (notably) different particle size/thickness. For example they could employ some battery-grade LFP material with good high C-rate performance.

We thank the reviewer for the great question and we agree that the formation and relaxation of the non-equilibrium Li solid solution phases strongly depend on the particle dimensions. Specifically, we studied two other particles with different dimensions, Comm-FePO₄ and EG-FePO₄, besides Synthesized-FePO₄ particles mainly used in the manuscript. The detailed discussions are included in the above **Point #A**. In summary, we first verified the morphology and structure of Comm-FePO₄ and EG-FePO₄ particles (**Figures R11, R12, and Table R4**), followed by evaluating the battery material performance in **Figure R13**. Finally, we investigated the solid solution formation during the high C-rate Li-seeding step and the consequential Li extraction performance in **Figure R14** and **Figure R15**. There is one point we would like to bring up about solid solution decomposition. For the platelet particles, the [010] channels are exposed to the solutions. It was reported by this paper¹ that the phase decomposition involves solvation and desolvation of Li ions to move from one channel to the other channel. This process might not exist in commercial spherical particles since not all exposed surfaces are (010) faces. This is a complicated problem that it is hard for us to draw conclusions on whether this solvation and desolvation will affect the SS formation fraction and decomposition rate. However, we think it is reasonable that if the solvation and desolvation process can slow down the SS phase decomposition, it will enhance the SS phase formation from the beginning.

Below we summarize our experimental results mainly from above **Point #A**:

B.1 Particles with smaller sizes or dimensions are easier to form solid solutions under similar high C rates which is consistent to the findings that Li solubility is found to increase with decreasing particle size due to the increased interfacial energy per unit volume²⁻⁴.

B.2 Smaller particles show larger high-Li phase fraction at the same condition of seeding. Specifically, for the smallest EG-FePO₄ particles, ~ 97% of the total SS phases generated are high-Li SS.

B.3 Selectivity is improved for the initially Li-seeded hosts, compared with empty hosts for all these three different particles, demonstrating the reliability of the seeding method. Besides, as the particle thickness decreases, the selectivity of lithium extraction increases for both seeded or empty hosts.

B.4 For the smallest EG-FePO₄ platelet-like particles, the improvement in selectivity by the Li seeding method is not as significant as that for the bigger Synthesized-FePO₄ platelet-like particles. However, the Li selectivity is much higher. This indicates that the inherent solid-solution (de)-lithiation mechanism of smaller particles can induce the “intrinsic seeding” process for empty EG-FePO₄ in 1: 1000 Li to Na mixed solutions. This is beneficial since smaller seeding capacity range is needed to achieve the same Li/(Li+Na) ratio which enlarges the capacity usage of the host for Li extraction.

We have added these data and additional discussions in our revised manuscript accordingly.

References:

1. Y. Li, H. Chen, K. Lim, H. D. Deng, J. Lim, D. Fraggedakis, P. M. Attia, S. C. Lee, N. Jin, J. Moškon, Z. Guan, W. E. Gent, J. Hong, Y.-S. Yu, M. Gabersček, M. S. Islam, M. Z. Bazant, W. C. Chueh, *Nat. Mater.* 2018, 17, 915.
2. N. Meethong, H.-Y. S. Huang, W. C. Carter, Y. M. Chiang, *Electrochemical and Solid State Letters* 2007, 10.
3. G. Kobayashi, S.-i. Nishimura, M.-S. Park, R. Kanno, M. Yashima, T. Ida, A. Yamada, *Adv. Funct. Mater.* 2009, 19, 395.
4. H. Liu, F. C. Strobridge, O. J. Borkiewicz, K. M. Wiaderek, K. W. Chapman, P. J. Chupas, C. P. Grey, *Science* 2014, 344, 7.

C) How does FP crystal size/thickness affect the possible practical rate of the process of Li extraction from Na-Li brine mixtures?

We thank the reviewer for the question. We compared the rate capability between Synthesized-FePO₄ and EG-FePO₄ particles in **Figure R16**. For the smaller EG-FePO₄ particles, although smaller capacity was observed using the current synthesis method, similar rate capability was observed for both particles. Additionally, smaller EG-FePO₄ showed higher intercalation voltage (lower overpotential) and smaller hysteresis at 4C intercalation which indicates that the selectivity at higher extraction rate can be better than the large Synthesized-FePO₄ particles. This is based on our results (**Supplementary Figure 8** in the manuscript) that larger C rate in the extraction step will lead to larger overpotential for Li intercalation which decreases Li selectivity. In the case of small EG-FePO₄ particles, a larger voltage differences between Li and Na intercalation at high rates might be maintained which could lead to higher Li selectivity. We think our synthesized EG-FePO₄ particles did not exhibit their full potential yet due to the intrinsic defects as can be seen from the smaller delivered capacities. Definitely, further work needs to be done on improving the synthesis method to improve the quality of the EG-FePO₄ particles.

We integrated **Figure R16a** into **Supplementary Figure 23** and included **Figure R16b** as **Supplementary Figure 24** in the SI.

Figure R16 Electrochemical cycling of the (a) Synthesized-FePO₄ and (b) EG-FePO₄ in 1 M LiCl aqueous solution and under different current densities/C rates. For Synthesized-FePO₄, 147 mA/g equals 1C; while for EG-FePO₄, 126 mA/g equals 1C.

D) Based on the answering the question of the particle size/thickness effect in upper points A), B) and C) the authors should be able to provide guidelines for the most suitable FP host material particle size (thickness) that would enable most selective end most efficient Li extraction from mixed Na-Li salt solutions.

Thanks for the great suggestion. Based on our current results, we observe improved selectivity for particles with smaller dimensions (**Figure R15**). This is an important aspect to guide the selection of particles. Besides SS phase formation, similar to battery, the evaluation of Li-extraction from natural sources also depends on other parameters including the rate capability, structure stability, and delivered capacity. For the smaller EG-FePO₄ particles, smaller capacity was observed using the current synthesis method as shown in **Figure R13** and **Figure R16**. More work needs to be done to improve the synthesis method to increase the specific capacity for smaller particles. If larger capacity can be achieved, then the advantage is obvious for choosing smaller particles. We would like to thank the reviewer for all the comments on the particle dimensions. The size effect study during the revision process indeed opens a new direction that looks promising to us. More works are toned to be done to give a precise size range of particle dimensions that would have optimized Li selectivity and stability.

References:

1. J. Kim, H. Lee, H. Cha, M. Yoon, M. Park, J. Cho, *Adv. Energy Mater.* 2018, 8, 1702028.
2. H. Li, J. Li, X. Ma, J. R. Dahn, *Journal of The Electrochemical Society* 2018, 165, A1038.

E) How does the total amount of solid-solution fraction depend upon rate of lithiation during the Li pre-seeding step? Figure S7 shows relative amounts of the selected solid-solution phase fractions but it does not show the total (absolute) amount of the sum of all the solid-solution compositions. It is not clear for example what is the sum of all solid-solution compositions for C/10 pre-lithiation. At this (low) rate of lithiation the expected solid-solution formation is expected to be much lower compared to high C-rates. At very low rates of partial lithiation the solid-solution formation should tend to very small values since the active particle population should tend to zero when current density (C-rate) goes to zero. At least this is the current understanding of the topic for particle population lithiation in Li ion battery systems with organic electrolytes.

We thank the reviewer for the great question. We added the total SS fraction for each seeding conditions and updated the figure as **Figure R17** below. We observed a relatively lower high-Li SS and total SS fractions for the 20% seeded host at 0.1C comparing to higher C rates. It looks like the intrinsic solid solution property of small particles (Synthesized-FePO₄/EG-FePO₄) makes generating solid solution possible under slow C rates.

The formation of SS phases is a complicated process that involves intercalation, solid diffusion and cross channel rearrangement. This process is determined mainly by the particle size, morphology, surface chemistry and solution composition. For the platelet particles, the rearrangement (decomposition of SS phases) involves (de)solvation processes which could be different from the commercial FePO₄ particles. We observed a much lower SS fractions in the case of commercial FePO₄ particles with large/thick dimensions comparing to the platelet particles (Synthesized-FePO₄/EG-FePO₄) (**Figure R15a**). After 20% Li seeding under 4C, the total SS fraction is only 0.22 ± 0.009 .

Figure R17 Solid solution fractions under different seeding C rates with the same 20% seeding range $L(0.2)_{0.1/2/4/6/8C}$.

We updated the **Supplementary Figure 7** in the manuscript.

References:

1. N. Meethong, H.-Y. S. Huang, W. C. Carter, Y. M. Chiang, *Electrochemical and Solid State Letters* 2007, 10.
2. G. Kobayashi, S.-i. Nishimura, M.-S. Park, R. Kanno, M. Yashima, T. Ida, A. Yamada, *Adv. Funct. Mater.* 2009, 19, 395.
3. H. Liu, F. C. Strobridge, O. J. Borkiewicz, K. M. Wiaderek, K. W. Chapman, P. J. Chupas, C. P. Grey, *Science* 2014, 344, 7.

F) It is not clearly described and explained by the authors what are the reasons for the statement that high-rate Li pre-seeded FP particles would preferentially insert Na at the edges of a particles (Scheme in Figure 2a and Scheme S1). Should we not expect that the corresponding pre-seeded Li solid-solution be formed somehow randomly within a particle? Please do explain this your hypothesis more in a detail. Do you have any experimental data (EDS elemental mapping, local electron diffraction, SEND) to support it?

We thank the reviewer for pointing this out, and we did not realize the schematic drawing can be misleading. The preferential insertion of Na at the edges of a particle is for the empty FePO₄ hosts without the initial Li seeding process. As shown in **Figures 1c and 1d** in the manuscript, besides Li and Na phase-separation, we witnessed a non-uniform Na mapping signal across the particle with higher intensity near the edges. For the high-rate Li pre-seeded FePO₄ particles, as pointed out by the reviewer, the pre-seeded Li solid-solution should be formed randomly within a particle, which further leads to the random distribution of Na after doing extraction in 1:1000 Li:Na solution. To verify the assumption, EDS was used to map the elemental distribution of L(0.2)_{4C}-LN(0.7)_{0.1C} particles (20% Li-seeding under 4C, with 50% capacity used for Li extraction under 0.1C). As shown in **Figure R18**, we do see a more uniform distribution of Na within the particle and Na are not concentrated at the edges

Given the analysis above, we updated the scheme of high C rates Li seeding process and Li-Na co-intercalation process (**Figure R19**). The Na phase domains are drawn as randomly formed within the particle. We also updated the schematic illustration of Li-Na co-intercalation pathways in **Figure R20**.

Figure R18 STEM image and its corresponding EDS elemental mapping (Na, O, Fe) of the $L(0.2)_{4C}$ - $LN(0.7)_{0.1C}$ particle.

Figure R19 Schematic showing high C rates Li seeding process and Li-Na co-intercalation process. The inset illustrates the possible intercalation pathways at the electrode-solution (E/S) interface.

Figure R20 Schematic illustration of Li-Na co-intercalation pathways for empty FePO_4 hosts, low-rate Li pre-seeded hosts, and high-rate Li pre-seeded hosts.

We updated the **Figure 2** and **Supplementary Scheme 1** in the manuscript.

References:

1. J. C. Lu, S. C. Chung, S. Nishimura, A. Yamada, *Chemistry of Materials* 2013, 25, 4557.
2. C. Delmas, M. Maccario, L. Croguennec, F. Le Cras, F. Weill, *Nat. Mater.* 2008, 7, 665.
3. Y. Li, F. El Gabaly, T. R. Ferguson, R. B. Smith, N. C. Bartelt, J. D. Sugar, K. R. Fenton, D. A. Cogswell, A. L. D. Kilcoyne, T. Tyliczszak, M. Z. Bazant, W. C. Chueh, *Nat. Mater.* 2014, 13, 1149.
4. Y. Li, H. Chen, K. Lim, H. D. Deng, J. Lim, D. Fraggedakis, P. M. Attia, S. C. Lee, N. Jin, J. Moškon, Z. Guan, W. E. Gent, J. Hong, Y.-S. Yu, M. Gaberšček, M. S. Islam, M. Z. Bazant, W. C. Chueh, *Nat. Mater.* 2018, 17, 915.

G) Based on the DFT results that you show in Figure 4d what should one expect in the voltage curves V vs Ag/AgCl RE? For example, let say that we pre-lithiate FP up to $\text{Li}_{0.5}\text{FePO}_4$ and afterwards: i) continue lithiation with low rate (e.g. $-C/100$), or ii) wash the electrode and transfer it in Na-system (pure Na-electrolyte) and perform intercalation of Na at low rate. Should we observe a distinct voltage step to lower voltage when switching from Li to Na intercalation? Do you have any experimental confirmation in regard of this? Please provide some additional explanation of the DFT results in Figure 4d – e.g. can be the difference in ΔG values directly translated in the difference in the corresponding voltage curves?

We thank the reviewer for the great suggestion. We conducted the following tests. We compared the Li-Na intercalation potential difference of either the empty host or the Li pre-seeded host. As shown in **Figure R21**, the dash-dot lines show the 0.01C intercalation curves of the empty host ($\text{L}(0)$) in 1 M LiCl and 1 M NaCl aqueous solutions, until 12.5% capacity is used, in which the potential difference (ΔV_1) is 0.23 V. Similarly, the solid lines represent the 0.01C intercalation curves of the 50% Li pre-seeded hosts under 4C ($\text{L}(0.5)_{4C}$), in which the potential difference (ΔV_2) is 0.27 V, larger than that of the empty host. Two main conclusions we would like to draw based on the results. First, both the empty host and Li pre-seeded host

show the Li preference over Na, indicated by the higher intercalation potential of Li^+ over Na^+ . Second, for $\text{L}(0.5)_{4\text{C}}$, the Li pre-seeded host exhibited a stronger Li preference than the empty host, supported by a larger Li-Na intercalation potential difference ($\Delta V_2 > \Delta V_1$). Generally, our observations match our DFT trend. However, there could be some numerical discrepancies for the calculated and experimental values. For the experimental part, as shown in **Supplementary Figure 5** in the SI, a mixture of different SS phases is generated under 4C with 50% of Li seeding, rather than the pure intermediate $\text{Li}_{0.5}\text{FePO}_4$ phase. For the calculated values, we did not consider contributions from finite temperature effects and configurational entropy, so they are strictly valid for 0K. Also, although we tested a large number of supercell configurations, the specific sampling of configurations that we explored may still play a role in the energy differences calculated. Overall, both experimental and computational results show a similar Li preference trend.

Figure R21 Intercalation curves of empty hosts ($\text{L}(0)$; dash-dot lines) and 4C 50% Li-preseeded hosts ($\text{L}(0.5)_{4\text{C}}$; solid lines) in either 1 M LiCl (green) or 1 M NaCl (orange) aqueous solution until 12.5% of capacity are used under 0.01C (equivalent to 1.47 mA/g).

We included **Figure R21** as **Supplementary Figure 14** in the Supporting Materials and added the following text into the revised manuscript:

“We also compared the Li and Na intercalation potential difference of both the empty host and the 50% Li pre-seeded host under 4C. As shown in Supplementary Figure 14, the potential difference (ΔV_1) is 0.23 V for empty host and is 0.27 V (ΔV_2) for $\text{L}(0.5)_{4\text{C}}$ host. The exhibited Li preference is stronger for $\text{L}(0.5)_{4\text{C}}$, than the empty host ($\Delta V_2 > \Delta V_1$). Therefore, both experimental and computational results show the same trend of Li preference for Li pre-seeded hosts.”

H) How it would behave FP host material with very small particles – let say nano-LFP that was shown to have intrinsic solid-solution (de)-lithiation mechanism in the whole Lithium compositional span? Would be nano-LFP the most- or the least-suitable material for selective Li extraction from Li-Na solution mixtures?

Based on our current results, we did observe improved selectivity for particles with smaller dimension, in our case [010] dimension of ~ 150 nm (**Figure R15**). Therefore, [010] dimension in the nanoscale can help improve the Li selectivity. However, there is also a decrease in the delivered specific capacity when we changed to smaller-sized particles, as shown in **Figure R13** and **Figure R16**. Similar to the battery study, in the field of Li-extraction from natural sources, other parameters like the rate capability, structure stability, and delivered capacity are also important. With the current synthesis method, there is a trade-off in selectivity and capacity. If smaller particles can be improved to have both large specific capacity and stability, it can be promising for Li extraction. More work needs to be done in the future to investigate all these aspects.

References:

1. J. Kim, H. Lee, H. Cha, M. Yoon, M. Park, J. Cho, *Adv. Energy Mater.* 2018, 8, 1702028.
2. H. Li, J. Li, X. Ma, J. R. Dahn, *Journal of The Electrochemical Society* 2018, 165, A1038.

References provided by the reviewer:

R1: Liu, H. et al., Capturing metastable structures during high-rate cycling of LiFePO₄ nanoparticle electrodes. *Science* 344 (6191), 1451–1452 (2014).

R2: Zhang, X. et al., Rate-induced solubility and suppression of the first-order phase transition in olivine LiFePO₄. *Nano Letters* 14, 2279–2285 (2014).

R3: Li, Y. et al., Fluid-enhanced surface diffusion controls intraparticle phase transformations. *Nature Materials* (2018).

REVIEWER COMMENTS

Reviewer #1 (Remarks to the Author):

The authors answered correctly to all my questions and comments. In my opinion the manuscript is ready for its publication.

I would like to thank the authors for their effort answering my comments and improving the article and congrats them for the excellent work done

Reviewer #2 (Remarks to the Author):

The authors have produced an interesting, original piece of work, with potential for practical applications. The content is described well and after addressing the comments from reviewers, the manuscript is very sound and thorough. I recommend publication.

Reviewer #3 (Remarks to the Author):

The authors have in their response to the questions and concerns of the referees provided quite large amount of additional data and information. Certainly I can agree that the authors have put additional efforts in defending their work and reasoning.

On the other hand the authors do not provide convincing enough data to support their main hypothesis. Namely, they propose that the high Li solid solution phase(s) are the main reason behind the Li/Na insertion selectivity. I think this is somewhat overemphasized statement. First of all, the authors themselves clearly present data that show that Li intercalation into FePO₄ host structure is thermodynamically (e.g. Supplementary Figure 14) as well as kinetically (e.g. Supplementary Figure 23) favored over Na intercalation. Furthermore, in the Supplementary Figure 19 there are shown C/10 galvanostatic charge/discharge curves in water system. Surprisingly there is no explanation provided for the observed very low performance of the "EG- FePO₄ particles". At the same time the authors in their response to the referees show the same data (Figure R13) and there is a sentence: "The existence of defects may be the reason for the decreased capacity of commercial FePO₄ and synthesized EG- FePO₄ particles." And in the response there is another sentence: "We think our synthesized EG-FePO₄ particles did not exhibit their full potential yet due to the intrinsic defects as can be seen from the smaller delivered capacities." Strangely, there is no mentioning of the defects in the new (revised) versions of Supplementary of the main text.

Overall, it seems that the authors selectively show and describe in the main paper only the data that support their hypothesis, while on the other hand – although the additional data is presented in the Supplementary – in some cases the authors do not even mention the observed findings that could potentially confront their main story-line.

Let's deep more into the topics.

First of all we should differentiate between good performance LFP materials that are preferred to be used as a Li host material in Li-ion battery systems for energy storage and LFP materials used in this study for the purpose of selective Li/Na insertion. It is evident from the data that the authors have provided that the LFP materials used in the current study are low performance materials from the point of view of energy storage. Well, this is not the problem directly, since this fact is not of main importance for the present study. But on the other hand, the authors often relay their reasoning on the observations found in the Li-ion battery field. For example, they simply generalize the observation found for battery LFP materials where it was observed that for the particles of nano-size(!!!) there is general agreement that those nano-LFP do not phase-separate but rather there was observed equilibrium solid-solution mechanism of (de)-insertion. The authors in this study simply suggest that their observation of large fractions of solid-solution found for their "Synthesized-FePO₄" and "EG-LiFePO₄" samples compared to "Comm-LiFePO₄" sample is a natural consequence of the increased interfacial energy per unit volume for smaller particles. Well, this reasoning (suggested by the authors) is far from being exact and should not be presented as being straightforward! The authors base their claim of this effect of particle size based on SEM images presented in Supplementary Figure 15. Well, for me is rather hard to deduce from Supplementary Figure 15a that the particles of "Comm-LiFePO₄" sample shown exhibit "radii mainly in range from 1 μm to 4 μm" as claimed by the authors. I can observe rather large fraction of much smaller particles – in fact probably the primary particles in "Comm-LiFePO₄" sample are much smaller compared to the particles of "EG-LiFePO₄" sample. Consequently, one can suspect that the claim of the authors where they provide the relation between the particle size and amount of solid-solution fraction formed during 4C lithiation is grounded on non-proven starting assumption.

Deconvolution of XRD

Furthermore, the authors seem to avoid to show part of the XRD data and corresponding analysis. I did a brief checking of their analysis of the XRD spectra of "Li-seeded" FP materials. Please see the attachment SSF from XRD of various LFP.pdf. First I took images from Supplementary Figure 16 and I had compared the spectra of the samples of "Synthesized-FePO₄" and "EG-LiFePO₄" that were pre-lithiated with 4C for 20% of Lithiation – L(0.2)4C. For me the spectra in b) ("Synthesized-FePO₄") and c) ("EG-LiFePO₄") appear to be quite similar. The positions of the peaks are practically at the same angles. I wonder if it is really true that the sample "EG-LiFePO₄" really includes practically solely High-Li solid-solution while "Synthesized-FePO₄" also a significant portion of Low-Li solid-solution. Further on I checked the composition of the sample L(0.2)4C that is presented in Figure 2d (in the main text). I read out the values of fractions of the individual solid-solution "phases" (Slide 2 in the pdf) and calculated back the corresponding Li contents in the "phases" and obtained the sum of about 18.4 %. Further on I added the Li content of LFP phase from the estimated fraction (Slide 3 in the pdf) of roughly 10%. Thus the obtained total weighted sum of Li (18.4 % + 10%) gives estimation of about 28%.

The latter observation automatically raises many questions. First of all, why the authors this not include this verification into the Supplementary Table 5 in SI? Furthermore, already presented result for weighted sum of Li of test at L(0.3)4C shows slight deviation from the expected composition 0.3. But this is not yet drastic. While on the other hand the verification of the deconvolution of L(0.2)4C shows large deviation and I wonder how is it for the case of test at L(0.1)4C – again not provided by the authors.

But this is not yet all. Even more questionable are the results of the deconvolution for the case of the sample "EG-LiFePO₄" at L(0.2)4C pre-lithiation test shown in Supplementary Figure 16c. The authors claim that during this 20% pre-lithiation "seeding" step of "EG-LiFePO₄" material they obtain total solid-solution fraction of about 43 or 44%. In the main

text they further note: "...most of the SS phases generated in the EG-FePO₄ electrode are high-Li SS (0.42 ± 0.013), with low-Li SS being only $\sim 3\%$ of the total SS phases (Supplementary Figure 16b, Supplementary Figure 16c, and Supplementary Figure 17a)." Thus one can understand from this authors claim that the insertion of 20% of Li (relative to the practical C/10 capacity of 126 mAh/g) into the particles of "EG-LiFePO₄" will lead to formation of 43% - 44% of High-Li solid solution phase. Thus, this would mean that (in average) the total content of Li in these solid solution phases should be:

Li content(solid-solution phases) larger than $0.5 \times (43\% - 44\%) = 22\%$

Then additional contribution of Li from LFP has to be added – it can be estimated to be in the order of 10-15%. Together yielding:

Total (weighted sum of) Li larger than: (32 - 37)%

And finally, if then we consider the fact, that the "EG-LiFePO₄" material exhibited only 126 mAh/g C/10 capacity (Supplementary Figure 19) we might calculate that the actual stoichiometry produced during the "seeding" step was $(126/170) \times 20\% =$ approximately 15%. With these facts in mind we come up to the following observation (please see the Slide 4 in the attached pdf file):

Insertion of about 15% of Li into FP host lattice in some strange way produces in the case of "EG-LiFePO₄" material more than (32 - 37)% of Li being found in the structure! Well this is very serious violation of the law of conservation of mass!!!

The main aspects that need to be changed in the paper:

- 1) Please do provide better quality SEM images of all three LFP materials with actual clear determination/estimation of the particles (crystallite) dimensions (thickness, and lateral dimensions). For the case of the "Comm-LiFePO₄" sample there should be clearly seen secondary and primary particles!
- 2) Please provide determination of total (weighted sum of) Li for the experiments discussed above: L(0.1)₄C and L(0.2)₄C for "Synthesized-FePO₄" and L(0.2)₄C for "EG-LiFePO₄". Please to provide explanation for the observed deviation of the result from the expected value!
- 3) The authors should comment the low observed practical (de)insertion capacity of LFP materials – particularly for the case of "EG-LiFePO₄". The authors are encouraged that they discuss that the low observed capacity could be due to e.g highly defective structure of the "EG-LiFePO₄" material (the text is actually already provided in their answer to the reviewers) and maybe even some additional reasons for observed low capacity should be addressed (e.g. large LFP crystallite size etc.).
- 4) The authors should seriously refine the abstract and conclusions and clearly state that in addition to the proposed high Li solid solution phase(s) being the origin of the enhanced Li/Na selectivity during extraction of ions from the brine solution there other important aspects to be considered – like defect level of the FP host structure.
- 5) The authors should clearly state in the main text that the "Synthesized-FePO₄" and "EG-LiFePO₄" materials are not battery-grade materials – and that the observed high fractions of solid solutions after high-current pre-lithiation are specific property of those materials not shared with common battery-grade materials.

Suggestion

I suggest that the authors do serious re-work of the organization of the data in the paper and that they provide the additional explanations for the observations presented in Supplementary (as proposed above in the points 1-5). In addition to that the authors should provide clear answers to the exposed questions (especially the ones initiated in the pdf attachment). If they do not agree with this proposal, I recommend rejection.

Taken from: Supplementary Figure 16

The spectra in b) and c) appear to be quite similar. The positions of the peaks are practically at the same angles.

Solid-solution composition of $L(0.2)_{4C}$ for "Synthesized-FePO4"

Figure 2d

Li0.750: $0.58/2 = 0.029$
 $0.029 \cdot 0.75 = 0.0217$

Li0.625: $1.34/2 = 0.067$
 $0.067 \cdot 0.625 = 0.0419$

Li0.500: $3.44/2 = 0.172$
 $0.172 \cdot 0.5 = 0.086$

Li0.250: $2.45/2 = 0.1225$
 $0.1225 \cdot 0.25 = 0.0306$

Li0.125: $0.55/2 = 0.0275$
 $0.0275 \cdot 0.125 = 0.00344$

Summ SSF = 0.418

Weighted sum (Li) = 0.184

Solid-solution composition of $L(0.2)_{4C}$ for "Synthesized-FePO₄"

Taken from: Supplementary Figure 16

Estimation.
From the LFP: ~ 0.1

Taken from: Figure 2

From the solid-solution:
Weighted sum (Li) = 0.184

Total (weighted sum of) Li: ~ 0.18 + 0.1 = 0.28

Total (weighted sum of) of Li for the L(0.2)_{4C} test using “EG-LiFePO₄” material

Taken from: Supplementary Figure 16c

Estimation of Li content in the LFP phase: 0.1 – 0.15

Total (weighted sum of) Li: > (10-15)% + 22%

High-Li solid-solution phases prevailing:

Li from the solid-solution phases: > 0.5*(43-44)% = ~22%

The “seeding” step:
(126/170)*20% ~15% Li

Total (weighted sum of) Li: > (32-37)% !!!

Response to reviewers' comments:

We would like to thank the reviewers for their valuable comments for improving our work. Please see the point-by-point response below. We have revised the manuscript based on the latest suggestions. The major changes in our revised manuscript have been marked in **Red**.

Reviewer #1:

The authors answered correctly to all my questions and comments. In my opinion the manuscript is ready for its publication. I would like to thank the authors for their effort answering my comments and improving the article and congrats them for the excellent work done.

We thank the reviewer for the highly positive feedback on our work.

Reviewer #2:

The authors have produced an interesting, original piece of work, with potential for practical applications. The content is described well and after addressing the comments from reviewers, the manuscript is very sound and thorough. I recommend publication.

We thank the reviewer for the highly positive comments to our manuscript and responses.

Reviewer #3:

The authors have in their response to the questions and concerns of the referees provided quite large amount of additional data and information. Certainly I can agree that the authors have put additional efforts in defending their work and reasoning.

On the other hand the authors do not provide convincing enough data to support their main hypothesis. Namely, they propose that the high Li solid solution phase(s) are the main reason behind the Li/Na insertion selectivity. I think this is somewhat overemphasized statement. First of all, the authors themselves clearly present data that show that Li intercalation into FePO₄ host structure is thermodynamically (e.g. Supplementary Figure 14) as well as kinetically (e.g. Supplementary Figure 23) favored over Na intercalation.

We thank the reviewer for bringing up this point. The thermodynamics and kinetics of FePO₄ host indeed prefers Li over Na, which has been shown by us and many other works (See **Supplementary Table 7 for 5 publications to date**). Especially, in our previous work¹, we plotted the voltage difference between Li and Na intercalation of FePO₄ (mainly thermodynamic preference) as an indicator for its feasibility for Li extraction at different Li/Na ratios. We briefly mentioned this point in the previous version of the manuscript in the first paragraph of the introduction as "...One-dimensional (1D) olivine FePO₄ is a promising host material owing to its appropriate working potentials, framework stability, thermodynamic Li intercalation preference, and lower Li migration barrier...". To make this point clearer and avoid confusion, we added more information in the main text as follows:

"One-dimensional (1D) olivine FePO₄ is a promising host material owing to its appropriate working potentials, framework stability, thermodynamic Li intercalation preference, and lower Li migration barrier^{1,4,7,12,13}. Specifically, the calculated lithiation voltage of olivine FePO₄ host is around 3.45 V vs. Li/Li⁺ (= 0.213 V vs. Ag/AgCl), which is higher than the sodiation voltage (3.08 V vs. Na/Na⁺ = 0.173 V vs. Ag/AgCl)¹³. The migration barrier of Li ion is only 0.17 eV, smaller than that of Na ion (0.29 eV)¹³. Even with the intrinsic material favorability to Li, during electrochemical Li extraction at low Li to Na ratio, co-intercalation occurs with Na as the main competitor^{7,9,13,14}."

As can be seen from all the published work, Na will start to compete with Li when the Li/Na ratio is below ~ 1/100s (the exact number depends on particle and running conditions). But for many unconventional sources that contain large amount of Li such as geothermal brines and oilfield flow back waters, the Li/Na ratio can be in the range of 1/1000 or below²⁻⁴. Therefore, there is still strong need to keep improving the Li to Na selectivity to enable electrochemical intercalation for Li extraction from a broader range of sources. Keep improving Li selectivity will benefit both the FePO₄ host structural stability and Faradaic Efficiency which are related to cost and feasibility of application. However, there is currently lack of knowledge in the field to guide the direction of improvement to increase the Li selectivity of the FePO₄ host material.

Therefore, the Li seeding method reported in this work to create high-Li solid solution phases is one step

further to improve the Li to Na selectivity beyond the intrinsic thermodynamic and kinetic preference. We named the strategy as a promotion to Li competitiveness in the title.

Before addressing the detailed questions from the reviewer, we would like to summarize the key points in this work that we think has fulfilled the knowledge gap:

1. Our first contribution is to reveal how Li and Na compete during co-intercalation. As mentioned in the later sentence in the 1st paragraph of the introduction, how FePO₄ response to Li and Na co-intercalation is unknown (Page 3 line 9: “Despite intriguing proof of concept, the FePO₄ host structure response upon Li and Na competitive co-intercalation remains unknown. The intercalation pathways and storage sites are critical in determining the energy barriers for both Li and Na intercalation (including formation enthalpy, migration barrier, nucleation barrier, and interfacial energy), affecting selectivity.”). We show that Li and Na tend to phase separate in olivine FePO₄ hosts with both computational and experimental evidence (**Figure 1 in the manuscript**). Scanning electron nanodiffraction, for the first time, was used for co-intercalation phase detection, and it showed Li and Na domains at a single particle level within 10 nm resolution (**Figure 1c in the manuscript**). This is a key piece of knowledge. It serves as both inspiration and guide for us to design the Li seeding method. This knowledge by itself is valuable and it can also serve as inspiration for others to design different strategies to promote the Li competitiveness.
2. We proposed a strategy of using Li seeding to pre-create Li phases to improve Li selectivity and we demonstrated that this strategy is effective. We analyzed the Li phases after Li seeding and discovered a strong correlation of pre-created high-Li solid solution phases to Li selectivity improvement. With seeding, we achieved a ~ 3.8-fold increase of Li to Na selectivity, with Li accounting for ~ 94% of the occupied total host sites. The Li occupancy percentage is an important factor for Li extraction since it is related to both Faradaic Efficiency and structure stability (future studies is needed to draw a quantitative relationship between stability and FE%, which we think is beyond the study of this work). The Li selectivity promotion by high-Li solid solution phases is also supported by our DFT calculations, which show that high-Li solid solution phases have larger energy differences between Li and Na intercalation favoring Li (**Figure 4d in the manuscript**).

We thank the reviewer for the question about the effect of material characteristic of FePO₄ (including size, morphology, quality, and so on) on Li selectivity during the first round of revision. The high-Li solid solution correlation to Li selectivity is indeed only demonstrated using one type of platelet particle in our first manuscript. We see the material forms questions as two-fold now: 1. Whether the high-Li solid solution correlation with Li selectivity is general for all particles. 2. Can the Li selectivity be improved even more by dialing the material forms. In the first round of revision, we put more attention to the 2nd question “Can the Li selectivity be improved even more from dialing the material forms” and we mainly focused on commenting the improved selectivity from EG-FePO₄ particles. We guess the reviewer’s questions are more to the 1st aspect of “Whether the high-Li solid solution correlation with Li selectivity is general for all particles.”. Below we will answer and discuss the effect of material characteristics in the reply to the five questions from the reviewer.

References:

1. C. Liu, Y. B. Li, D. C. Lin, P. C. Hsu, B. F. Liu, G. B. Yan, T. Wu, Y. Cui, S. Chu, *Joule* 2020, 4, 1459.
2. Warren, I. *Techno-Economic Analysis of Lithium Extraction from Geothermal Brines*. Golden, CO: National Renewable Energy Laboratory. NREL/TP-5700-79178. (National Renewable Energy Laboratory, 2021).
3. *Technical Development Document for the Effluent Limitations Guidelines and Standards for the Oil and Gas Extraction Point Source Category*. (United States Environmental Protection Agency, 2016).
4. A. Kumar, H. Fukuda, T. A. Hatton, J. H. Lienhard, *ACS Energy Letters* 2019, 4, 1471.

Furthermore, in the Supplementary Figure 19 there are shown C/10 galvanostatic charge/discharge curves in water system. Surprisingly there is no explanation provided for the observed very low performance of the "EG- FePO₄ particles". At the same time the authors in their response to the referees show the same data (Figure R13) and there is a sentence: "The existence of defects may be the reason for the decreased capacity of commercial FePO₄ and synthesized EG-FePO₄ particles." And in the response there is another sentence: "We think our synthesized EG-FePO₄ particles did not exhibit their full potential yet due to the intrinsic defects as can be seen from the smaller delivered capacities." Strangely, there is no mentioning of the defects in the new (revised) versions of Supplementary of the main text.

Overall, it seems that the authors selectively show and describe in the main paper only the data that support their hypothesis, while on the other hand – although the additional data is presented in the Supplementary – in some cases the authors do not even mention the observed findings that could potentially confront their main story-line.

Let's deep more into the topics.

First of all we should differentiate between good performance LFP materials that are preferred to be used as a Li host material in Li-ion battery systems for energy storage and LFP materials used in this study for the purpose of selective Li/Na insertion. It is evident from the data that the authors have provided that the LFP materials used in the current study are low performance materials from the point of view of energy storage. Well, this is not the problem directly, since this fact is not of main importance for the present study. But on the other hand, the authors often relay their reasoning on the observations found in the Li-ion battery field. For example, they simply generalize the observation found for battery LFP materials where it was observed that for the particles of nano-size(!!!) there is general agreement that those nano-LFP do not phase-separate but rather there was observed equilibrium solid-solution mechanism of (de)-insertion. The authors in this study simply suggest that their observation of large fractions of solid-solution found for their "Synthesized-FePO₄" and "EG-LiFePO₄" samples compared to "Comm-LiFePO₄" sample is a natural consequence of the increased interfacial energy per unit volume for smaller particles. Well, this reasoning (suggested by the authors) is far from being exact and should not be presented as being straightforward!

The authors base their claim of this effect of particle size based on SEM images presented in Supplementary Figure 15. Well, for me is rather hard to deduce from Supplementary Figure 15a that the particles of "Comm-LiFePO₄" sample shown exhibit "radii mainly in range from 1 μm to 4 μm" as claimed by the authors. I can observe rather large fraction of much smaller particles – in fact probably the primary particles

in “Comm-LiFePO₄” sample are much smaller compared to the particles of “EG-LiFePO₄” sample. Consequently, one can suspect that the claim of the authors where they provide the relation between the particle size and amount of solid-solution fraction formed during 4C lithiation is grounded on non-proven starting assumption.

Deconvolution of XRD

Furthermore, the authors seem to avoid to show part of the XRD data and corresponding analysis. I did a brief checking of their analysis of the XRD spectra of “Li-seeded” FP materials. Please see the attachment SSF from XRD of various LFP.pdf. First I took images from Supplementary Figure 16 and I had compared the spectra of the samples of “Synthesized-FePO₄” and “EG-LiFePO₄” that were pre-lithiated with 4C for 20% of Lithiation – L(0.2)4C. For me the spectra in b) (“Synthesized-FePO₄”) and c) (“EG-LiFePO₄”) appear to be quite similar. The positions of the peaks are practically at the same angles. I wonder if it is really true that the sample “EG-LiFePO₄” really includes practically solely High-Li solid-solution while “Synthesized-FePO₄” also a significant portion of Low-Li solid-solution. Further on I checked the composition of the sample L(0.2)4C that is presented in Figure 2d (in the main text). I read out the values of fractions of the individual solid-solution "phases" (Slide 2 in the pdf) and calculated back the corresponding Li contents in the "phases" and obtained the sum of about 18.4 %. Further on I added the Li content of LFP phase from the estimated fraction (Slide 3 in the pdf) of roughly 10%. Thus the obtained total weighted sum of Li (18.4 % + 10%) gives estimation of about 28%.

The latter observation automatically raises many questions. First of all, why the authors this not include this verification into the Supplementary Table 5 in SI? Furthermore, already presented result for weighted sum of Li of test at L(0.3)4C shows slight deviation from the expected composition 0.3. But this is not yet drastic. While on the other hand the verification of the deconvolution of L(0.2)4C shows large deviation and I wonder how is it for the case of test at L(0.1)4C – again not provided by the authors.

But this is not yet all. Even more questionable are the results of the deconvolution for the case of the sample “EG-LiFePO₄” at L(0.2)4C pre-lithiation test shown in Supplementary Figure 16c. The authors claim that during this 20% pre-lithiation “seeding” step of “EG-LiFePO₄” material they obtain total solid-solution fraction of about 43 or 44%. In the main text they further note: "...most of the SS phases generated in the EG-FePO₄ electrode are high-Li SS (0.42 ± 0.013), with low-Li SS being only ~ 3% of the total SS phases (Supplementary Figure 16b, Supplementary Figure 16c, and Supplementary Figure 17a)." Thus one can understand from this authors claim that the insertion of 20% of Li (relative to the practical C/10 capacity of 126 mAh/g) into the particles of “EG-LiFePO₄” will lead to formation of 43% - 44% of High-Li solid solution phase. Thus, this would mean that (in average) the total content of Li in these solid solution phases should be:

Li content(solid-solution phases) larger than $0.5 \cdot (43\% - 44\%) = 22\%$

Then additional contribution of Li from LFP has to be added – it can be estimated to be in the order of 10-15%. Together yielding:

Total (weighted sum of) Li larger than: (32 - 37)%

And finally, if then we consider the fact, that the “EG-LiFePO₄” material exhibited only 126 mAh/g C/10 capacity (Supplementary Figure 19) we might calculate that the actual stoichiometry produced during the “seeding” step was $(126/170)*20\%$ = approximately 15%. With these facts in mind we come up to the following observation (please see the Slide 4 in the attached pdf file):

Insertion of about 15% of Li into FP host lattice in some strange way produces in the case of “EG-LiFePO₄” material more than (32 - 37)% of Li being found in the structure! Well this is very serious violation of the law of conservation of mass!!!

The main aspects that need to be changed in the paper:

1) Please do provide better quality SEM images of all three LFP materials with actual clear determination/estimation of the particles (crystallite) dimensions (thickness, and lateral dimensions). For the case of the “Comm-LiFePO₄” sample there should be clearly seen secondary and primary particles!

Thanks for the reviewer’s question. We updated the SEM images for all three particles and included the corresponding particle dimension distribution in **Figure R1, Figure R2, and Figure R3 (as Supplementary Figure 19, 20 and 18 respectively)**. Specifically, for the Synthesized-LiFePO₄ particles, the average lateral dimension along the long axis is $\sim 1.93 \mu\text{m}$, with the [010] channel length (thickness) of $\sim 270 \text{ nm}$. For the EG-LiFePO₄ particles, the average lateral dimension along the long axis is $\sim 0.50 \mu\text{m}$, with the [010] channel length (thickness) of $\sim 97 \text{ nm}$, both of which are the smallest among the three particles. The Comm-LiFePO₄ particles were bought from *MTI Corporation* (Item Number: Lib-LFPOS21). As shown in **Figure R3c, R3d and R3e**, the average dimension of the primary particles we acquired is $\sim 430 \text{ nm}$. Besides, the dimension of the secondary particles is $\sim 2.93 \mu\text{m}$ (**Figure R3b, R3f and R3g**).

We updated **Supplementary Figure 18, 19, and 20** and modified the dimension description in the revised manuscript:

“We investigate the effect of particle characteristics on the formation of solid solutions and Li competitiveness. We studied two other particles with various forms. One is commercial LiFePO₄ particles (Comm-LiFePO₄), which were bought from *MTI Corporation* (Item Number: Lib-LFPOS21). As shown in Supplementary Figure 18, the average dimension of the primary ellipsoidal shape particles is $\sim 430 \text{ nm}$. Besides, the dimension of the secondary particles is $\sim 2.93 \mu\text{m}$. As a comparison, Supplementary Figure 19 shows the original particles (Synthesized-LiFePO₄) which we used for all the experiments unless mentioned. The lateral dimension along the long axis is $\sim 1.93 \mu\text{m}$, with the [010] channel length $\sim 270 \text{ nm}$. Besides, we synthesized a new platelet-like particle (EG-LiFePO₄) as shown in Supplementary Figure 20 with smaller lateral ($\sim 97 \text{ nm}$) and thickness dimensions ($\sim 0.50 \mu\text{m}$). All the particles have dimensions for the diffusion direction below $1 \mu\text{m}$. Both Comm-LiFePO₄ particles and EG-LiFePO₄ particles follow the same chemical Li extraction process to prepare empty FePO₄ hosts (See Methods for more details).”

Based on the primary particle size, the Comm-LiFePO₄ particles are not necessarily larger than the Synthesized-LiFePO₄ particles. We agree with the reviewer that when the morphology (or category of shape) of the FePO₄ is different and the defect level is different, size cannot be used as a single measurement for

particle comparisons. Also, size alone can not explain the selectivity difference between Synthesized-LiFePO₄ particles and Comm-LiFePO₄ particles. More details regarding the comparison is provided in Q3. In the revised manuscript, we added the discussion of particle characteristics in the abstract, in the last part of the main text (“Effects of particle characteristics on solid solution seeding and Li selectivity”), as well as in the conclusion.

Figure R1 SEM images of Synthesized-LiFePO₄ particles (a, c) with corresponding particle dimension distribution along the long axis (b) or thickness (d). (Only the isolated and fully exposed particles were counted; the thin white lines in the SEM images denoted the measured length.)

Figure R2 SEM images of EG-LiFePO₄ particles (a, c) with corresponding particle dimension distribution along the long axis (b) or thickness (d). (Only the isolated and fully exposed particles were counted; the thin white lines in the SEM images denoted the measured length.)

Figure R3 SEM images for Comm-LiFePO₄ particles. (a) Comm-LiFePO₄ particles dropped on carbon tape, (b) Zoom-in SEM for Comm-LiFePO₄ secondary particles dispersed on Si wafer, (c) Zoom-in SEM for Comm-LiFePO₄ primary particles on Si, additional SEM images and corresponding lateral distribution summary of primary (d, e) and secondary (f, g) Comm-LiFePO₄ particles dispersed on Si wafer. (Only the isolated and fully exposed particles were counted; the thin white lines in the SEM images denoted the measured length.)

2) Please provide determination of total (weighted sum of) Li for the experiments discussed above: L(0.1)4C and L(0.2)4C for “Synthesized-FePO₄” and L(0.2)4C for “EG-LiFePO₄”. Please to provide explanation for the observed deviation of the result from the expected value!

We thank the reviewer for the question and appreciate the careful evaluations.

As mentioned previously, in this paper for the first time, we proposed a strategy to improve the Li to Na selectivity by pre-creating Li SS phases. The validation of this strategy is done by comparing Li selectivity ($\text{Li}/(\text{Li}+\text{Na})_{\text{net}}$) at different seeding percentage (different level of solid solution phase creation) with our model platelet synthesized-FePO₄. Our first conclusion is that this strategy is effective since Li selectivity increases with the seeding range. Then we want to investigate whether all solid solution phases are equally effective in promoting the Li selectivity and we did XRD deconvolution for the analysis. We provided all the deconvolution results for the reviewer’s reference in **Figure R4** and **Table R1 (also as Supplementary Figure 5 and Supplementary Table 5)**. As mentioned by the reviewer, we did see deviations in the calculated weighted sum of Li. We plotted the calculated weighted sum of Li from XRD fittings versus the electrochemically intercalated Li amount, as shown in **Figure R5 (Supplementary Figure 6 in SI)**. The relationship between the calculated weighted sum of Li and the depth of intercalation has good linearity ($R^2 = 0.994$). The deviation of the XRD fitted Li amount from electrochemical seeding amount indicates the possibility of unidentified system error. Even with error, this good linearity supports our analysis of the correlation of Li selectivity to Li solid solution phases since the error cannot be randomly affecting either the low-Li or high-Li solid solution phase fractions. Otherwise, there will not be a good linearity.

We think there could be two reasons that cause the deviations:

1. As mentioned in the original SI, in order to have a quantitative measurement, we did LeBail refinement for the whole XRD patterns. However, the standard LeBail refinement was possible only for the end phases of LFP and FP because the lattice parameters of all intermediate phases were completely interchangeable¹. With the help of Vegard’s law, we could achieve a calculation of Li from XRD patterns. The intermediate phases do not correspond to a single phase of a specific concentration but all Li concentrations (x) in LiFePO₄, $0 < x < 1$ during the transition². In other words, the more accurate deconvolution of the XRD intensity band requires an infinite number of phases, which is assumed to be impossible and impracticable, which could also lead to overfitting issues. Since only nine different phases of state of charge were chosen to deconvolute the XRD patterns, this simplification assigned the other intermediate phases to the nine chosen ones, which could cause deviations of the calculated weighted sum of Li.
2. In response to the reviewer’s questions, we find that the intensity contributions from the PVDF, super P, and carbon cloth substrate could also introduce deviations by raising the background intensity, especially under the low depth of intercalation. We then tried a much flatter glassy carbon with better carbon quality as the substrate. The new glassy carbon substrate decreases the deviations caused by porous structures of carbon cloth. The fitting using glassy carbon substrates were introduced in detail below.

Synthesized-FePO₄

We did a new series of seeding using the new flat glassy carbon substrate and we did phase deconvolution following the same method. As shown in **Figure R6** and **Table R2** (also as **Supplementary Figure 7** and **Supplementary Table 6**), we achieved a better agreement between the calculated weighted sum of Li from XRD and the electrochemical seeded Li amount (linear relationship is also good as shown in **Figure R7a**). We also confirmed that seeding on the new substrate gives similar $\text{Li}/(\text{Li}+\text{Na})_{\text{net}}$ ratio (0.73 ± 0.01) by measuring 3 times the Li selectivity for $\text{L}(0.2)_{4\text{C}}$ conditions. The $\text{Li}/(\text{Li}+\text{Na})_{\text{net}}$ on old carbon cloth substrate is 0.72 ± 0.03 . Importantly, with the new series of seeding on glassy carbon substrates, we confirmed again the monotonically increasing trend of high-Li SS phases with increased seeding range as shown in **Figure R7b** (also as **Supplementary Figure 8b**). Moreover, the low-Li SS phases still did not correlate with the increasing Li seeding range. Therefore, our conclusion that high-Li solid solution phases can promote the Li selectivity is confirmed again for the platelet Synthesized- FePO_4 . We added these new data and the discussion of “*Potential reasons for the deviations of calculated weighted sum of Li from fitting*” in the SI.

Figure R4 Deconvoluted XRD patterns of (a) $\text{L}(0.1)_{4\text{C}}$, (b) $\text{L}(0.2)_{4\text{C}}$ (c) $\text{L}(0.3)_{4\text{C}}$, (d) $\text{L}(0.4)_{4\text{C}}$ and (e) $\text{L}(0.5)_{4\text{C}}$ for Synthesized- FePO_4 particles on carbon cloth substrate from one representative sample.

Table R1 Fitted phase fractions of Figure R6 with the calculated weighted sum of Li.

	LFP	L0.875FP	L0.75FP	L0.625FP	L0.5FP	L0.375FP	L0.25FP	L0.125FP	FP	Intercalated Li	Weighted sum of Li
L(0.1) _{4C}	0.07562	0	0.01964	0.04159	0.16232	0	0.14505	0	0.55578	0.1	0.23377
L(0.2) _{4C}	0.11218	0.00506	0.03933	0.06099	0.17481	0	0.11613	0.03272	0.45878	0.2	0.30475
L(0.3) _{4C}	0.14591	0.02684	0.01063	0.08277	0.18803	0	0.13107	0	0.41474	0.3	0.35589
L(0.4) _{4C}	0.16798	0.04583	0	0.12913	0.19123	0	0.10210	0.02974	0.33398	0.4	0.41365
L(0.5) _{4C}	0.19104	0.04145	0.07255	0.25751	0.02989	0.05795	0.04552	0.07888	0.2252	0.5	0.50058

Figure R5 Calculated weighted sum of Li from XRD fittings versus the electrochemically intercalated Li amount with the use of Synthesized-FePO₄ particles on carbon cloth. (Error bars representing the standard deviation of three replicate measurements.)

Figure R6 Deconvoluted XRD patterns of (a) L(0.1)_{4C}, (b) L(0.2)_{4C} (c) L(0.3)_{4C}, and (d) L(0.4)_{4C} for Synthesized-FePO₄ particles on glassy carbon substrate from one representative sample.

Table R2 Fitted phase fractions of Figure R6 with the calculated weighted sum of Li.

	LFP	L0.875FP	L0.75FP	L0.625FP	L0.5FP	L0.375FP	L0.25FP	L0.125FP	FP	Low-Li SS	High-Li SS	Intercalated Li	Weighted sum of Li
L(0.1) _{4C}	0.01511	0.00562	0.08468	0	0	0	0.13104	0	0.76355	0.13104	0.0903	0.1	0.1163
L(0.2) _{4C}	0.06107	0.01466	0.08725	0.05607	0	0	0.14645	0	0.63449	0.14645	0.15798	0.2	0.21099
L(0.3) _{4C}	0.06909	0.05342	0.05712	0.12553	0	0.04728	0.11082	0.04485	0.49188	0.20295	0.23607	0.3	0.28817
L(0.4) _{4C}	0.08447	0.11453	0.02793	0.14864	0.04111	0.04378	0.12912	0.01258	0.39784	0.18548	0.33221	0.4	0.36936

Figure R7 (a) Calculated weighted sum of Li from XRD fittings versus the electrochemically intercalated Li amount with the use of Synthesized-FePO₄ particles on glassy carbon; (b) High-Li SS fractions (Li_xFePO_4 , $x = 0.500/0.625/0.750/0.875$) and low-Li SS fractions (Li_xFePO_4 , $x =$

0.125/0.250/0.375) under the same seeding rate 4C (588 mA/g) with different seeding ranges L(0.1/0.2/0.3/0.4)_{4C} collected on glassy carbon. (Error bars representing the standard deviation of three replicate measurements.)

EG-FePO₄ particles

To explain the larger deviations observed for the EG-FePO₄ particles, first, we changed the old carbon cloth substrate to the new glassy carbon substrate. Since this requires new slurry, we did a new batch of synthesis of the EG-LiFePO₄ particles. We would like to point out that this new batch of particles have similar morphologies comparing to the previous batch of EG-LiFePO₄ particles (as shown in **Figure R2**) and similar electrochemical capacity (125 mAh/g under 0.1C, updated **Supplementary Figure 30** in the SI). Even with the glassy carbon substrate, the calculated weighted sum of Li from XRD fittings still exceeds the amount of Li from seeding (**Figure R8a and Table R3**). We then checked the quality of the chemically Li-extracted EG-FePO₄ hosts. We used aqua regia solution to fully dissolve the assembled electrodes to measure the remaining Li content in the hosts. As shown in **Table R4**, there is ~ 23.5 at. % Li remained in the EG-FePO₄ host. This showed that the chemical route initially used cannot realize complete Li⁺ removal, which indicates the existence of defects in the synthesized EG-FePO₄ particles. We do agree with the reviewer that we should emphasize more the important roles of defects for determining solid solution formation and EG-FePO₄ is not a high-quality battery cathode. An interesting thing is that these trapped Li species are in the form of solid solution phases, as evidenced by the strong solid solution intensity band and weak LiFePO₄ peak in the XRD patterns collected on glassy carbon (**Figure R8a**). These trapped Li remains in the host after a round of intercalation and deintercalation (after Li recovery) as confirmed by the XRD patterns. The fitted phase fractions after recovery are similar to that of the raw particles (**Figure R8b and Table R3**). This indicates that the defect-induced trapped Li did not move during the (de)intercalation. The trapped Li amount from ICP measurement is close to that from XRD fitting. Also, if we add up the ~ 23.5% pre-trapped Li and ~ 15% seeded Li (L(0.2)_{4C} normalized by theoretical capacity of 170 mAh/g), we will get ~ 38.5% Li, which now have a better match with the calculated weighted sum of Li from the XRD pattern (~ 34.9%) (**Table R3**). Therefore, we think the mismatch of Li amount from XRD fitting to electrochemical seeding is mainly due to the trapped Li in the host. This analysis shows again that the EG-FePO₄ has a high defect level. We have added these data and explanation as supporting data in **Supplementary Figure 23, Supplementary Table 10, and Supplementary Table 11** and added discussion of the EG-FePO₄ particles in the last part of the main text (“Effects of particle characteristics on solid solution seeding and Li selectivity”).

Figure R8 Deconvoluted XRD patterns of EG-FePO₄ on glassy carbon (a) raw electrodes, (b) after Li recovery, and (c) After 4C 20% Li seeding, L(0.2)_{4C}.

Table R3 Fitted phase fractions of Figure R8 with the calculated weighted sum of Li.

EG-FePO ₄	LFP	L0.875FP	L0.75FP	L0.625FP	L0.5FP	L0.375FP	L0.25FP	L0.125FP	FP	Intercalated Li	Weighted sum of Li
Raw	0.08255	0.10463	0.01853	0	0	0.11015	0.10872	0.245	0.33041	0	0.28711
After recovery	0.08856	0.1	0.01685	0	0	0.10857	0.12296	0.23901	0.32405	0	0.29003
L(0.2) _{4C}	0.1117	0.09476	0.00316	0.05839	0.03014	0.13481	0.09826	0.2001	0.26867	0.2	0.34868

Table R4 Li and Fe content measured by ICP-MS. The chemically Li-extracted EG-FePO₄ hosts were washed with distilled water for 3-5 times, then dissolved with aqua regia solution for three days to ensure full dissolution. The resulting solution was diluted with 3% HNO₃ for ICP-MS measurement.

EG-FePO ₄	Label	Li (ppb)	Fe (ppb)	Li/Fe (at. %)	Li/Fe _{ave} (at. %)
Raw	S1	7.405	246.620	24.0%	23.5%
	S2	7.843	267.724	23.4%	
	S3	19.095	662.835	23.0%	

Comparison among three types of particles:

For the Comm-LiFePO₄, we also retake the XRD on the glassy carbon substrate. The comparison of the XRD patterns of the three seeded L(0.2)_{4C} particles are shown in **Figure R9 (Supplementary Figure 21 in the SI)**. The breakdown of low-Li and high-Li SS phase fractions are shown in **Figure R10 (Supplementary Figure 22 in the SI)**. The new fitting data shows that the Li selectivity follows the trend of the high-Li

solid solution phases after seeding. However, we agree that morphologies, size, and defect levels have to be considered together when comparing different particles. The relationship between high-Li SS fraction to Li selectivity across different particles might not be in the same linear relationship, which definitely deserves further investigation. In the revised manuscript, we added the discussion of particle characteristics in the abstract, in the last part of the main text (“Effects of particle characteristics on solid solution seeding and Li selectivity”), as well as in the conclusion.

Figure R9 Example XRD patterns of (a) Comm-FePO₄, (b) Synthesized-FePO₄, and (c) EG-FePO₄ electrodes after 20% of Li seeding under 4C collected on glassy carbon. For Comm-FePO₄, 135 mA/g equals a rate of 1C; for Synthesized-FePO₄, 147 mA/g equals a rate of 1C; for EG-FePO₄, 125 mA/g equals a rate of 1C (Dotted lines: obtained patterns; Solid lines: fitted patterns).

Figure R10 (a) Total SS, Low-Li SS, and High-Li SS fractions for the three electrodes under the same seeding rate 4C (540 mA/g for Comm-FePO₄; 588 mA/g for Synthesized-FePO₄; 500 mA/g for EG-FePO₄) with the 20% seeding range L(0.2)_{4C}. (b) Recovered Li/(Li+Na)_{net} of the three electrodes until 70% of the

total capacity used under 0.1C, with either initial 20% of Li seeding under 4C (L(0.2)_{4C}-LN(0.7)_{0.1C}) or without any initial seeding process (L(0)-LN(0.7)_{0.1C}). (Error bars represent the standard deviation of three replicate sample measurements.)

References:

1. M. Hess, T. Sasaki, C. Villevieille, P. Novák, Nature Communications 2015, 6, 8169.
2. Y. Orikasa, T. Maeda, Y. Koyama, H. Murayama, K. Fukuda, H. Tanida, H. Arai, E. Matsubara, Y. Uchimoto, Z. Ogumi, Journal of the American Chemical Society 2013, 135, 5497.

3) The authors should comment the low observed practical (de)insertion capacity of LFP materials – particularly for the case of “EG-LiFePO₄”. The authors are encouraged that they discuss that the low observed capacity could be due to e.g highly defective structure of the “EG-LiFePO₄” material (the text is actually already partially provided in their answer to the reviewers) and maybe even some additional reasons for observed low capacity should be addressed (e.g. large LFP crystallite size etc.).

We thank the reviewer for the suggestion. The low capacity of EG-LiFePO₄ indeed has a high defect level. There is still ~ 23.5 at. % Li remained in the EG-FePO₄ host after chemical extraction. The chemical route initially used cannot realize complete Li⁺ removal, which indicates the existence of defects in the synthesized EG-FePO₄ particles, even though we already tried annealing to decrease the cation disorder. Furthermore, the ~ 23.5% pre-trapped Li is also consistent with the 125 mAh/g delivered capacity, which is around 74% of the theoretical capacity of LiFePO₄. Overall, at this stage, we think the main reason for the observed low capacity is due to defects in the hosts.

Here, we do agree with the reviewer that we should have clarified more thoroughly in our original manuscript about the effects of defects in the EG-FePO₄ particles besides size differences. Since both size and defects level would affect the lithium transport as well as the solid solution evolution. At this stage, we agreed with the reviewer that we can not attribute solely to the decreased channel length of EG-FePO₄ particles since it also has defect-induced trapped Li. The majority of trapped Li is in the form of solid solution phases, which benefits our Li selectivity at a penalty of decreased capacity.

We have revised the manuscript to reflect the new discussion on the effect of particle characteristics on Li solid solution formation and Li selectivity as below on page 13 line 9:

“Effects of particle characteristics on solid solution seeding and Li selectivity”

The ion insertion reaction of Li_xFePO₄ with different particle characteristics has been studied intensively. And the formation of solid solution in olivine FePO₄ can be a very complicated process that is affected by many factors, such as the temperature^{18,26}, particle size/morphology^{32,38-40}, applied current densities^{25,28-30,51}, and defects^{21,41}. The earliest reports have shown that the (de)intercalation of Li⁺ goes through a phase separation reaction into Li-rich and Li-poor phases at room temperature^{15,52}. At elevated temperature (> 400 °C), single-phase reaction was observed in the whole composition range (0 < x < 1 in Li_xFePO₄)²⁶. In

addition, the miscibility gap has been found to reduce with the reduction of particle size, even vanish when the particles reach the critical nano-size region ($d_c \leq 22$ nm)^{32,38,39}. Moreover, both computational^{25,30} and experimental^{28,29} results have demonstrated that, at elevated (de)lithiation rates, phase separation is suppressed and replaced with a solid solution pathway. Particles with different morphologies may also have various response to the same global current density, even with similar particle size. It is demonstrated that platelet particles have a much lower exchange current density than ellipsoidal particles, which would increase the active particle population and promote uniform solid-solution domains^{29,51}. Importantly, defects play a significant role in controlling the intercalation phase transformation pathway. It is found that particle size can be considered as a good but not sufficient condition to anticipate single phase solid solution formation²¹. Different amounts of non-stoichiometry and cationic mixing could lead to different phase transformation, even with the same particle size. With considerable Li/Fe disorder, solid solution formation in the whole composition range can be realized²¹.

We then investigate the effect of particle characteristics on the formation of solid solutions and Li competitiveness. We studied two other particles with various forms. One is commercial LiFePO₄ particles (Comm-LiFePO₄), which were bought from *MTI Corporation* (Item Number: Lib-LFPOS21). As shown in Supplementary Figure 18, the average dimension of the primary ellipsoidal shape particles is ~ 430 nm. Besides, the dimension of the secondary particles is ~ 2.93 μm. As a comparison, Supplementary Figure 19 shows the original particles (Synthesized-LiFePO₄) which we used for all the experiments unless mentioned. The lateral dimension along the long axis is ~ 1.93 μm, with the [010] channel length ~ 270 nm. Additionally, we synthesized a new platelet-like particle (EG-LiFePO₄) as shown in Supplementary Figure 20 with smaller lateral (~ 97 nm) and thickness dimensions (~ 0.50 μm). All the particles have dimensions for the diffusion direction below 1 μm. Both Comm-LiFePO₄ particles and EG-LiFePO₄ particles follow the same chemical Li extraction process to prepare empty FePO₄ hosts (See Methods for more details).

Comm-FePO₄, Synthesized-FePO₄, and EG-FePO₄ showed capacity of 135 mA/g, 147 mA/g, and 125 mA/g, respectively (See Supplementary Materials for capacity evaluation). It is worth mentioning that not all particles are good battery quality particles, especially EG-FePO₄. Their low delivered capacity indicates the high level of defects. Supplementary Figure 21 shows example XRD patterns collected on flat glassy carbon for Comm-FePO₄, Synthesized-FePO₄, and EG-FePO₄ electrodes with 20% Li seeding under 4C. We seeded 20% Li for each type of particles and summarized the solid solution fractions in Supplementary Figure 22a. In the case of Comm-FePO₄ ellipsoidal shape particles, the total SS fraction generated is the least (Total SS = 0.234 ± 0.002) with the intercalated Li⁺ ions mainly formed fully occupied LiFePO₄ phase, as indicated by the pronounced LiFePO₄ (020) peak in Supplementary Figure 21a. And the calculated weighted sum of Li is around 17% from the XRD pattern (Supplementary Figure 22a), close to the 20% Li seeding. The ellipsoidal shape and low defect level of Comm-FePO₄ particles could increase the miscibility gap, which can suppress solid solution formation^{29,32,38,39,51}. Meanwhile, we achieved the lowest selectivity with Comm-FePO₄ particles ($\text{Li}/(\text{Li}+\text{Na})_{\text{net}} = 0.26 \pm 0.01$, in Supplementary Figure 22b).

For the new platelet-like EG-LiFePO₄ particles, we noticed that there is ~ 23.5 at. % Li remained in the host after chemical Li-extraction process (Supplementary Table 10). The chemical route initially used

cannot removal Li^+ completely, indicating the existence of high defect level in the EG- FePO_4 particles which is consistent with their smaller capacity. It is worth noting that these trapped Li species are in the form of solid solution phases, as evidenced by the strong solid solution intensity band in Supplementary Figure 23a. Furthermore, the XRD patterns and the fitted phase fractions didn't change after Li recovery deintercalation (Supplementary Figure 23 and Supplementary Table 11), demonstrating the immobility of these defect-induced trapped Li during (de)intercalation. The EG- FePO_4 particle shows a slightly higher Li selectivity (Supplementary Figure 22b) after 20% seeding and without Li seeding ($\text{Li}/(\text{Li}+\text{Na})_{\text{net}} = 0.81 \pm 0.01$ for $\text{L}(0.2)_{4\text{C}}\text{-LN}(0.7)_{0.1\text{C}}$ and $\text{Li}/(\text{Li}+\text{Na})_{\text{net}} = 0.74 \pm 0.01$ for $\text{L}(0)\text{-LN}(0.7)_{0.1\text{C}}$), compared with Li pre-seeded Synthesized- FePO_4 particle ($\text{Li}/(\text{Li}+\text{Na})_{\text{net}} = 0.72 \pm 0.03$ for $\text{L}(0.2)_{4\text{C}}\text{-LN}(0.7)_{0.1\text{C}}$).

For the comparison among the three types of particles, the Li selectivity shows the same trend as the high-Li solid solution fractions for 20% seeded samples (Supplementary Figure 22b). However, the high-Li solid solution fraction to Li selectivity correlation across different particles might not follow the same linear relationship. Therefore, when comparing different particles, morphologies, sizes, and defect levels have to be taken into consideration since they can act together and play a complex role in determining the intercalation pathway and phase formation. Besides solid solution formation, some other aspects may also affect Li/Na selectivity. For example, coherency strain energy has different anisotropies and magnitude when changing from FePO_4 to Li_xFePO_4 or Na_xFePO_4 , dependent on the particle size and the particle morphology, which may also be a significant factor for Li competitiveness⁴⁰. At the current stage, we demonstrated that pre-creation of Li solid solution phases is an effective strategy to improve the Li selectivity beyond the intrinsic thermodynamic and kinetic material preference to Li. More systematic studies on other structural factors could bring new opportunities in the future to facilitate the Li extraction process.”

4) The authors should seriously refine the abstract and conclusions and clearly state that in addition to the proposed high Li solid solution phase(s) being the origin of the enhanced Li/Na selectivity during extraction of ions from the brine solution there other important aspects to be considered – like defect level of the FP host structure.

We thank the reviewer for the suggestion. We did revisions to the abstract and conclusion to reflect the discussion on the effect material characteristics on SS phase formation and Li selectivity.

For the Abstract part, we added the following sentence:

“Since solid solution formation pathway depends on particle characteristics (defect level, size, and morphology), particle control presents an additional avenue for dialing the Li selectivity further.”

For the Conclusion part, we added the following sentence:

“Moreover, particle characteristic is a critical factor in determining the solid solution formation, which affects co-intercalation behavior and Li selectivity. In our study, with a similar level of seeding, the solid solution fractions are higher in the platelet particles synthesized but lower for commercial ellipsoidal-shaped particles. Additionally, as the defect level in the particles increases, the solid solution fractions

increase, which leads to a higher Li selectivity. However, the influence of material characteristics (including morphologies, sizes, and defects) can be complex and requires future work to reveal their interplays.”

5) The authors should clearly state in the main text that the “Synthesized-FePO₄” and “EG-LiFePO₄” materials are not battery-grade materials – and that the observed high fractions of solid solutions after high-current pre-lithiation are specific property of those materials not shared with common battery-grade materials.

We thank the reviewer for the suggestion.

In the discussion of “Effects of particle characteristics on solid solution seeding and Li selectivity”, on page 13, we added:

“Comm-FePO₄, Synthesized-FePO₄, and EG-FePO₄ showed capacity of 135 mA/g, 147 mA/g, and 125 mA/g, respectively (See Supplementary Materials for capacity evaluation). It is worth mentioning that not all particles are good battery quality particles, especially EG-FePO₄. Their low delivered capacity indicates the high level of defects.”

We also comment on the defect level of EG-FePO₄ on page 14:

“For the new platelet-like EG-LiFePO₄ particles, we noticed that there is ~ 23.5 at. % Li remained in the host after chemical Li-extraction process (Supplementary Table 10). The chemical route initially used cannot removal Li⁺ completely, indicating the existence of high defect level in the EG-FePO₄ particles which is consistent with their smaller capacity.”

In the conclusion we added:

“Moreover, particle characteristic is a critical factor in determining the solid solution formation, which affects co-intercalation behavior and Li selectivity. In our study, with a similar level of seeding, the solid solution fractions are higher in the platelet particles synthesized but lower for commercial ellipsoidal-shaped particles. Additionally, as the defect level in the particles increases, the solid solution fractions increase, which leads to a higher Li selectivity. However, the influence of material characteristics (including morphologies, sizes, and defects) can be complex and requires future work to reveal their interplays.”

Suggestion

I suggest that the authors do serious re-work of the organization of the data in the paper and that they provide the additional explanations for the observations presented in Supplementary (as proposed above in the points 1-5). In addition to that the authors should provide clear answers to the exposed questions (especially the ones initiated in the pdf attachment). If they do not agree with this proposal, I recommend rejection.

We appreciate the reviewer’s suggestions to help improve the manuscript. We have addressed the above five points and modified the manuscript accordingly.

REVIEWER COMMENTS

Reviewer #3 (Remarks to the Author):

I greatly appreciate all the effort of the authors in providing additional explanations and clarifications of their experimental work, analyses performed, and general methodology applied. Moreover, I thank the authors for providing the missing part of the XRD data and the corresponding de-convolution information.

The authors have considerably revised the manuscript. Many aspects of the work are now much more clearly presented and more openly (even critically) discussed. Importantly, the role of probable (partially) defective structure of $\text{LiFePO}_4/\text{FePO}_4$ host is now briefly included in the discussion. More precise and detailed refinement analysis of the XRD data potentially confirming defected structure (e.g. Fe-Li antisite defects) is missing – but not critical for the main aim of this paper.

The main experimental technique employed by the authors in the present work is XRD. It seems that the corresponding de-convolution analysis presents some issues. The authors now provide additional and very important information. Namely, they expose the fact that the applied initial chemical delithiation of starting LFP material cannot completely remove Li, and further discuss that "...trapped Li species are in the form of solid solution phases..." (Supplementary Figure 23a). It has to be noted that this is a crucial piece of information that was missing in the previous versions of the manuscript. With this information it is now possible to understand many aspects of the present work.

The remaining "remanent" Li in Li_xFePO_4 host that represents residual Li being less accessible (or even partially trapped?) in the FePO_4 host structure among others now directly answers about the large magnitudes of solid-solution fractions and corresponding mismatch between calculated weighted sum of Li and the actual amount of Li being electrochemically inserted during seeding step. The authors have now provided the explanation: "...the mismatch of Li amount from XRD fitting to electrochemical seeding is mainly due to the trapped Li in the host". This is very important statement. It should be presented in the main text, since it is critical for a reader to understand why there are so large amounts of solid-solution found after the Li seeding step. And even more importantly to better understand the origin of the Li/Na selectivity enhancement.

I recommend just slight (final) rework of the paper:

- A) Based on the present findings provided by the authors it is obvious that the large portion of the Li solid-solution found (in XRD) after the Li seeding step is in fact due to the pre-existing "remanent" (maybe partially trapped) Lithium already present in the Li_xFePO_4 host before the seeding. The inability to fully delithiate starting Li_1FePO_4 particles via chemical route probably might be taken as indirect proof of defect structure of Li_xFePO_4 host material. Accordingly, the scheme in Figure 2 should be adopted to include and clearly show the initial state with partial (remanent, potentially trapped) Li in the Li_xFePO_4 structure representing important part of the total Li solid-solution after the seeding step.
- B) I kindly ask the authors to provide at the images of XRD deconvolution also the values of the solid-solution fractions in the Li_xFePO_4 material before the seeding. This will allow to directly observe what portion of the Li solid-solution has origin in the (remanent, potentially trapped) Li and what portion is additionally formed during the seeding.
- C) In the tables presenting the fitted phase fractions after Li seeding an additional column should be introduced to show the amount of remanent Li in the Li_xFePO_4 material before the seeding.
- D) A clear sentence should be included in the abstract and conclusions describing that both:
 - i) remanent Li in Li_xFePO_4 host and
 - ii) seeding step build up final Li solid-solution fraction

that induces enhanced Li/Na selectivity.

After this final modifications are done, I recommend the work to be published. I am sure the paper will be interesting to read and study both for researchers from battery field as well as from other disciplines. Finally, I would like to thank the authors for their willingness to refine the manuscript and provide additional data.

Response to reviewers' comments:

We would like to thank the reviewer for the valuable comments for improving our work. Please see the point-by-point response below. We have revised the manuscript based on the latest suggestions. The major changes in our revised manuscript have been marked in Red.

Reviewer #3:

I greatly appreciate all the effort of the authors in providing additional explanations and clarifications of their experimental work, analyses performed, and general methodology applied. Moreover, I thank the authors for providing the missing part of the XRD data and the corresponding de-convolution information.

The authors have considerably revised the manuscript. Many aspects of the work are now much more clearly presented and more openly (even critically) discussed. Importantly, the role of probable (partially) defective structure of $\text{LiFePO}_4/\text{FePO}_4$ host is now briefly included in the discussion. More precise and detailed refinement analysis of the XRD data potentially confirming defected structure (e.g. Fe-Li antisite defects) is missing – but not critical for the main aim of this paper.

The main experimental technique employed by the authors in the present work is XRD. It seems that the corresponding de-convolution analysis presents some issues. The authors now provide additional and very import information. Namely, they expose the fact that the applied initial chemical delithiation of starting LFP material cannot completely remove Li, and further discuss that “...trapped Li species are in the form of solid solution phases...” (Supplementary Figure 23a). It has to be noted that this is a crucial piece of information that was missing in the previous versions of the manuscript. With this information it is now possible to understand many aspects of the present work.

The remaining "remanent" Li in Li_xFePO_4 host that represents residual Li being less accessible (or even partially trapped?) in the FePO_4 host structure among others now directly answers about the large magnitudes of solid-solution fractions and corresponding mismatch between calculated weighted sum of Li and the actual amount of Li being electrochemically inserted during seeding step. The authors have now provided the explanation: “...the mismatch of Li amount from XRD fitting to electrochemical seeding is mainly due to the trapped Li in the host”. This is very important statement. It should be presented in the main text, since it is critical for a reader to understand why there are so large amounts of solid-solution found after the Li seeding step. And even more importantly to better understand the origin of the Li/Na selectivity enhancement.

I recommend just slight (final) rework of the paper:

A) Based on the present findings provided by the authors it is obvious that the large portion of the Li solid-solution found (in XRD) after the Li seeding step is in fact due to the pre-existing "remanent" (maybe partially trapped) Lithium already present in the Li_xFePO_4 host before the seeding. The inability to fully delithiate starting Li_1FePO_4 particles via chemical route probably might be taken as indirect prove of defect

structure of Li_xFePO_4 host material. Accordingly, the scheme in Figure 2 should be adopted to include and clearly show the initial state with partial (remanent, potentially trapped) Li in the Li_xFePO_4 structure representing important part of the total Li solid-solution after the seeding step.

Thanks for the reviewer's suggestion. We have updated the scheme in **Figure R1 (Figure R1, also as updated Figure 2 in the revised manuscript)**. We specified the initial state of the hosts as "Chemically Li extracted FePO_4 " and added note in the figure caption as

"Depending on the characteristics of the synthetic LiFePO_4 (defect level), there could be remnant Li after chemical extraction. See main text for more information."

Figure R1 Seeding and quantification of Li SS phases. **a**, Schematic showing high C rates Li seeding process and Li-Na co-intercalation process. The inset illustrates the possible intercalation pathways at the electrode-solution (E/S) interface. **Depending on the characteristics of the synthetic LiFePO_4 (defect level), there could be remnant Li after chemical extraction. See main text for more information.** **b**, Normalized XRD (dotted line: obtained; solid line: fitted) patterns of FePO_4 electrodes before ($\text{L}(0)$) and after seeding with different amounts of Li (10%, 20%, 30% and 40% of the 147 mAh/g total capacity) under 4C (588 mA/g), labeled as $\text{L}(0.1/0.2/0.3/0.4)_{4\text{C}}$. The normalization is based on the intensity of (020) peak for FePO_4 at 30.9° . The (020) peak of LiFePO_4 is centered at 29.8° . The intensity bands between the two end-up phases are the intermediate SS phases. **c**, An example of deconvoluted XRD pattern for the quantification of SS

phases and corresponding R-squared value (R^2). The obtained pattern (black dots) of $L(0.3)_{4C}$ is fitted with nine different phases of Li_xFePO_4 with $x = 0/0.125/0.250/0.375/0.500/0.625/0.750/0.875/1$, as calculated based on Vegard's law for the (211) and (020) of the $LiFePO_4$ (Green) and $FePO_4$ (Purple) end phases. See Supplementary Materials for more fitting details. **d**, Averaged accumulative SS phase fractions of $L(0/0.1/0.2/0.3/0.4)_{4C}$.

(Error bars representing the standard deviation of three replicate measurements.)

B) I kindly ask the authors to provide at the images of XRD deconvolution also the values of the solid-solution fractions in the Li_xFePO_4 material before the seeding. This will allow to directly observe what portion of the Li solid-solution has origin in the (remanent, potentially trapped) Li and what portion is additionally formed during the seeding.

We thank the reviewer for the suggestion. We added the XRD of $L(0)$ in **Figure R1b (also as Figure 2b in the revised manuscript)** and the accumulative phase fraction of $L(0)$ in **Figure R1d (also as Figure 2d in the revised manuscript)**. Additionally, we have included the XRD deconvolution pattern of chemically Li-extracted $FePO_4$ hosts before seeding in **Figure R2 (also as updated Supplementary Figure 5 in SI)** and added corresponding fitted phase fractions in **Table R1 (also as updated Supplementary Table 5 in SI)**. As shown in **Figure R3 (also as Supplementary Figure 6)**, the linearity between the calculated weighted sum of Li from XRD and the electrochemical seeded Li amount is still good with the $L(0)$ point.

We also updated the XRD deconvolution information of $L(0)$ on the flat glassy carbon substrate in **Figure R4, Table R2 and Figure R5 (also as Supplementary Figure 7, Supplementary Table 6 and Supplementary Figure 8 in SI)**. By improving the background contributions from the porous carbon cloth substrate, we can see from **Table R3** that ~ 0.07 Li per formula was left after the chemical extraction before the seeding process which could be attributed to defect-induced trapping of Li.

We also add the relating information in the main text as

“The fitted accumulative phase fractions for samples representing the starting and four seeding ranges are summarized in Figure 2d. The calculated weighted sum of Li from XRD fittings shows a good linear relationship with the electrochemical seeded Li amount (Supplementary Figure 5, Supplementary Table 5 and Supplementary Figure 6). However, Li amounts showed deviations at the low seeding percentage. For example, before Li seeding ($L(0)$), the weighted sum of Li is ~ 0.17 (Supplementary Table 5). This deviation could be from the remnant Li trapped in the hosts after chemical extraction or the contributions from the system substrate effect when the background intensity is unignorable. The deviations caused by porous structures of carbon cloth can be decreased by using flat glassy carbon as the substrate. As shown in Supplementary Figure 7, Supplementary Table 6 and Supplementary Figure 8, a better quantitative agreement between the calculated weighted sum of Li and the seeding amount is achieved. By eliminating the background intensity contributions, we can see from Supplementary Table 6 that ~ 0.07 Li per formula was left in the host after the chemical extraction step before the seeding process which could be from defect-induced Li trapping.”

Figure R2 Deconvoluted XRD patterns of (a) Chemically Li-extracted FePO₄ before seeding L(0), (b) L(0.1)_{4C}, (c) L(0.2)_{4C}, (d) L(0.3)_{4C}, (e) L(0.4)_{4C} and (f) L(0.5)_{4C} for Synthesized-FePO₄ particles on carbon cloth substrate from one representative sample.

Table R1 Fitted phase fractions of Figure R2 with the calculated weighted sum of Li.

	LFP	L0.875FP	L0.75FP	L0.625FP	L0.5FP	L0.375FP	L0.25FP	L0.125FP	FP	Intercalated Li	Weighted sum of Li
L(0)	0.04559	0	0.00688	0.07342	0.0824	0	0.11461	0.06579	0.61132	0	0.17471
L(0.1) _{4C}	0.07562	0	0.01964	0.04159	0.16232	0	0.14505	0	0.55578	0.1	0.23377
L(0.2) _{4C}	0.11218	0.00506	0.03933	0.06099	0.17481	0	0.11613	0.03272	0.45878	0.2	0.30475
L(0.3) _{4C}	0.14591	0.02684	0.01063	0.08277	0.18803	0	0.13107	0	0.41474	0.3	0.35589
L(0.4) _{4C}	0.16798	0.04583	0	0.12913	0.19123	0	0.10210	0.02974	0.33398	0.4	0.41365
L(0.5) _{4C}	0.19104	0.04145	0.07255	0.25751	0.02989	0.05795	0.04552	0.07888	0.2252	0.5	0.50058

Figure R3 Calculated weighted sum of Li from XRD fittings versus the electrochemically intercalated Li amount with the use of Synthesized-FePO₄ particles on carbon cloth. (Error bars representing the standard deviation of three replicate measurements.)

Figure R4 Deconvoluted XRD patterns of (a) Chemically Li-extracted FePO₄ before seeding L(0), (b) L(0.1)_{4C}, (c) L(0.2)_{4C} (d) L(0.3)_{4C}, and (e) L(0.4)_{4C} for Synthesized-FePO₄ particles on glassy carbon substrate from one representative sample.

Table R2 Fitted phase fractions of Figure R4 with the calculated weighted sum of Li.

	LFP	L0.875FP	L0.75FP	L0.625FP	L0.5FP	L0.375FP	L0.25FP	L0.125FP	FP	Low-Li SS	High-Li SS	Intercalated Li	Weighted sum of Li
L(0)	0	0.02175	0.02672	0	0	0	0.1191	0	0.83166	0.1191	0.04847	0	0.06962
L(0.1) _{4C}	0.01511	0.00562	0.08468	0	0	0	0.13104	0	0.76355	0.13104	0.0903	0.1	0.1163
L(0.2) _{4C}	0.06107	0.01466	0.08725	0.05607	0	0	0.14645	0	0.63449	0.14645	0.15798	0.2	0.21099
L(0.3) _{4C}	0.06909	0.05342	0.05712	0.12553	0	0.04728	0.11082	0.04485	0.49188	0.20295	0.23607	0.3	0.28817
L(0.4) _{4C}	0.08447	0.11453	0.02793	0.14864	0.04111	0.04378	0.12912	0.01258	0.39784	0.18548	0.33221	0.4	0.36936

Figure R5 (a) Calculated weighted sum of Li from XRD fittings versus the electrochemically intercalated Li amount with the use of Synthesized-FePO₄ particles on glassy carbon; (b) High-Li SS fractions (Li_xFePO₄, x = 0.500/0.625/0.750/0.875) and low-Li SS fractions (Li_xFePO₄, x = 0.125/0.250/0.375) under the same seeding rate 4C (588 mA/g) with different seeding ranges L(0/0.1/0.2/0.3/0.4)_{4C} collected on glassy carbon. (Error bars representing the standard deviation of three replicate measurements.)

C) In the tables presenting the fitted phase fractions after Li seeding an additional column should be introduced to show the amount of remanent Li in the Li_xFePO₄ material before the seeding.

We thank the reviewer for the suggestion. We have included the fitted phase fractions in **Table R1 (also as updated Supplementary Table 5 in SI)**.

D) A clear sentence should be included in the abstract and conclusions describing that both: i) remanent Li in Li_xFePO₄ host and ii) seeding step build up final Li solid-solution fraction that induces enhanced Li/Na selectivity.

We thank the reviewer for the suggestion.

We added the discussion of remanent Li in the abstract as

“Exploiting this phase separation, we increase the Na intercalation energy barrier by using partially filled 1D Li channels via non-equilibrium solid solution (SS) Li seeding or remnant Li in the solid solution phases.”

We added the discussion of remanent Li in the conclusion as

“Additionally, as the defect level in the particles increases, the trapped remnant Li amount increases. These trapped Li exists as solid solution phases and leads to a higher Li selectivity”

After this final modifications are done, I recommend the work to be published. I am sure the paper will be interesting to read and study both for researchers from battery field as well as from other disciplines. Finally, I would like to thank the authors for their willingness to refine the manuscript and provide additional data.

We appreciate the reviewer’s suggestions to help improve the manuscript.

REVIEWER COMMENTS

Reviewer #3 (Remarks to the Author):

It seems that we have come to kind of a compromise situation with the authors. The authors still insist in presenting their Li seeding step with high-Li solid-solution phase(s) buildup as being of the only phenomena of the crucial importance for the observed Li/Na selectivity of the Li_xFePO_4 host material.

In my personal opinion it is evident from the main experimental data (XRD: Fig. 2b-2d, Supplementary Fig. 5, Supplementary Table 5, Supplementary Fig. 7, Supplementary Table 6,) as well as from chemical analysis (ICP-MS, Supplementary Table 10) that after chemical extraction of Li from the initial LFP the obtained host material surely includes remaining (non-extracted) Li in the structure, thus the host should be accordingly labeled as Li_xFePO_4 . And it is not about the labeling itself but more about the physical significance – the defective nature of the starting LFP material affects the chemical extraction of Li with initial solid-solution formation, additional solid-solution buildup during the seeding step, and consequentially the Li/Ni selectivity.

For example Supplementary Scheme 1 completely neglects those facts and shows initial host structure as “empty FePO_4 ”. Moreover, in the previous round of review the authors were asked to: “Accordingly, the scheme in Figure 2 should be adopted to include and clearly show the initial state with partial (remanent, potentially trapped) Li in the Li_xFePO_4 structure representing important part of the total Li solid-solution after the seeding step.”

The authors should modify Figure 2 (and Supplementary Scheme 1) to more realistically show the principle applied in the present work. Host material for the seeding should be labeled as: “ Li_xFePO_4 after chemical extraction of Li (remnant quantity x of Li in the structure)”.

Response to reviewers' comments:

We highly appreciate the reviewer's valuable suggestions and insightful comments to help us improve the quality of our work. Please see the point-by-point response below. We have revised the manuscript based on the latest suggestions. The major changes in our revised manuscript have been marked in Red.

Reviewer #3:

It seems that we have come to kind of a compromise situation with the authors. The authors still insist in presenting their Li seeding step with high-Li solid-solution phase(s) buildup as being of the only phenomena of the crucial importance for the observed Li/Na selectivity of the Li_xFePO_4 host material.

In my personal opinion it is evident from the main experimental data (XRD: Fig. 2b-2d, Supplementary Fig. 5, Supplementary Table 5, Supplementary Fig. 7, Supplementary Table 6,) as well as from chemical analysis (ICP-MS, Supplementary Table 10) that after chemical extraction of Li from the initial LFP the obtained host material surely includes remaining (non-extracted) Li in the structure, thus the host should be accordingly labeled as Li_xFePO_4 . And it is not about the labeling itself but more about the physical significance – the defective nature of the starting LFP material affects the chemical extraction of Li with initial solid-solution formation, additional solid-solution buildup during the seeding step, and consequentially the Li/Ni selectivity.

For example Supplementary Scheme 1 completely neglects those facts and shows initial host structure as “empty FePO_4 ”. Moreover, in the previous round of review the authors were asked to: “Accordingly, the scheme in Figure 2 should be adopted to include and clearly show the initial state with partial (remanent, potentially trapped) Li in the Li_xFePO_4 structure representing important part of the total Li solid-solution after the seeding step.”

The authors should modify Figure 2 (and Supplementary Scheme 1) to more realistically show the principle applied in the present work. Host material for the seeding should be labeled as: “ Li_xFePO_4 after chemical extraction of Li (remnant quantity x of Li in the structure)”.

We appreciate the reviewer's suggestions. We have updated the scheme in **Figure R1 (also as updated Figure 2 in the revised manuscript)**. We specified the initial state of the hosts as “ Li_xFePO_4 after chemical Li extraction” and added note in the figure caption as

“The initial host was prepared by chemical extraction. x ' denotes the remnant quantity of Li in the structure. See main text for more information.”

We also updated the scheme in **Figure R2 (also as updated Supplementary Scheme 1 in the SI)**.

Figure R1 Seeding and quantification of Li SS phases. **a**, Schematic showing high C rates Li seeding process and Li-Na co-intercalation process. The inset illustrates the possible intercalation pathways at the electrode-solution (E/S) interface. The initial host was prepared by chemical extraction. x' denotes the remnant quantity of Li in the structure. See main text for more information. **b**, Normalized XRD (dotted line: obtained; solid line: fitted) patterns of FePO_4 electrodes before ($L(0)$) and after seeding with different amounts of Li (10%, 20%, 30% and 40% of the 147 mAh/g total capacity) under 4C (588 mA/g), labeled as $L(0.1/0.2/0.3/0.4)_{4\text{C}}$. The normalization is based on the intensity of (020) peak for FePO_4 at 30.9° . The (020) peak of LiFePO_4 is centered at 29.8° . The intensity bands between the two end-up phases are the intermediate SS phases. **c**, An example of deconvoluted XRD pattern for the quantification of SS phases and corresponding R-squared value (R^2). The obtained pattern (black dots) of $L(0.3)_{4\text{C}}$ is fitted with nine different phases of Li_xFePO_4 with $x = 0/0.125/0.250/0.375/0.500/0.625/0.750/0.875/1$, as calculated based on Vegard's law for the (211) and (020) of the LiFePO_4 (Green) and FePO_4 (Purple) end phases. See Supplementary Materials for more fitting details. **d**, Averaged accumulative SS phase fractions of $L(0/0.1/0.2/0.3/0.4)_{4\text{C}}$. (Error bars representing the standard deviation of three replicate measurements.)

Figure R2 Schematic illustration of Li-Na co-intercalation pathways for chemically Li extracted Li_xFePO_4 hosts (x' denotes the remnant quantity of Li in the structure due to the defect), low-rate Li pre-seeded hosts, and high-rate Li pre-seeded hosts.

REVIEWERS' COMMENTS

Reviewer #3 (Remarks to the Author):

The current version of the paper (together with the Supplementary data and further explanation provided therein) allows for the reader to understand the experimental procedures applied and (with some effort from his/her side) enables to recognize that Li/Na selectivity of Li_xFePO_4 host material is a combination of remnant (less accessible or potentially trapped) Li in Li_xFePO_4 and the high-rate electrochemical seeding of Li.

I would like to thank again the authors for their willingness to cooperate and provide the initially missing data, description, and for broadening of the extent of the explanations.

I would like to gently remind the authors that the main purpose of the scientific research should be the effort to reveal the reality/truth in the studied topic, since this is the only way to gain our understanding of the nature of the researched phenomena.

I recommend publishing of the work in the state as here presented in this round of review.